# Evolutionary Prediction Games

**Eden Saig**
Technion – Israel Institute of Technology
Haifa, Israel
edens@cs.technion.ac.il

**Nir Rosenfeld**
Technion – Israel Institute of Technology
Haifa, Israel
nirr@cs.technion.ac.il

## Abstract

When a prediction algorithm serves a collection of users, disparities in prediction quality are likely to emerge. If users respond to accurate predictions by increasing engagement, inviting friends, or adopting trends, repeated learning creates a feedback loop that shapes both the model and the population of its users. In this work, we introduce *evolutionary prediction games*, a framework grounded in evolutionary game theory which models such feedback loops as natural-selection processes among groups of users. Our theoretical analysis reveals a gap between idealized and real-world learning settings: In idealized settings with unlimited data and computational power, repeated learning creates competition and promotes competitive exclusion across a broad class of behavioral dynamics. However, under realistic constraints such as finite data, limited compute, or risk of overfitting, we show that stable coexistence and mutualistic symbiosis between groups becomes possible. We analyze these possibilities in terms of their stability and feasibility, present mechanisms that can sustain their existence, and empirically demonstrate our findings.

## 1 Introduction

Accurate predictions have become essential for any platform that supports user decision-making. Improvements in prediction accuracy often directly translate to better quality of service with benefits to both the platform and its users. This is one reason why modern platforms ranging from content recommendation and online marketplaces to personalized education and medical services have come to rely on machine learning as their backbone, and are investing much effort and resources in continually improving their predictions. However, while generally beneficial, promoting accuracy blindly can have unexpected, and in some cases undesired, consequences [9, 59, 24]. It is therefore important to understand the possible downstream and long-term effects of learning on social outcomes.

Conventional learning approaches aim to maximize accuracy on a given, predetermined data distribution. But in social settings, the distributions are composed of those users who *choose* to use the platform. When such choices depend on the quality of predictions, learning becomes a driver of user self-selection, and thus gains influence over its user population. This creates a feedback loop: model deployment shifts the population, and population changes trigger model retraining. We are interested in understanding the general tendencies and possible long-term outcomes of this process.

In particular, our focus is on user feedback dynamics in which accurate predictions encourage engagement or adoption, and prediction errors discourage them. Such dynamics arise across domains: accurate recommendations drive network-effect growth as users invite peers [39]; successful content strategies on social media are mimicked by others [4]; precise credit-risk models reduce premiums and therefore attract more clients to loan programs [32]; and medical providers with higher diagnostic accuracy draw more patients [3]. Common to the above is that individual choices adhere to some form of group structure in the population, and that in aggregate, these groups tend to grow when prediction quality is higher. Note that this notion of "group" is flexible, as groups can represent different demographics, behaviors, or roles, and with memberships being either inherent or chosen.

39th Conference on Neural Information Processing Systems (NeurIPS 2025).

Focusing on user choices as the primary driving force, we seek to analyze the impact of learning on group proportions over time. For this purpose, we adopt a novel evolutionary perspective and model the joint dynamics of learning and user choices through the lens of *natural selection*. Under the assertion that some degree of predictive error always exists, our key modeling point is that accuracy becomes, in effect, a *scarce resource* over which different groups in the population "compete". By associating each group's accuracy with its evolutionary fitness, we obtain an evolutionary process in which learning is the driver of selective pressure, and consequently, a determinant of long-term population outcomes. We can then ask questions regarding the long-term tendencies of the population composition, overall and per-group accuracies, and the affect of different modeling and algorithmic choices on temporal trends and long-term evolutionary outcomes.

**Contributions.**   Our main conceptual innovation is to analyze these feedback loops using a *population game*—a core component of evolutionary game theory useful for studying the dynamics of agents driven by local interaction rules [41, 27, 52]. Evolutionary game theory relates between game-theoretic properties of population games (e.g., Nash equilibria), and the dynamics of a wide variety of natural selection dynamics, such as imitation, word-of-mouth influence, or social learning, all of which are supported by our analysis. This makes the framework well-suited for our setting, as it allows us to address the common thread in the wide variety of feedback loops described above in a unified way.

Towards this, we first propose a novel game, called an *evolutionary prediction game*, in which evolutionary fitness is associated with prediction accuracy. This allows us to study the co-evolution of a (re)trained model and the population of its users. We then analyze the structure of games induced by different learning algorithms and their corresponding dynamics, and characterize the possible outcomes under different settings. Our results include conditions under which 'survival of the fittest' is a likely outcome; mechanisms that nonetheless enable coexistence; a discussion of the role and effects of retraining; and connections between survival, accuracy, and (evolutionary) stability. We also discuss connections to fairness and social conservation.

We conclude by complementing our analysis with experiments using both synthetic and real data and coupled with simulated dynamics. Our empirical results shed light on when and how certain user groups are likely to dominate, disappear, or coexist, and demonstrate how different design choices can shape social outcomes—even if inadvertently. Together, these highlight the importance of understanding, anticipating, and accounting for the long-term effects of learning on its user population.

## 1.1   Related Work

**Evolutionary game theory.**   Evolutionary game theory was originally developed as a framework for modeling natural selection processes in biology [44, 42, 41, 27, 56], and has since been applied to the analysis of large populations of myopic interacting agents in economics [52, 20], and to the study of online learning algorithms [19, 7]. From a technical standpoint, the games we propose and analyze are unique in this space as they are defined implicitly as solutions to statistical optimization problems. Our results for oracle classifiers reflect the principle of competitive exclusion [23], while our analysis of non-optimal classifiers relates to long-studied questions of coexistence [11].

**Performativity.**   Our work relates to the emerging field of *performative prediction* [48], which studies learning in settings where model deployments affect the underlying data distribution, with emphasis on (re)training dynamics and equilibrium. A central effort in this area is to identify general global properties (e.g., appropriate notions of smoothness) that guarantee convergence. Our work complements this effort by proposing *structure*, in the form of user self-selection [65, 28], which provides an efficient low-dimensional representation of the distribution map. This allows us to work in the notoriously challenging *stateful* performative setting [5], and to characterize stronger notions of stability.

**Fairness.**   The notion of equilibrium in evolutionary prediction games relates to the fairness criterion of *overall accuracy equality* [61]. The main distinction is that in prediction games, the question is not only *whether* groups are treated fairly, but also *which* groups are even considered, since fairness can be measured only for groups that are observed. Complementing other voiced concerns on long-term fairness outcomes [25, 40, 14, 49], our work adds a novel counterfactual perspective: if we observe fair outcomes at present, could this be because some groups were historically driven out of the game?

See Appendix B for an extended discussion and additional related literature.

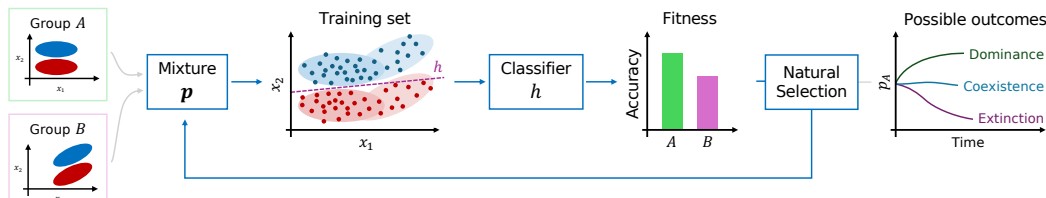

Figure 1: Natural selection in a two-group setting. Population is a mixture $\boldsymbol{p}$ of groups; Classifier $h$ is learned using data from the mixture distribution $D_{\boldsymbol{p}}$; Evolutionary fitness is associated with prediction accuracy; Differences in fitness drive change in mixture coefficients; Possible long-term tendencies are dominance (only $A$ survives), extinction (only $B$ survives), or coexistence (both survive).

## 2 Setting

The core of our setting is based on a standard supervised learning setup in which examples describe user data. Denote features by $x \in \mathcal{X}$ and labels $y \in \mathcal{Y}$. For a given distribution $D$ over pairs $(x, y)$, and given a training set $S = \{(x_i, y_i)\}_{i=1}^{n} \sim D^n$, the goal in learning is to use $S$ to find a predictor $h : \mathcal{X} \to \mathcal{Y}$ from a class $\mathcal{H}$ which minimizes the expected error under some loss function $\ell : \mathcal{Y} \times \mathcal{Y} \to \mathbb{R}$. For concreteness, we focus on classification tasks with the 0-1 loss, $\ell(\hat{y}, y) = \mathbb{1}\{\hat{y} \neq y\}$, whose minimization is equivalent to maximizing expected accuracy:

$$\text{acc}_D(h) = \mathbb{P}_{(x,y) \sim D}[h(x) = y] \tag{1}$$

Denote the learning algorithm by $\mathcal{A}(S)$, and by $h \sim \mathcal{A}(D)$ the process mapping distributions to learned classifiers, with randomness due to sampling. Since solving Eq. (1) is both computationally and statistically hard, practical learning algorithms often resolve to optimizing an empirical surrogate on the sampled data. One question we will ask is how such compromises affect long-term outcomes.

**Population structure.** As data points $(x, y)$ are generated by users, the data distribution $D$ represents a population. We assume the population is a mixture of $K$ groups. Each group $k \in [K]$ is associated with a group-specific distribution $D_k$ over $(x, y)$ pairs, which remains fixed, and with a relative size $p_k$, which evolves over time. The overall data distribution is a mixture denoted by $D = D_{\boldsymbol{p}} = \sum_k p_k D_k$, where $p_k$ is the current relative proportion of group $k$ (such that $\sum_k p_k = 1$ and $p_k \geq 0 \ \forall k$). Given some $h$, the marginal expected accuracy on group $k$ is $\text{acc}_k(h) = \text{acc}_{D_k}(h)$. The *population state* vector $\boldsymbol{p} = (p_1, \ldots, p_K)$ will be our main object of interest. As $\boldsymbol{p}$ encodes group proportions, the space of population states is the $(K-1)$-dimensional simplex $\Delta^K$, and since $\boldsymbol{p}$ determines $D_{\boldsymbol{p}}$, for brevity we denote $h \sim \mathcal{A}(\boldsymbol{p}) = \mathcal{A}(D_{\boldsymbol{p}})$ and $\text{acc}_{\boldsymbol{p}}(h) = \text{acc}_{D_{\boldsymbol{p}}}(h)$.

**Dynamics.** When a classifier is deployed, users make decisions according to the quality of the predictions they receive. Collectively, individual responses reshape the population composition $\boldsymbol{p}$, resulting in a *subpopulation shift* [62]. The updated population then serves as the training distribution for the next classifier—which forms a feedback loop between $\boldsymbol{p}$ and $h$. For training, we consider a simple procedure of repeated training on the current distribution. In terms of user behavior, our framework accommodates a wide family of dynamics, including reproduction, network effects, imitation, social influence, rational decision-making, and competition, as alluded to in Section 1. In Appendix C we define these formally and provide additional examples. In Section 3 we make our assumptions explicit.

**Outcomes.** Denote the initial population state by $\boldsymbol{p}^0$, and the initial classifier by $h^0 \sim \mathcal{A}(\boldsymbol{p}^0)$. We consider dynamics which evolve towards fixed points, and denote the corresponding point by $\boldsymbol{p}^*$. Denote the classifier by $h^* \sim \mathcal{A}(\boldsymbol{p}^*)$. We characterize fixed points in terms of three key properties:

- **Population composition:** For the population, dynamics may act as a force diminishing certain groups (i.e., driving towards a state satisfying $p_k^* = 0$ for some $k \in [K]$). In such cases, we say that the population is *dominated* by the remaining groups. When the population composition at rest has at least two groups (i.e., $p_k^* > 0$ for multiple values of $k$), we say that these groups *coexist*.
- **Accuracy:** For the classifier, dynamics may result in overall performance improving over time, (i.e., $\text{acc}(h^*) > \text{acc}(h^0)$), deteriorating, or remaining unchanged.

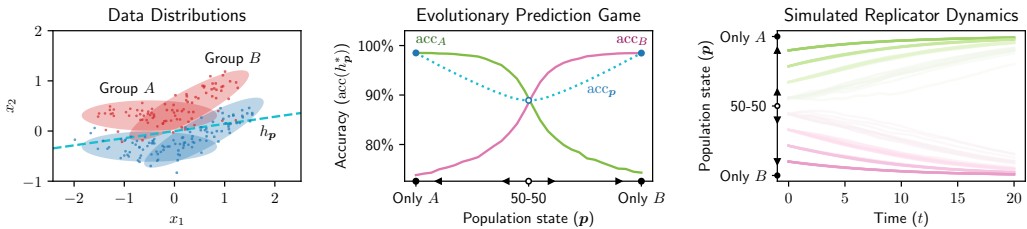

Figure 2: Evolutionary dynamics for two groups, induced by an oracle linear classifier (Section 4). **(Left)** Data distributions in feature space. Dashed line demonstrates the optimal linear classifier $h_{\boldsymbol{p}}$ for the uniform mixture $\boldsymbol{p} = (0.5, 0.5)$. **(Center)** Evolutionary prediction game $F_k(\boldsymbol{p}) = \mathrm{acc}_k(h_{\boldsymbol{p}})$. The game has two stable equilibria with single-group dominance, and one unstable coexistence equilibrium. Overall population accuracy $\mathrm{acc}_{\boldsymbol{p}}(h_{\boldsymbol{p}})$ is convex, and maximized at the boundries. **(Right)** Replicator dynamics induced by the game, for various initial states $\boldsymbol{p}^0$. Populations evolve towards fixed points with single-group dominance.

- **Stability:** For the joint system, some fixed points $\boldsymbol{p}^*$ are *stable* and attract all neighboring states, while other fixed points are unstable and may cause dynamics to diverge under small perturbations.

An illustration of the setting is provided in Figure 1.

## 3 Evolutionary Prediction Games

To analyze the feedback loop, we leverage *evolutionary game theory*, which is a general framework for the analysis of natural selection processes [42]. The framework models evolutionary interactions between individuals from $K$ groups in a large mixed population: Given a population state $\boldsymbol{p} \in \Delta^K$, each group $k \in [K]$ is associated with a scalar *fitness function*, $F_k(\boldsymbol{p})$, which quantifies the average evolutionary fitness of individuals in that group. Together, the fitness functions $F = (F_1, \dots, F_K)$ define a *population game* $F : \Delta^K \to \mathbb{R}^K$, which governs the dynamics of $\boldsymbol{p}$.

Formally, population games are symmetric normal-form games for a continuum of agents. Evolutionary game theory uses population games to map myopic, often local agent decisions to group-level dynamics governing the evolution of group proportions $\boldsymbol{p}$ over time. Modeling the game at the group level serves as useful abstraction for studying the general tendencies and possible long-term outcomes of local interactions in aggregate. Note that the notion of "group" is flexible and can support memberships that are either inherent (e.g., by demographics) or chosen (e.g., service provider).

**Accuracy as evolutionary fitness.** We formalize the interplay between learning and population dynamics through a novel type of population game—an *evolutionary prediction game*. The game associates the evolutionary fitness of each group with its expected marginal accuracy $\mathrm{acc}_k$ under a classifier $h$ trained on data sampled from the mixture $D_{\boldsymbol{p}}$:

**Definition 1** (Evolutionary Prediction Game). *Let $\mathcal{A}$ be a learning algorithm, and let $D_1, \dots, D_K \in \Delta(\mathcal{X} \times \mathcal{Y})$ be group distributions. Denote by $\mathrm{acc}_k(h)$ the marginal accuracy of group $k \in [K]$ under classifier $h \sim \mathcal{A}(\boldsymbol{p})$. The* evolutionary fitness *of group $k$ under population state $\boldsymbol{p}$ is:*

$$F_k(\boldsymbol{p}) = \mathbb{E}_{h \sim \mathcal{A}(\boldsymbol{p})}[\mathrm{acc}_k(h)] \tag{2}$$

*Together, the tuple $F(\boldsymbol{p}) = (F_1(\boldsymbol{p}), \dots, F_K(\boldsymbol{p}))$ defines an* evolutionary prediction game*.*

In the notation $F_k(\boldsymbol{p})$, we assume for brevity that the learning algorithm $\mathcal{A}$ and group distributions $D_k$ are clear from context, and we state them explicitly otherwise.

Evolutionary prediction games capture an implicit interaction between groups: while each group's fitness $F_k(\boldsymbol{p})$ is given by the accuracy of its own members, the classifier $h_{\boldsymbol{p}}$ is trained using data from the entire population. Thus, the evolution of different groups becomes coupled through the learning process. For example, Figure 2 (Left) illustrates a binary classification setting over two groups $\{A, B\}$. Figure 2 (Center) shows the corresponding evolutionary prediction game $F(\boldsymbol{p})$ induced by an oracle linear classifier (defined in Section 4). At $\boldsymbol{p} = (0.7, 0.3)$, we have $F_A(\boldsymbol{p}) \approx 0.97$ and $F_B(\boldsymbol{p}) \approx 0.78$, and thus the fitness of group $A$ is higher when the classifier is trained on data sampled from $D_{\boldsymbol{p}}$.

**Equilibrium.**    Nash equilibrium in general population games is defined as follows:

**Definition 2** (Nash equilibrium of a population game; e.g. [52])**.** *Let $F(\boldsymbol{p})$ be a population game, and denote* $\mathrm{support}(\boldsymbol{p}) = \{k \mid p_k > 0\}$. *A state* $\boldsymbol{p}^* \in \Delta^K$ *is a Nash equilibrium of $F(\boldsymbol{p})$ if it satisfies:*

$$\mathrm{support}(\boldsymbol{p}^*) \subseteq \arg\max_{k \in [K]} F_k(\boldsymbol{p}^*) \tag{3}$$

This also applies to evolutionary prediction games; e.g., the game in Figure 2 (Center) has three Nash equilibria at $\{(1,0), (0.5, 0.5), (0,1)\}$, and all population games have at least one equilibrium [52].

**Induced population dynamics.**    Formally, we encode dynamics as $\dot{\boldsymbol{p}} = V_F(\boldsymbol{p})$, where $\dot{\boldsymbol{p}}$ is the time derivative of $\boldsymbol{p}$, and $V_F : \Delta^K \to T\Delta^K$ is a tangent vector field induced by the game $F(\boldsymbol{p})$. Our results apply under mild structural assumptions on $V_F$, namely: (i) $V_F$ is continuous; (ii) the direction of flow is aligned with fitness, formally $V_F(\boldsymbol{p}) \cdot F(\boldsymbol{p}) > 0$ whenever $V_F \neq 0$, a property known as *positive correlation*; (iii) Fixed points of the dynamical system coincide with equilibria of $F(\boldsymbol{p})$, formally through either *Nash stationarity* or *imitative dynamics*. All are satisfied by the user-level dynamics highlighted in Section 1 and Appendix C, where they emerge as group-level properties in the large-population limit. Moreover, many of them converge towards the canonical *replicator equation*, $\dot{p}_k = p_k \left( F_k(\boldsymbol{p}) - \bar{F}(\boldsymbol{p}) \right)$, where $\bar{F}(\boldsymbol{p}) = \sum_{k'} p_{k'} F_{k'}$ is the average fitness across the population [57]. We also note that time-scales of convergence are determined (and can be adjusted) by the scale of $V_F$. See Appendix D.2 for formal definitions.

**Fairness.**    In the context of fairness, a classifier satisfies *overall accuracy equality* if all groups receive equal prediction accuracy [61]. We show that this criterion is satisfied in expectation by classifiers trained on equilibria mixtures of evolutionary prediction games (proof in Appendix D.3.2):

**Proposition 1.** *Let $F(\boldsymbol{p})$ be an evolutionary prediction game induced by a learning algorithm $\mathcal{A}$, and let $\boldsymbol{p}^*$ be a Nash equilibrium. Then $h \sim \mathcal{A}(D_{\boldsymbol{p}^*})$ satisfies overall accuracy equality in expectation.*

As outcomes of the dynamics correspond to equilibria of a game, this lends to a natural form of emergent fairness. However, note that this criterion only applies to groups that appear in the data: in Section 6.3, we further show that a system may appear as fair due to other groups being driven out.

**Heterogeneous fitness.**    Finally, in some settings, incentives or outside alternatives may vary across groups. This may be captured using the notion of *retention functions* [25], which translate marginal prediction accuracy to the tendency to remain engaged. In such cases, the transformed fitness function becomes $F_k(\boldsymbol{p}) = \nu_k \left( \mathbb{E}_{h_{\boldsymbol{p}}}[\mathrm{acc}_k(h)] \right)$, where $\nu_k : [0,1] \to \mathbb{R}$ is a group-dependent, continuous strictly-increasing function. A simple example is $\nu_k(x) = x - b_k$ (for $b_k > 0$) which allows groups to differ in the minimum accuracy their users require to keep using the system. All of our results apply to any set of strictly monotone $\nu_k$ for $K = 2$, and to positive affine $\nu_k$ with per-group offsets $b_k$ for $K > 2$. For clarity we state our main results for $\nu_k(x) = x$, but also discuss them for broader $\nu$.

## 4    Competitive Exclusion Under Oracle Classifiers

To understand how populations and classifiers evolve jointly, we start by characterizing natural selection in an 'ideal' setting, in which a classifier is trained repeatedly using unlimited resources:

**Definition 3** (Oracle classifier)**.** *Let $\mathcal{H}$ be a hypothesis class, and let $D$ be a data distribution. An* oracle classifier *with respect to $\mathcal{H}$ is a minimizer of the 0-1 loss in expectation over $D$:*

$$h^{\mathrm{opt}} \in \arg\min_{h \in \mathcal{H}} \mathbb{E}_{(x,y) \sim D}[\mathbb{1}\{h(x) \neq y\}] \tag{4}$$

Oracle classifiers represent ERM classifiers in the population limit. This regime abstracts away the complexities due to estimation and approximation errors of practical learning algorithms (i.e., algorithms that learn from finite data in reasonable time). Nonetheless, analysis of the dynamics remains challenging due to the $\arg\min$ operator. We focus on settings in which $h^{\mathrm{opt}}$ exists, and denote by $\mathcal{A}^{\mathrm{opt}}(D)$ the (theoretical) learning algorithm which returns the oracle classifier with respect to $D$.[1] We assume that fitness functions are continuous, and tie-breaking is consistent. To simplify presentation, we first assume that each group has a distinct oracle classifier, and then address the general case. Our central result characterizes natural selection induced by oracle classifier retraining:

---

[1]The Bayes-optimal classifier $h^{\mathrm{Bayes}}(x) = \arg\max_y p(y \mid x)$ is a special case of this definition for $\mathcal{H} = \mathcal{Y}^{\mathcal{X}}$.

**Theorem 1.** *Let $\mathcal{H}$ be a hypothesis class, and denote by $\mathcal{A}^{\mathrm{opt}}$ the oracle learning algorithm with respect to $\mathcal{H}$. Assume that at each population state $\boldsymbol{p}$ the oracle classifier $h_{\boldsymbol{p}} \sim \mathcal{A}^{\mathrm{opt}}(D_{\boldsymbol{p}})$ is deployed. Then it holds that:*

1. ***Accuracy:** Overall accuracy increases over time, $\frac{\mathrm{d}}{\mathrm{d}t}\mathrm{acc}_{\boldsymbol{p}}(h_{\boldsymbol{p}}) \geq 0$.*

2. ***Stability:** A stable equilibrium always exists, and there can be multiple such equilibria.*

3. ***Competitive exclusion:** For all stable equilibria, $|\mathrm{support}(\boldsymbol{p}^*)| = 1$.*

4. ***Coexistence:** Equilibria with $|\mathrm{support}(\boldsymbol{p}^*)| \geq 2$ may exist, but are unstable.*

**Proof sketch.** The proof leverages structural properties that emerge despite the complexity of the learning problem. Our main technical tool is the framework of *potential games*, which are population games that can be expressed as a gradient of a scalar potential function defined over the simplex. We start by showing that the expected accuracy of any fixed classifier is linear in $\boldsymbol{p}$, then the core of the proof is a convexity argument: the optimality of $\mathcal{A}^{\mathrm{opt}}$ implies that $\mathrm{acc}_{\boldsymbol{p}}(h_{\boldsymbol{p}})$ is convex as a pointwise maximum of linear functions, with gradients given by marginal accuracies. From this we identify $f(\boldsymbol{p}) = \mathrm{acc}_{\boldsymbol{p}}(h_{\boldsymbol{p}})$ as a potential function, and leverage the known correspondence between equilibria of population games and local extrema of potential functions. Finally, distinct oracle classifiers imply strict convexity of $f(\boldsymbol{p})$, and the stability of single-population states follows from the fact that convex functions over the simplex are locally-maximized at the vertices, and the accuracy condition follows from the fact that $f(\boldsymbol{p})$ is a Lyapunov function. Full proof appears in Appendix D.5.

**Extensions.** To simplify presentation, the statement of Theorem 1 assumes that the oracle classifiers for each group are distinct. When this assumption doesn't hold (e.g., when $D_k = D_{k'}$ for some $k \neq k'$), a generalized version of the theorem applies: rather than single-group stable equilibria, dynamics converge towards convex combinations of groups that share an identical oracle classifier with identical marginal accuracy (See Appendix D.5.1). For heterogeneous fitness where retention functions $\nu_k$ may differ across groups, we show that the retention functions we support induce potential games with transformed potential functions. Survival is determined according to the maximizers of the transformed potential function, and the quantity that increases over time becomes the average fitness $\sum_k p_k \nu_k \left(\mathrm{acc}_k(h_{\boldsymbol{p}})\right)$ (See Appendix D.5.2). Finally, we also note that a similar characterization also holds for classifiers trained once at the outset (See Appendix D.4).

**Interpretation.** Theorem 1 suggests that for repeatedly trained oracle classifiers, natural selection will amplify the dominance of groups having the highest fitness at the outset. From an evolutionary perspective, this reflects the *competitive exclusion principle* [23], which states that natural selection tends towards exclusive survival of the fittest when multiple species compete over the same resources. Stability implies that once a group dominates, other groups will remain excluded. While this is the general trend, coexistence is nonetheless *enabled* by retraining, in the form of possible equilibria in which multiple groups survive. That is, there may exist population states $\boldsymbol{p}^*$ such that $p_k^* > 0$ for several groups $k$, and for which this remains to hold when the classifier is retrained. The crux, unfortunately, is that such points are *evolutionarily unstable*, in the sense that small perturbations around $\boldsymbol{p}^*$ will push dynamics away. This implies that optimal retraining—unconstrained by limited data and compute—induces a strong form of competition between groups. But the existence of mixed equilibria suggests that other algorithms might be able to encourage symbiosis, which we explore next.

## 5 Avenues for Coexistence

Intuitively, oracle classifiers drive dynamics away from mixture equilibria because the learning objective is fully aligned with, and so reinforces, self-selection. This motivates the natural question: Can repeated training drive dynamics also towards coexistence, and when? Here we illustrate two ways for symbiosis to arise: as a result of inherent limitations of learning in practice (i.e., limited data or compute), and by applying an ad-hoc, dynamics-aware learning algorithm designed to stabilize outcomes.

**A context-dependent view on coexistence.** It is important to emphasize that coexistence in our setting is neither innately 'good' or 'bad', but rather, its desirability depends on context. Hence, the question of whether coexistence *should* be encouraged depends on the task at hand, the nature of user choices, and the goals of the system designer. For instance, recommendation platforms may

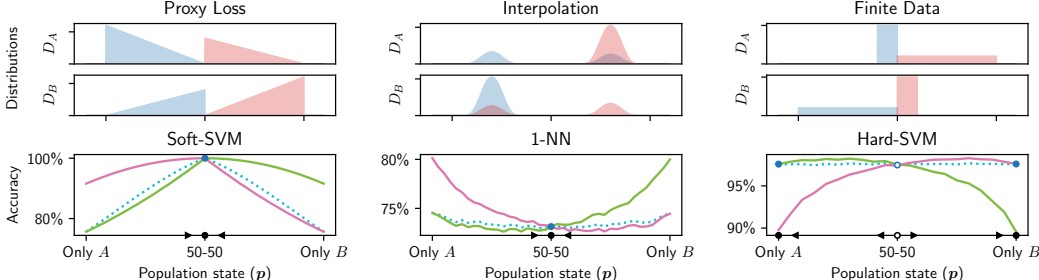

Figure 3: Graphical illustration of theoretical coexistence results. Top row presents data distributions, bottom row shows the corresponding prediction games. **(Left)** Stable and mutualistic coexistence induced by the use of a proxy loss (Theorem 2). **(Center)** Stable coexistence induced by interpolation (Appendix D.7). **(Right)** Mutualistic coexistence induced by a finite training set (Appendix D.8).

aim to preserve both content and audience diversity, making coexistence desirable. Conversely, in consumer markets having multiple competing standards (e.g., video encoding format, compression schemes, payment platforms), there are benefits to converging on the single best alternative. This context-dependent stance aligns well with the notion of evolution, which in itself is neutral.

## 5.1 Learning in Practice

While the goal in learning is to minimize the expected 0-1 loss, in practice learning often resorts to optimizing proxy objectives over sampled data via empirical risk minimization. Here we show that, perhaps surprisingly, basic aspects of this approach—namely the use of surrogate losses, access to finite data, and interpolation methods—can create conditions that facilitate *mutualism*, in which all groups benefit from coexistence. For each aspect, we demonstrate through a carefully crafted example how it can act as a mechanism that enables stable coexistence. Here we focus our discussion on surrogate loss as a representative construction; for finite data and memorization, we illustrate the underlying distribution and induced game in Figure 3, and defer the formal analysis to Appendix D.6.

**Coexistence by surrogate loss.** A common approach to empirical risk minimization is to replace the 0-1 loss with a convex surrogate loss, such as the *hinge loss* used in the classic Soft-SVM algorithm [13]. This enables optimization, but introduces bias: whereas the 0-1 loss applies uniform penalties, surrogates penalize misclassified points relative to their distance from the decision boundary of $h$. Our next result uses this artifact to construct a game having two groups that are complementary in their incurred bias, and hence balance each other to induce mutualistic coexistence.

**Theorem 2.** *There exists an evolutionary prediction game in which learning with the hinge loss and $\ell_2$ regularization induces a mixed equilibrium that is both stable and fitness-maximizing.*

The construction is outlined in Figure 3 (Left), and proof is provided in Appendix D.6. The proof leverages the bias of hinge loss against minority classes: For each group alone, minimizing the hinge loss biases the learned classifier against the minority class, which is suboptimal. But since each group has a different minority class, these biases cancel out at the mixed equilibrium. The surprising property is that this state is also evolutionarily stable: when one group grows slightly (i.e., $p_1 = 0.5 + \epsilon$), it loses more from the fact that the other group shrinks (i.e., $p_2 = 0.5 - \epsilon$) than it gains from its own growth. In this sense, each group needs the other to maintain its existence, giving rise to stable and mutualistic symbiosis. Appendix D.6.3 expands on this idea to show that varying the amount of regularization leads to *bifurcations*, in which the game transitions between 3, 1, and 5 equilibria.

**Interpretation.** In the example game above, as in those we construct for finite data (Appendix D.8; Figure 3 (Right)) and interpolation (Appendix D.7; Figure 3 (Center)), mutualism is enabled by how user self-selection compensates for the algorithm's imperfections. When the resulting bias is complementary across groups, this can give rise to (implicit) cooperation. One interpretation is that the existence of each group acts as regularization on others: from the perspective of group $k$, the objective can be written as $\arg\max_h \mathrm{acc}_k(h) + \lambda R(h)$ where $R(h) = \sum_{j \neq k} \mathrm{acc}'_k(h)$ is a data-dependent regularizer with coefficient $\lambda = \frac{1}{p_k} \sum_{j \neq k} p_j$. Nonetheless, our results should not

be taken to imply that proxy losses or finite data are a good *means* to achieve coexistence; rather, they highlight how mutualism can organically materialize when learning affects different groups differently. We explore this further empirically in Section 6.2.

## 5.2 Stabilizing Coexistence Equilibria

Thus far we have considered evolutionary prediction games in which learning pursues the conventional objective of maximizing accuracy at each timestep on the current distribution. But if the learner is aware of population dynamics and of how the learned classifier may influence outcomes, then it makes sense to consider learning algorithms that take this into account. The question of course is how.

Here we focus on a particular aspect of this general question that arises from our last result in Sec. 4, namely: given a system with a desirable but unstable mixed equilibrium $p^*$, how do we stabilize it? We propose a conceptually simple solution that works by reweighing examples in the standard accuracy objective in a way that accounts for dynamics. The main idea is to invert the natural tendency of dynamics to push away from the unstable equilibrium, and instead pull towards it, by training 'as if' the distribution was at a different state $p'$ than its actual current state $p$:

**Proposition 2.** *Let $\mathcal{A}^{\mathrm{opt}}(p)$ be an oracle algorithm with equilibrium $p^*$. If $p^*$ has full support (and thus is unstable), then it becomes stable under $\mathcal{A}'(p) = \mathcal{A}^{\mathrm{opt}}(2p^* - p)$.*

Proof in Appendix D.9. The idea is that $\mathcal{A}'$ trains 'as if' the state is $2p^* - p$, which can be achieved by weighing examples from each group $k$ by $w_k = \frac{p_k}{2p_k^* - p_k}$. This inverts the natural tendency of dynamics to push away from the unstable equilibrium, and instead pull towards it. The statement regards optimal classifiers and requires knowledge of $p^*$, but in practice we can set $w$ according to an estimated $\tilde{p}$. We further explore this approach in Section 6.1.

# 6 Experiments

To demonstrate how some of the principles we have discussed so far apply in more realistic settings, we now turn to explore evolutionary prediction games empirically using real data and simulated dynamics. We include three experiments. The first experiment complements our results on Sec. 5.2 by demonstrating how stabilization can improve both overall and per-group accuracies via coexistence. The second complements Sec. 5.1, showing how mutualism can arise organically due to group balancing, here from complementarities in label noise. The third experiment explores fairness under population dynamics (Prop. 1) to illustrate the limitations of static fairness measures.

## 6.1 Mutualistic Coexistence From Data Augmentation

Data augmentation is a method for enriching the training set with class-preserving transformations of inputs (e.g., camera angle, source of lighting) [e.g., 55]. Augmentations are typically considered as artificial constructs, but there are also cases where they emerge naturally (e.g., medical images from different hospitals or imaging devices). Here we show that when different sources correspond to different groups, data pooling can serve to foster beneficial coexistence in which all groups gain from the existence of others. This suggests that 'natural' augmentations can be effective also in the long run.

**Method.** We use the CIFAR-10 image recognition dataset [36], which consists of 60,000 32x32 color images in 10 classes, with 6,000 images per class. As horizontal image flips are considered class-preserving for this dataset, group $A$ consists of images sampled directly from CIFAR-10, and group $B$ consists of images under horizontal flip. We use a ResNet-9 network for prediction [26], and train it using the `ffcv` framework with default optimization parameters [38]. For each $p$, we measure the prediction model's accuracy on the original images from the CIFAR test set (representing $\mathrm{acc}_A(h_p)$), and on their flipped counterparts (representing $\mathrm{acc}_B(h_p)$). For stabilization, we use the method described in Section 5.2 with $p^* = (0.5, 0.5)$. We simulate the evolutionary process using discrete replicator dynamics, and use linear interpolation to determine intermediate fitness values.

**Results.** Figure 4 (Left) presents the estimated evolutionary prediction game, averaged across 20 repetitions. The game has three equilibria: Two stable equilibria with single-group dominance ($92.6 \pm 0.1\%$ accuracy), and an unstable coexistence equilibrium ($93.5 \pm 0.1\%$ accuracy). Although fitness

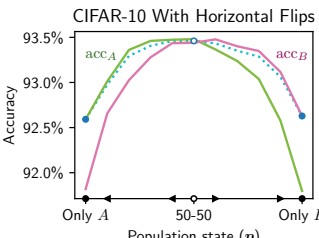 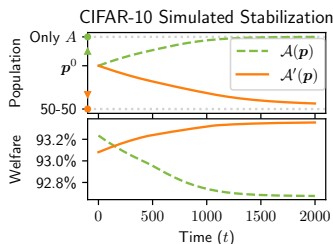 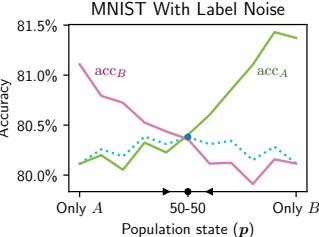

Figure 4: Empirical evaluation in two-group settings. **(Left)** Game induced by CIFAR-10 with Resnet-9 and groups representing horizontal flips. The game has unstable mutualism (Section 6.1). **(Center)** Simulated stabilization of replicator dynamics on the CIFAR-10 game (Section 5.2). **(Right)** Game induced by MNIST with label noise, showing stable mutualism (Section 6.2).

is typically higher for the larger group, the mixed equilibrium attains the highest overall population accuracy, suggesting that each groups benefits from the existence of the other. Reaching this point however requires stabilization; Figure 4 (Center) shows how our algorithm from Section 5.2 is able to attain stable beneficial coexistence, despite using an empirical proxy objective. Appendix F includes additional results on sampling noise, time to convergence, and sensitivity analysis for stabilization.

## 6.2 Stable Coexistence From Overparameterization

Modern neural networks are frequently overparameterized, and are known to achieve strong test-time performance despite memorizing the training data [e.g., 64]. Recent research attributes this phenomenon to the implicit biases introduced by the training process [e.g., 60]. Building on these ideas, here we demonstrate that stable coexistence can emerge when an overparameterized neural network is trained on data in which groups have complementing label noise. Our construction considers a multi-class settings where each group includes a majority of *noisy* examples from some class, and a minority of clean examples from other classes. This aims to captures settings in which user groups (e.g., by country) differ in the marginal distribution $p(y)$ over classes (e.g., spoken language), and where the dominant class is more likely to be misclassified (e.g., due to many accents).

**Method.** We use the MNIST dataset, set $K = 2$, and split the classes unevenly between two groups: Group $A$ is biased towards even digits $\{0, 2, 4, 6, 8\}$, whereas group $B$ is biased towards odd digits, both with a 4:1 imbalance. For each group, we introduce label noise to the majority classes, mapping the true label of each digit $d$ to the next digit with same parity with probability $0.2$ (i.e. for group $A$, label noising stochastically maps $0 \mapsto 2, 2 \mapsto 4$, etc.). We train a convolutional neural network for 200 epochs using stochastic gradient decent with momentum. The training and test sets are split independently, and for each $p$ we measure the prediction model's accuracy on both splits of the test sets (representing $\mathrm{acc}_A(h_p)$ and $\mathrm{acc}_B(h_p)$). Under the given parameters, the label noising process leads to an accuracy upper bound of $84\%$ assuming perfect recognition.

**Results.** Figure 4 (Right) shows the estimated evolutionary prediction game, averaged across 50 repetitions. Note how label noise has 'flipped' the game in that accuracy is higher for the minority group (i.e., $\mathrm{acc}_B > \mathrm{acc}_A$ when $p_B < p_A$, and vice versa). This indicates that groups naturally balance each other in this setting. As a result, the game has a stable coexistence equilibrium, with $80.4 \pm 0.2\%$ test accuracy (vs. the theoretical upper bound of $84\%$ due to label noise). Training accuracy is $98.7 \pm 0.1\%$, suggesting that interpolation is indeed the likely phenomenon at play. See Appendix D.7 for further discussion on how interpolation facilitates coexistence as an underlying mechanism.

## 6.3 Population Dynamics and Fairness

Lastly, we examine how population dynamics interact with notions of fairness. Our goal here is to show that a system can appear to be fair at a given point in time—but only because some groups have been driven out of the game, and fairness is measured on the remaining observable groups. Our message here is that fairness should consider not only what has happened, but also what *could* have happened, particularly with respect to groups that perhaps had previously been excluded.

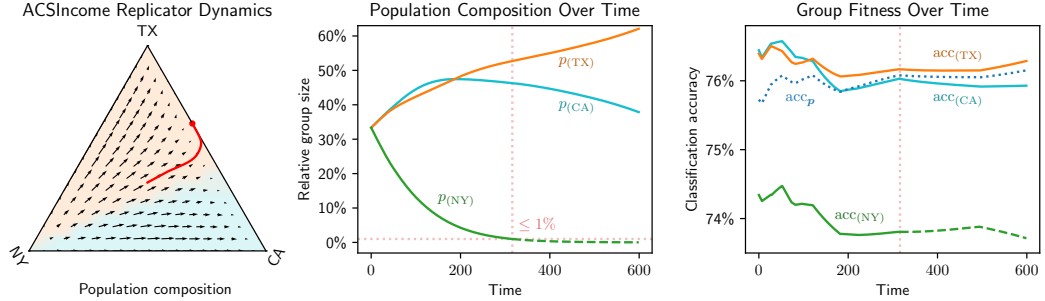

Figure 5: Evolutionary dynamics in a three-group setting, induced by a linear SVM trained on the ACSIncome dataset (Section 6.3). **(Left)** Replicator dynamics on the three-group simplex. Red line indicates a trajectory starting from the uniform mixture, and background colors indicate basins of attraction. **(Center)** Population composition over time for the same trajectory. Proportion of NY users becomes negligible at $t \approx 316$. **(Right)** Fitness over time. Effective group disparity is significant at the outset, but shrinks by an order of magnitude after NY users are driven out.

**Method.** We use a subset of the Folktables ACSIncome dataset [17], a tabular income-level classification task based on US census data. We associate groups with data from three states in the US, with the idea that firms operate in different states but train a model on data aggregated from all sources. Specifically, we set $K = 3$, and use 2018 census data from the states of California (195,665 datapoints), New York (103,021), and Texas (135,924). We use linear SVM as the learning algorithm, and randomly hold out 5,000 points from each state for evaluating accuracy. To compute fitness at a given population state $p$, we sample a dataset of size $n = 1000$ from the mixture, train a classifier, and measure group accuracies using the held-out sets. Results are averaged over 10 repetitions to obtain mean fitness values, and we simulate replicator dynamics starting from the uniform population state.

**Results.** Figure 5 shows how evolution transitions between two dynamical phases: an initial phase (up to time step $\sim$200) where one group (NY) diminishes while the other two (CA,TX) remain balanced with similar accuracies, and a second phase (from time step $\sim$300) in which the two remaining groups compete until one of them dominates. This illustrates a possible shortcoming of static fairness definitions: At the onset, all groups are present, but disparity in accuracy is $\approx 2\%$. As time progresses, the system becomes more "fair" — but only because the first group has been suppressed (i.e., its relative size is $\leq 1\%$). Auditing the prediction algorithm for fairness at this point in time creates an appearance of fairness. This holds only for the two groups that remain (disparity $\approx 0.2\%$), and there is insufficient observed data to help identify the preceding existence of a missing group. Appendix F.4 provides additional support, exploring similar phenomena under different learning algorithms.

# 7 Discussion

This work introduces evolutionary prediction games, a unified framework for analyzing self-selection feedback loops between learning algorithms and user populations. Our analysis reveals a gap: in idealized settings, repeated training drives competitive exclusion, whereas practical constraints enable a rich variety of long-term outcomes, including stable coexistence and mutualistic interactions.

The study of population dynamics is central to biology and ecology, a task for which population games have proven to be highly effective. We believe there is need to ask similar questions regarding the dynamics of social populations, as they become increasingly susceptible to the effects of predictive learning algorithms. The social world is of course quite distinct from the biological, but there is nonetheless much to gain from adopting an ecological perspective. One implication of our results is that, without intervention, learning might drive a population of users to states which would otherwise be unfavorable. Ecologists address such problems by studying and devising tools for effective conservation. Similar ideas can be used in learning for fostering and sustaining heterogeneous user populations. Our work intends to stir discussion about such ideas within the learning community.

## Acknowledgements

The authors would like to thank Mor Nitzan, Oren Kolodny, Ariel Procaccia, Fernando P. Santos, Jill-Jênn Vie, Evyatar Sabag, and Anonymous Reviewers for their insightful remarks and valuable suggestions. Nir Rosenfeld is supported by the Israel Science Foundation grant no. 278/22. Eden Saig is supported by the Israel Council for Higher Education PBC scholarship for Ph.D. students in data science.

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

# A Broader Impact

Our work seeks to shed light on the question: how does the deployment of learned classifier impact the long-term composition of the population of its users? Posing this question as one of equilibrium under a novel population game, our results describe the possible evolutionary outcomes under simple conditions (training once), optimal conditions (perfect classifiers), and realistic settings (namely empirical surrogate loss minimization). The fact that learning can influence population dynamics can have major implications on social and individual outcomes. Our work intends to illuminate this aspect of learning in social contexts, and to provide a basic and preliminary understanding of such interactions.

**Scope and limitations.** As the bulk of our results are theoretical in nature, naturally they rely on several assumptions. Some regard the learning process: for example, we consider only simple accuracy-maximizing algorithms that optimize over a fixed-size training set sampled only from the current (and not past) data distribution. Others concern dynamics: for example, we assume no exogenous forces (i.e., which could prevent extinction), intra-group distribution shifts, or inter-group dependencies (other than those indirectly formed through the classifier). Whether our results hold also when these assumptions are relaxed is an important future question.

It is also important to interpret our results appropriately and with care. For example, statements which establish 'extinction' as a likely outcome apply only limiting behavior, and so are silent on trajectories, rates, or any finite timepoint. Our results also rely on users being partitioned into non-overlapping groups. In reality, group memberships are rarely exclusive (or even fixed), and the dependence of individual behavior on group outcomes (in our case, marginal group accuracy—which defines fitness) is unlikely to be strict.

The above points should be considered when attempting to draw conclusions from our theoretical results to real learning problems. However, and despite these limitations, we believe the message we convey—which is that learning in social settings *requires* intervention and conservation to ensure sustainability of social groups and individuals—applies broadly and beyond our framework's scope.

**Diversity vs. homogeneity.** Our results suggests that systems of the type we analyze will tend towards less diverse population compositions. Nonetheless, and as we state throughout, our results should be taken as describing the *tendency* of population dynamics — when selective pressure is the primary force, when the learning algorithm is optimal retraining, absent any interventions, and at the limiting equilibrium.

Of course, in reality, many systems are diverse. Possible explanations for why a particular system at a given point in time can sustain several groups are:

- **Time**: dynamics have not yet reached equilibrium, or converge very slowly.

- **Learning**: the algorithm is not optimal (e.g., uses a proxy or finite data), or is designed to balance groups (e.g., via steering, regularization, or robust training).

- **Intervention**: system administrators may intentionally intervene to facilitate diversity; for example, consider how firms invest significant resources in marketing (e.g., Meta reports investing about \$2B per year[2]) to ensure a constant influx of users from diverse social groups.

- **Dynamics**: Other forces are at play, such as lack of alternatives, cross-group influence, rigidity of demand, non-linear heterogeneous utility, etc.

Nonetheless, an important but subtle point is that when we observe a system consisting of several groups, we do not observe other groups that could have been supported, as they may have already been driven out. Thus, what we perceive as a mixed state may actually be a snapshot of a process in which "weaker" groups have already vanished, with others possibly to follow.

In biology, the question of coexistence is prevalent, despite consensus that natural selection is the primary force at play. The study of coexistence therefore focuses on how it is sustained despite natural selection. This has revealed many biological and ecological mechanisms that can facilitate coexistence, such as resource partitioning, environment variation, and stabilization via intermediates. In this work we focus on learning itself as a possible stabilizing mechanism (Sections 5,6), which we

---

[2]Meta Platforms, Form 10-K - Annual report, 2024, SEC Accession No. 0001326801-25-000017. "Advertising Expense".

view as a first step towards the study of coexistence in this context (as also noted by Reviewer oLKx). We also believe that prediction games in particular, and evolutionary game theory more generally, can aid future research in exposing additional mechanisms that can facilitate diversity in environments with learning.

**Practical implications.** Evolutionary prediction games can help designers and operators of ML systems anticipate performative population shifts, and integrate this foresight into their decision-making. One key message is that applying conventional retraining risks favoring some groups over others in the long term, even if unintentionally. Evolutionary prediction games can help understand which groups are at risk, why they are at risk, and highlight possible avenues for interventions, if such are needed. One example is to decide on using a stabilizing learning algorithm (such as the one we propose in Prop. 2) instead of retraining to help steer towards desired states. Another is to target the retention of at-risk groups, either by providing benefits or reducing costs (e.g., via subsidies), in order to improve their fitness. A third is to understand if and when to increase the influx of certain groups, e.g. via marketing efforts, to balance the natural tendencies of the system. We will extend the discussion to emphasize these insights.

# B  Further Discussion of Related Work

## B.1  Performative Prediction

**Stateful performative prediction.** Performative prediction [48] is a framework for modeling the effects of predictive models on the data distributions they are tested and trained on (see [24] for a recent survey). Dynamics induced by evolutionary prediction games are a special case of the *stateful performative prediction* framework [5, 50], which models stateful feedback loops. The performative effects of a predictor $h \in \mathcal{H}$ are formally modeled using a transition map $D_{t+1} = \mathcal{T}(D_t, h)$, where $D_t, D_{t+1} \in \Delta(\mathcal{X} \times \mathcal{Y})$ are the data distributions at times $t, t + 1$, and $h \in \mathcal{H}$ is the predictor being deployed. The fixed points of the dynamics induced by evolutionary prediction games (i.e. equilibrium states $\boldsymbol{p}^*$), correspond to *fixed point distributions* as defined in [5, Definition 4], as they satisfy $D_{\boldsymbol{p}^*} = \mathcal{T}(D_{\boldsymbol{p}^*}, h_{\boldsymbol{p}^*})$, where $h_{\boldsymbol{p}^*}$ is the classifier trained on $D_{\boldsymbol{p}^*}$. Moreover, we note that the notion of evolutionary stability is stronger than the notion of fixed point distributions, as evolutionarily stable points also remain stable under small perturbations around $\boldsymbol{p}^*$.

**Structural assumptions reduce effective dimensionality.** Compared to the general performative prediction framework, natural selection dynamics induce transition maps which are lower-dimensional, and thus more amenable for analysis. The general space of feature-label distributions is infinite-dimensional under the common assumption $\mathcal{X} \subseteq \mathbb{R}^d$, and the space of classifiers $\mathcal{H}$ is high-dimensional when the hypotheses class $\mathcal{H}$ is complex (e.g., deep neural networks). Thus, the general performative prediction framework allows for transition operators which are infinite-dimensional and notoriously challenging to analyze (e.g., as noted by Conger et al. [12]). In contrast, in dynamics induced by evolutionary prediction games, the stateful transition operator $\mathcal{T}$ operates within a low-dimensional subspace of all possible feature label-distributions (mixtures of fixed distributions), and depends on predictors through low-dimensional statistics (their marginal accuracy on each group).

Formally, given a collection of group distributions $D_1, \ldots, D_K \in \Delta(\mathcal{X} \times \mathcal{Y})$, denote the space of mixtures by $\Delta_{\mathrm{mix}} = \left\{ \sum_k p_k D_k \mid \boldsymbol{p} \in \Delta^K \right\}$, and note that $\Delta_{\mathrm{mix}} \subseteq \Delta(\mathcal{X} \times \mathcal{Y})$. Any mixture $D \in \Delta_{\mathrm{mix}}$ can be represented by a point on the $(K-1)$-dimensional simplex $\boldsymbol{p} \in \Delta^K$. Additionally, as performative response depends on predictor accuracy, the transition operator depends on the classifier $h$ through a $K$-dimensional vector of accuracies $\mathbf{acc}(h) = (\mathrm{acc}_1(h), \ldots, \mathrm{acc}_K(h)) \in [0,1]^K$. Therefore, transition operators induced by the evolutionary model can be viewed as lower-dimensional mappings $\boldsymbol{p}' = \mathcal{T}(\boldsymbol{p}, \mathbf{acc}(h))$ from a $(2K-1)$-dimensional space to a $(K-1)$-dimensional space.

**Further relations to performative risk and forecasting.** The gaps between the average accuracies of equilibrium points relates to the distinction between performative optimality and performative stability [e.g., 43]. Additionally, convex functions over the simplex (and their sub-gradients) emerge as a key component in our analysis of oracle classifiers (Section 4), and also appears as a key component in the analysis of proper scoring rules and their performative counterparts [45, 8] through the fundamental characterization theorem [21, 53]. From this perspective, interpreting the convex potential of oracle-learner games as a generator of proper scoring rules is an intriguing direction for future work.

### B.2 Long-Term Fairness

Group fairness is often formalized through measures of group disparity, such as the difference in error rates across groups (see, e.g., [18, 2]). However, these measures may fail to capture fairness in dynamic settings. The long-term fairness literature therefore studies the long-term implications of population dynamics on algorithmic fairness; one straightforward example is that if a classifier is trained to comply with fairness constraints, dynamics can cause this guarantee to erode over time [40] (See [22] for a recent survey). Our work proposes a complementing (and somewhat nuanced) counterfactual perspective, suggesting that a system can appear to comply with a fairness definition, but only because some groups were already driven out of the system, and disparity is measured only over the groups that remain.

Several works in the long-term fairness literature also consider replicator (and replicator-like) dynamics of different kinds and variations as models for user response behavior [e.g., 25, 49, 63, 15], albeit not in an evolutionary game setting. Perhaps closest to our approach is Hashimoto et al. [25], who study worst-group accuracy dynamics under repeated training in a mixture population that evolves via stochastic replicator-like dynamics supplemented by a strong, balanced influx of new users that join the system regardless of its performance (parameterized by $b_k$ in their model). While their stability guarantees rely on the assumption of this strong user flow, our results offer a complementary view, showing that stable coexistence may emerge even without an exogenous user flow, under standard empirical risk minimization, and without compromising worst-group accuracy (see Section 5). We also extend their ERM instability results to a wider family of behavioral dynamics, and establish global convergence guarantees, rather than local (see Section 4, and Appendix C).

### B.3 Dynamics-Aware Learners

Finally, our framework models the learning algorithm as an entity which is only informed by the current state of population $h \sim \mathcal{A}(\boldsymbol{p})$, and in particular does not take into account the implications of classifier deployment. While still common in many practical scenarios (see, e.g., [16, 33, 34]), we also note that various practical systems already optimize for long-term objectives. Examples includes learners that reduce hidden incentives to modify the population of users [37, Sec. 7.3], adaption to shifting user preferences [31, 6], or directly maximizing long-term engagement via reinforcement learning (e.g., [1]). While our focus here is on local learning rules, analyzing dynamics-aware learners from an evolutionary perspective is an intriguing direction for future inquiry.

## C  Microfoundations

Natural selection dynamics appear in a variety of settings. This appendix offers a representative (though not exhaustive) set of examples showing how local, myopic user behaviors aggregate into population-level dynamics that satisfy our core assumptions. Using online social platforms as a running example, we show how different behavioral patterns (e.g., invitation, imitation, resource allocation) and different types group affiliations (e.g., social, behavioral, demographic) may give rise to natural-selection dynamics in various parts of the ecosystem. We note that similar microfoundations can be identified across a wide range of domains, such as finance, healthcare, and education.

### C.1  Reproduction

Consider a recommendation system based on prediction. Accurate recommendations motivate users to invite peers from their social group, while the presence of alternatives creates a risk of dropout.

Formally, assume that users from $K$ distinct groups interact with a prediction system over time, and denote by $N_k$ the number of users in group $k \in [K]$. On average, users interact with the system $\lambda > 0$ times per time step $t = 1, 2, \ldots$ (e.g., three times per day), and at each interaction they receive a prediction (e.g., content relevance). If a prediction is accurate, users invite a peer from their group with probability $\alpha$. Independently, a user may leave the system with probability $\beta$ (e.g., due to friction or better alternatives). In expectation, the size of each group is multiplied by the following factor:

$$N_k^{t+1} = \left(1 + \lambda \left(\alpha \cdot \mathrm{acc}_k(h^t) - \beta\right)\right) N_k^t$$

In the continuous-time limit, users interact with the system $\lambda dt$ times per step, and $dt \to 0$. The expected growth approaches the differential equation:

$$\dot{N}_k = \lambda(\alpha \cdot \mathrm{acc}_k(h) - \beta)N_k$$

Where $\dot{N}_k = \frac{\mathrm{d}}{\mathrm{d}t}N_k$ is the time derivative of $N_k$. Denote $F_k = \lambda(\alpha \cdot \mathrm{acc}_k(h) - \beta)$, such that $\dot{N}_k = F_k N_k$. Note that the growth multiplier $F_k$ is time-dependent when the classifier $h$ is retrained.

For group-level dynamics, denote by $N = \sum_k N_k$ the total number of users, and by $p_k = \frac{N_k}{N}$ the proportion of group $k$. The time derivative of $p_k$ is:

$$\begin{aligned}
\dot{p}_k &= \frac{\mathrm{d}}{\mathrm{d}t}\frac{N_k}{N} \\
&= \frac{\dot{N}_k N - \dot{N} N_k}{N^2} \\
&= \underbrace{\frac{N_k}{N}}_{=p_k}\left(\underbrace{\frac{\dot{N}_k}{N_k}}_{\dot{N}_k=F_k N_k} - \underbrace{\frac{\dot{N}}{N}}_{\dot{N}=\sum F_k N_k}\right) \\
&= p_k\left(F_k - \sum_{k'\in[K]} p_{k'}F_{k'}\right)
\end{aligned}$$

Thus, the aggregate dynamics of group proportions $\boldsymbol{p}$ satisfy the *replicator equation* [57], which satisfies our core assumptions. The fitness function in this case is $F_k(\boldsymbol{p}) = \lambda\alpha \cdot \mathrm{acc}_k(h_{\boldsymbol{p}}) - \lambda\beta$.

## C.2 Imitation

Consider a media platform fostering content creators, whose profit depends on the extent to which their content reaches their target audience through accurate predictions. A successful approach (e.g., content style, advertising strategy) of one creator is likely to be mimicked by others.

Formally, suppose there a large population of content producers aiming to market some good. Each producer has a content production strategy $k$ (e.g., short versus long videos), the price of the good is $r$, and the cost of production under strategy $k$ is $c_k$ (e.g., longer videos are typically more costly to produce). A prediction algorithm observes content and makes recommendation decisions to content consumers. A prediction is considered accurate if the content is relevant to the consumer, and each successful prediction yields a conversion (e.g., sale) with probability $\alpha$. Denote by $x \sim D_k$ a content item produced by strategy $k$, and its corresponding label by $y$. The expected profit of a producer using strategy $k$ is:

$$F_k = r \cdot \alpha \mathbb{P}_{D_k}[h(x) = y] - c_k$$

At each step, a content creator observes another creator's content (e.g., on social media). If their own strategy is less successful than the one they observe, they adopt the other's strategy with probability proportional to the difference in utilities. Denote the total size of the population by $N$ (here assumed to be large but fixed), the number of creators who adopt strategy $k$ by $N_k$, and the proportion of group $k$ by $p_k = N_k/N$. Sandholm [52, Chapter 4] shows that this form of *pairwise proportional imitation* converges in the large-population limit to the replicator dynamics [57], which satisfy our core assumptions.

## C.3 Online resource allocation

Natural-selection dynamics between groups of users can also arise from interaction between prediction algorithms in multi-stage pipelines. Consider a firm that acquires users though $K$ marketing campaigns, each targeting a different demographic. The firm uses a classifier to personalize the experience of incoming users, and each accurate prediction yields a conversion with probability $\alpha$. Campaign $k$ therefore generates expected utility:

$$F_k = r \cdot \alpha \cdot \mathrm{acc}_k(h) - c_k$$

where $r$ is the revenue per conversion, and $c_k$ is the average cost of user acquisition for campaign $k$. The firm has a fixed advertising budget $B$, which it dynamically allocates across marketing campaigns.

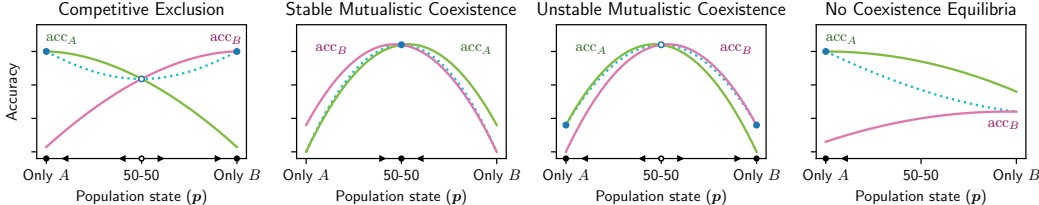

Figure 6: Graphical representation of two-group evolutionary prediction games (Definition 1). Groups are denoted by $\{A, B\}$. Solid lines describe expected group accuracies for each population state $\boldsymbol{p}$, dotted line describes population accuracy $\mathrm{acc}_{\boldsymbol{p}}(h)$, dots indicate Nash equilibria, and x-axis arrows indicate evolutionary stability. Schematic plots illustrate a selection of possible scenarios: **(Left)** Competitive exclusion. There are two stable equilibria, each dominated by a single group, with locally maximal population accuracy. Additionally, there is an unstable coexistence equilibrium with lower overall accuracy. **(Center Left)** Stable mutualistic coexistence. There is a stable coexistence equilibrium, where the accuracy of both groups is maximized. **(Center Right)** Unstable mutualisitic coexistence. Coexistence has maximal population accuracy, but it is unstable. **(Right)** No coexistence equilibria. There is a single equilibrium dominated by a single group.

Denote the spend on campaign $k$ by $p_k B$, and assuming the firm reassigns spend according to a *multiplicative weights* update rule [19] for dynamic allocation, we obtain:

$$p_k^{t+1} = p_k \frac{\beta^{F_k}}{\sum_{k'} \beta^{F_{k'}}}$$

where $\beta > 1$ is a hyper-parameter of the MW algorithm. As $\beta \to 1$, dynamics converge towards the replicator dynamics (cf. Equation (17)), which satisfies our core assumptions.

# D Deferred Proofs

## D.1 Preliminaries

**Distributions.** The set of distribution over a set $A$ is denoted by $\Delta(A)$, and the set of distributions over $[K]$ is denoted by $\Delta^K = \Delta([K]) = \{\boldsymbol{p} \mid \sum_k p_k = 1, p_k \geq 0 \ \forall k \in [K]\}$. The support of a distribution $\boldsymbol{p} \in \Delta^K$ is denoted by $\mathrm{support}(\boldsymbol{p}) = \{k \in [K] \mid p_k > 0\}$. We denote the space tangent to the $K$-simplex by $T\Delta^K = \{z \in \mathbb{R}^K \mid \sum_k z_k = 0\}$. We denote $\mathbf{1} = (1, \ldots, 1) \in \mathbb{R}^K$, and denote by $\boldsymbol{\Phi} = \boldsymbol{I} - \mathbf{1}\mathbf{1}^T$ the orthogonal projection matrix from $\mathbb{R}^K$ to $T\Delta^K$. For $k \in [K]$, we denote by $\mathbf{e}_k \in \Delta^K$ the singleton categorical distribution supported over $k$, representing a mixture which only contains the distribution $D_k$.

**Supervised learning.** We denote the space of features by $\mathcal{X}$, the space of labels by $\mathcal{Y}$, the hypotheses class by $\mathcal{H} \subseteq \mathcal{Y}^{\mathcal{X}}$, and the loss function by $\mathcal{L} : \mathcal{Y}^2 \to \mathbb{R}$. The expected loss of hypothesis $h \in \mathcal{H}$ under population distribution $D \in \Delta(\mathcal{X} \times \mathcal{Y})$ is denoted by $\mathcal{L}_D(h) = \mathbb{E}_{(x,y) \sim D}[\mathcal{L}(h(x), y)]$. A supervised learning algorithm $\mathcal{A}$ is a possibly-stochastic mapping from a distribution $D \in \Delta(\mathcal{X} \times \mathcal{Y})$ to a classifier $h \sim \mathcal{A}(D)$, formally $\mathcal{A} : \Delta(\mathcal{X} \times \mathcal{Y}) \to \Delta(\mathcal{H})$.

**Population state.** Each group $k$ is associated with a feature-label distribution $D_k \in \Delta(\mathcal{X} \times \mathcal{Y})$. When the population state is $\boldsymbol{p} \in \Delta^K$, the training state is a mixture distribution denoted by $D_{\boldsymbol{p}} = \sum_{k \in [K]} p_k D_k$. We use a vector subscript notation to distinguish between group distributions (e.g., $D_k$) and mixture distributions (e.g., $D_{\boldsymbol{p}}$). Given hypothesis $h \in \mathcal{H}$, the expected loss of group $k$ is denoted by $\mathcal{L}_k(h) = \mathbb{E}_{(x,y) \sim D_k}[\mathcal{L}(h(x), y)]$, and the vector of expected group losses is denoted by $\vec{\mathcal{L}}(h) = (\mathcal{L}_1(h), \ldots, \mathcal{L}_K(h))$. For $\boldsymbol{p} \in \Delta^K$, the expected loss of $h$ under distribution $D_{\boldsymbol{p}}$ is denoted by $\mathcal{L}_{\boldsymbol{p}}(h) = \mathbb{E}_{(x,y) \sim D_{\boldsymbol{p}}}[\mathcal{L}(h(x), y)]$.

**Population games.** We follow the notations of Sandholm [52], and include the key definitions here for completeness. Population games associate each group $k \in [K]$ with a fitness function $F_k(\boldsymbol{p})$, which maps a population state $\boldsymbol{p} \in \Delta^K$ to the evolutionary fitness of group $k$. A Nash

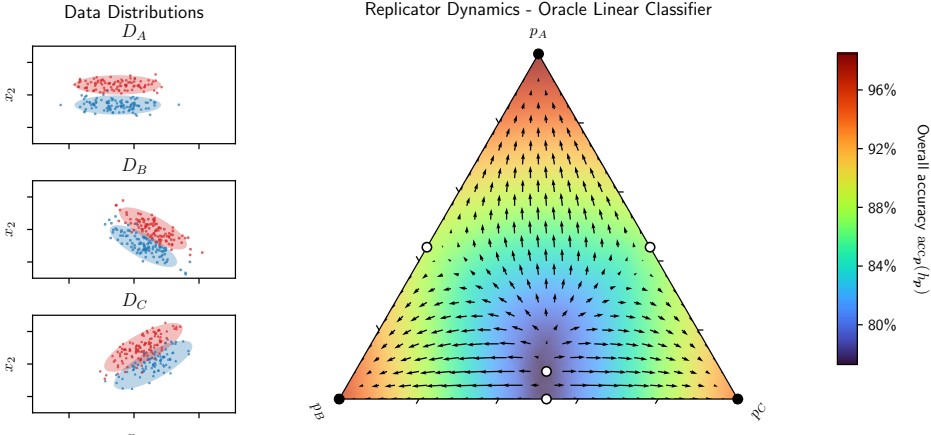

Figure 7: Evolutionary dynamics of an oracle classifier in a three-group setting. **(Left)** Data distributions of groups $\{A, B, C\}$. **(Right)** Phase diagram of replicator dynamics over the probability simplex, induced by the evolutionary prediction game. Background color represents the overall accuracy $\text{acc}_{\boldsymbol{p}}(h_{\boldsymbol{p}})$, vectors represent the flow of replicator dynamics, and markers represent fixed points, with marker color indicating stability. The dynamics have 7 fixed points: One unstable fixed point at the interior of the simplex (unstable coexistence equilibrium), three stable fixed points at the vertices of the simplex (stable equilibria with single-group dominance), and three additional unstable points at the boundaries (unstable restricted equilibria induced by the replicator dynamics).

Equilibrium (NE) of a population game is a population state $\boldsymbol{p}^*$ which satisfies $\text{support}(\boldsymbol{p}^*) \subseteq \arg\max_{k \in [K]} F_k(\boldsymbol{p}^*)$, and every population game admits at least one Nash equilibrium [52, Theorem 2.1.1]. A Restricted Equilibrium (RE) of a population game is a population state $\boldsymbol{p}^*$ which satisfies $\text{support}(\boldsymbol{p}^*) \subseteq \arg\max_{k \in \text{support}(\boldsymbol{p}^*)} F_k(\boldsymbol{p}^*)$, or equivalently $F_k(\boldsymbol{p}^*) = F_{k'}(\boldsymbol{p}^*)$ for all $k, k' \in \text{support}(\boldsymbol{p}^*)$. A population game is a potential game if it can be represented as a gradient of some function. Formally, $F : \Delta^K \to \mathbb{R}^K$ is a potential game if there exists $f : \Delta^K \to \mathbb{R}$ such that $F = \nabla f$, where $\nabla f$ is the gradient of $f$ over the simplex (see [52], Section 3.2 for rigorous definitions).

**Competition, mutualism, and evolutionary stability.** Coexistence equilibria are fixed points in which two groups or more persist $|\text{support}(\boldsymbol{p}^*)| \geq 2$. We say that coexistence is *competitive* if each group's accuracy at $\boldsymbol{p}^*$ is lower than its accuracy in isolation (i.e., when it is the sole group in the population). Conversely, we say that interactions are *mutualistic* if every group's accuracy at $\boldsymbol{p}^*$ exceeds the accuracy it obtains in isolation, indicating that coexistence benefits all groups (see Figure 6 for an illustration). Additionally, we say an equilibrium is *evolutionarily stable* if it attracts nearby population states (see [52, Section 8.3] for formal definitions).

### D.2   Evolutionary Dynamics

For our theoretical analysis, we assume that $\boldsymbol{p}$ evolves in time according to an ordinary differential equation of the form $\dot{\boldsymbol{p}} = V_F(\boldsymbol{p})$, where $V_F : \Delta^K \to T\Delta^K$ is a vector field tangent to the simplex, induced by the fitness function $F(\boldsymbol{p})$, assumed to be continuous (see [52, Theorem 4.4.1]). A common example for such dynamical system is the replicator equation $\dot{p}_k = p_k(F_k(\boldsymbol{p}) - \sum_{k'} F_{k'}(\boldsymbol{p}))$, which corresponds to the tangent vector field $V_F(\boldsymbol{p}) = \boldsymbol{p} \odot (F(\boldsymbol{p}) - \sum_{k'} p_{k'} F_{k'}(\boldsymbol{p})\mathbf{1})$, where $\odot$ is the Hadamard product (see Figure 7). The replicator dynamics are a common model of evolutionary dynamics (see Appendix C, and e.g. [58, 35]), and satisfies our core assumptions. Moreover, we also emphasize that our analysis applies to any $V_F$ which satisfies the core assumptions below.

With respect to $V_F$, we make the following assumptions:

- **Continuity:** We assume that $V_F(\boldsymbol{p})$ is continuous with respect to $\boldsymbol{p}$.

- **Positive correlation:** We assume that $V_F(\boldsymbol{p})$ is directionally aligned with the fitness vector $F(\boldsymbol{p})$ whenever a population is not at rest, formally $V_F(\boldsymbol{p}) \cdot F(\boldsymbol{p}) > 0$ whenever $V_F(\boldsymbol{p}) \neq \boldsymbol{0}$. For a detailed discussion, see [52, Section 5.2].

For correspondence to game equilibria, we assume either of the following assumptions:

- **Nash stationarity:** Under Nash stationarity, all fixed points $V_F(\boldsymbol{p}^*) = \boldsymbol{0}$ correspond to Nash equilibria $\boldsymbol{p}^*$ of the game $F$. See eq. (3) for the definition of Nash equilibrium, and [52, Section 5] for a detailed discussion.
- **Imitative dynamics:** Under imitative dynamics, agents adopt strategies based on observing and copying the behaviors of others in a population. See [52, Section 5.4] for a formal definition using the revision protocol formalism.

The replicator equation introduced in Section 3 is a special case of imitative dynamics, which satisfy our assumptions. In contrast to dynamics with Nash stationarity, imitative dynamics admit additional rest points at restricted equilibria of the population game (formally, states satisfying $F_k(\boldsymbol{p}^*) = F_{k'}(\boldsymbol{p}^*)$ for all $k, k' \in \text{support}(\boldsymbol{p}^*)$). However, we note that this does not affect our main results (Theorem 1 and its extensions, and our constructions in Section 5), as any Nash equilibrium is also a restricted equilibrium, and any non-Nash restricted equilibrium corresponds to an unstable fixed point of the imitative dynamics (see [52, Section 8.1]). See Figure 7 for an illustration of Nash and restricted equilibria in a three-group setting.

**Time scales.** Rates of convergence can be tuned by scaling the vector field $V_F$. For example, replacing $V_F$ with $\alpha V_F$ for $\alpha > 0$ compresses or stretches time by a factor of $\alpha$ while still maintaining our core assumptions. In particular, if $\boldsymbol{p}^t$ satisfies $\dot{\boldsymbol{p}} = V_F(\boldsymbol{p})$, then $\boldsymbol{q}^t = \boldsymbol{p}^{\alpha t}$ is a solution to $\dot{\boldsymbol{q}} = \alpha V_F(\boldsymbol{q})$ with the same initial conditions. Thus, our convergence and stability results hold irrespective of whether the dynamics evolve quickly or slowly.

**Dynamics induced by potential games.** Dynamics induced by potential games are closely related to the structure of the potential function $f(\boldsymbol{p})$. We informally state key results that are relevant to our setting: By [52, Theorem 3.1.3], a state $\boldsymbol{p}^*$ is a Nash equilibrium of $F$ if and only if it is an extremum of the potential function $f$. By [52, Lemma 7.1.1] potential function are Lyapunov functions, and by [52, Theorem 7.1.2], dynamics satisfying the positive correlation property converge towards Nash equilibria (restricted Nash equilibria in case of replicator dynamics). Finally, by [52, Theorem 8.2.1], a population state $\boldsymbol{p}^*$ is a stable equilbrium if and only if it is a local maximizer of $f(\boldsymbol{p})$.

## D.3 Basic Claims

### D.3.1 Linearity

**Proposition D.1** (Expected loss is linear in $\boldsymbol{p}$ for a fixed hypothesis). *For any $h \in \mathcal{H}$ and $\boldsymbol{p} \in \Delta^K$, The expected loss can be represented as a convex combination:*

$$\mathcal{L}_{\boldsymbol{p}}(h) = \sum_{k \in [K]} p_k \cdot \mathcal{L}_k(h)$$

*Proof.* Given a population state $\boldsymbol{p}$, treat it as a categorical random variable over population groups, denote by $k \sim \boldsymbol{p}$ the group from which a feature-label pair gets selected. Applying the law of total expectation and using the definition of $\mathcal{L}_{\boldsymbol{p}}(h)$, we obtain:

$$\begin{aligned}
\mathcal{L}_{\boldsymbol{p}}(h) &= \mathbb{E}_{(x,y)\sim D_{\boldsymbol{p}}}[\mathcal{L}(h(x), y)] \\
&= \mathbb{E}_{k\sim\boldsymbol{p}}\big[\mathbb{E}_{(x,y)\sim D_k}[\mathcal{L}(h(x), y)]\big] \\
&= \sum_{k \in [K]} p_k \cdot \mathbb{E}_{(x,y)\sim D_k}[\mathcal{L}(h(x), y)] \\
&= \sum_{k \in [K]} p_k \cdot \mathcal{L}_k(h)
\end{aligned}$$

$\square$

### D.3.2 Overall Accuracy Equality

Equilibria of evolutionary prediction games also satisfy a natural fairness criterion. A classifier $h$ satisfies *overall accuracy equality* if all groups have equal prediction accuracy [61]. In the context of natural selection, we consider this with respect to the groups currently present in the population:

**Definition D.1** (Overall accuracy equality; [e.g., 61]). *Let $h : \mathcal{X} \to \mathcal{Y}$, and let $\boldsymbol{p} \in \Delta^K$. $h$ satisfies overall prediction equality if $\mathrm{acc}_k(h) = \mathrm{acc}_{k'}(h)$ for all $k, k' \in \mathrm{support}(\boldsymbol{p})$.*

Under retraining, a different classifier $h \sim \mathcal{A}(\boldsymbol{p})$ is deployed at each time step. It is therefore natural to define this property with respect to the learning algorithm, rather than a single classifier. In the following definition, we take the expectation over the stochasticity of the sampling and learning process to representing the average over time under retraining:

**Definition D.2** (Overall accuracy equality in expectation). *Let $\boldsymbol{p}$ be a population state, and let $\mathcal{A}(\boldsymbol{p})$ be a learning algorithm. $\mathcal{A}$ satisfies overall prediction equality in expectation if $\mathbb{E}_{h \sim \mathcal{A}(\boldsymbol{p})}[\mathrm{acc}_k(h)] = \mathbb{E}_{h \sim \mathcal{A}(\boldsymbol{p})}[\mathrm{acc}_{k'}(h)]$ for all $k, k' \in \mathrm{support}(\boldsymbol{p})$.*

*Proof of Proposition 1.* Let $\boldsymbol{p}^*$ be a Nash equilibrium of the evolutionary prediction game induced by $\mathcal{A}(\boldsymbol{p})$. By eq. (3), it holds that $\mathrm{support}(\boldsymbol{p}^*) \subseteq \arg\max_k F_k(\boldsymbol{p}^*)$, and therefore there exists some $a \in \mathbb{R}$ such that $F_k(\boldsymbol{p}^*) = a$ for all $k \in \mathrm{support}(\boldsymbol{p}^*)$. Then by the definition of the evolutionary prediction game (Def. 1), it holds that: $\mathbb{E}_{h \sim \mathcal{A}(\boldsymbol{p})}[\mathrm{acc}_k(h)] = \mathbb{E}_{h \sim \mathcal{A}(D)}[\mathrm{acc}_{k'}(h)] = a$ for all $k, k' \in \mathrm{support}(\boldsymbol{p}^*)$. $\square$

Furthermore, we note that similar reasoning can be utilized to prove an analogous correspondence between overall accuracy fairness and restricted equilibria of the game.

### D.4 Training Once

We begin with a characterization of evolutionary dynamics for a simple setting in which the classifier is trained only once on the initial distribution, and henceforth kept fixed. This is useful as a first step since by fixing $h = h^0$, outcomes can be attributed solely to the induced changes in the population.

We will prove the following theorem:

**Theorem D.1.** *Let $\mathcal{A}$ be a learning algorithm. Denote the initial population state by $\boldsymbol{p}^0$, and the initial classifier by $h^0 \sim \mathcal{A}(\boldsymbol{p}^0)$. If $h^0$ is deployed at every time step (i.e., irrespective of the population state $\boldsymbol{p}$), then it holds that:*

1. ***Competitive exclusion:*** $\boldsymbol{p}^*$ *is supported on $k^* \in \arg\max_k \mathrm{acc}_k(h^0)$, i.e., initially-fittest groups.*

2. ***Stability:*** $\boldsymbol{p}^*$ *is stable.*

3. ***Accuracy:*** $\frac{\mathrm{d}}{\mathrm{d}t}\mathrm{acc}_{\boldsymbol{p}^t}(h^0) \geq 0$, *i.e., overall accuracy weakly improves over time.*

**Definition D.3** (Train-once prediction game). *Let $\boldsymbol{p}^0 \in \Delta^K$ be the population state at time 0, and denote by $h^0 \in \mathcal{H}$ the hypothesis learned at time 0. The train-once prediction game is a population game with the following fitness function:*

$$F_k(\boldsymbol{p}; h^0) = -\mathcal{L}_k(h^0)$$

**Definition D.4** (Constant game; [52, Section 3.2.4]). *A population game $F : \Delta^K \to \mathbb{R}^K$ is a constant game if there exist constants $\alpha_1, \ldots, \alpha_K \in \mathbb{R}$ such that $F_k(\boldsymbol{p}) = \alpha_k$ for all $\boldsymbol{p} \in \Delta^K$.*

**Proposition D.2.** *let $\mathcal{L}$ be a loss function, let $\boldsymbol{p}^0 \in \Delta^K$ be the population state at time 0, and let $h^0 \in \mathcal{H}$ be the hypothesis learned at time 0. Denote by $F_k(\boldsymbol{p})$ the corresponding train-once prediction game, and denote by $\boldsymbol{p}^*$ a Nash equilibrium of $F$. It holds that:*

1. $\mathrm{support}(\boldsymbol{p}^*) \subseteq \arg\min_{k \in [K]} \mathcal{L}_k(h^0)$

2. $\mathcal{L}_{\boldsymbol{p}^*}(h) \leq \mathcal{L}_{\boldsymbol{p}^0}(h)$

3. $\boldsymbol{p}^*$ *is stable.*

*Proof.* Fix $h_0 \in \mathcal{Y}^{\mathcal{X}}$. In the train-once setting, the expected marginal accuracies are constant as a function of $\boldsymbol{p}$, and therefore $F_k$ is a constant game by definition D.4. By [52, Proposition 3.2.13], $F_k$ a potential game with linear potential $f(\boldsymbol{p}) = -\vec{\mathcal{L}}(h^0) \cdot \boldsymbol{p}$. By [52, Theorem 3.1.3], the set of Nash equilibria is the set of local maximizers of $f(\boldsymbol{p})$, which is a convex combination over the set $\arg\min_{k \in [K]} \mathcal{L}_k(h^0)$ due to linearity. Hence:

$$\mathrm{support}(\boldsymbol{p}^*) \subseteq \arg\min_{k \in [K]} \mathcal{L}_k(h^0)$$

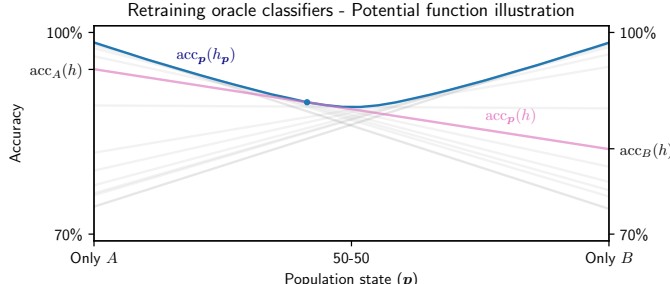

Figure 8: Illustration of the proof of Theorem 1. Pink line: The expected accuracy of a fixed classifier $h$ is linear in $\boldsymbol{p}$ (Proposition D.1). Blue line: The overall accuracy of the oracle classifier $h_{\boldsymbol{p}}$ is a pointwise maximum over linear functions, and is therefore convex (Proposition D.3). The gradient of $\mathrm{acc}_{\boldsymbol{p}}(h_{\boldsymbol{p}})$ over the simplex is the vector of marginal accuracies, showing that the overall accuracy is a potential function for the game (Proposition D.4). Properties of the game's equilibria then follow from convexity arguments.

satisfying (1).

At equilibrium $\boldsymbol{p}^*$, it holds that $\mathcal{L}_{\boldsymbol{p}^*} = \min_{k \in [K]} \mathcal{L}_k(h^0)$. Apply proposition D.1 to obtain:

$$
\begin{aligned}
\mathcal{L}_{\boldsymbol{p}^0}(h^0) &= \sum_k p_k^0 \mathcal{L}_k(h^0) \\
&\geq \sum_k p_k^0 \min_{k' \in [K]} \mathcal{L}_{k'}(h^0) \\
&= \min_{k' \in [K]} \mathcal{L}_{k'}(h^0) \\
&= \mathcal{L}_{\boldsymbol{p}^*}(h^0)
\end{aligned}
$$

satisfying (2).

Finally, observe that each $\boldsymbol{p}^*$ is stable as a maximizer of the linear potential function [52, Theorem 8.2.1]. □

*Proof of Theorem D.1.* Theorem D.1 is a special case of Proposition D.2 for the loss function: $\mathcal{L}_k(h) = -\mathrm{acc}_k(h)$. □

## D.5 Retraining

**Definition D.5** (Retraining prediction game with an oracle predictor)**.** *Let $\boldsymbol{p} \in \Delta^K$, let $\mathcal{L}$ be a loss function, and let $\mathcal{H}$ be a hypothesis class. Denote $h_{\boldsymbol{p}} \in \arg\min_{h \in \mathcal{H}} \mathcal{L}_{\boldsymbol{p}}(h)$, and assume ties are broken in a consistent way. The retraining prediction game with an optimal predictor is a population game with the following payoff function:*

$$
F_k(\boldsymbol{p}) = -\mathcal{L}_k(h_{\boldsymbol{p}})
$$

**Definition D.6** (Optimal loss function $l^*(\boldsymbol{p})$)**.** *Given hypotheses $\mathcal{H}$ and population state $\boldsymbol{p} \in \Delta^K$, the optimal loss at $\boldsymbol{p}$ is:*

$$
l^*(\boldsymbol{p}) = \min_{h \in \mathcal{H}} \mathcal{L}_{\boldsymbol{p}}(h)
$$

**Proposition D.3** ($l^*(\boldsymbol{p})$ is concave)**.** *The optimal loss $l^*(\boldsymbol{p})$ is a concave function of $\boldsymbol{p}$.*

*Proof.* By definition of $l^*(\boldsymbol{p})$:

$$
l^*(\boldsymbol{p}) = \min_{h \in \mathcal{H}} \mathcal{L}_{\boldsymbol{p}}(h)
$$

From proposition D.1, each $\mathcal{L}_{\boldsymbol{p}}(h)$ is linear in $\boldsymbol{p}$ for all $h \in \mathcal{H}$, and in particular concave. Hence, $l^*(\boldsymbol{p})$ is a point-wise minimum of concave functions, and is therefore concave. □

**Definition D.7** (Subgradient; [e.g., 51, Section 23]). *Let $f : \Delta^K \to \mathbb{R}$ be a convex function over the simplex. The vector $\boldsymbol{v} \in T\Delta^K$ is a subgradient of $f$ at point $\boldsymbol{p}$ if for all $\boldsymbol{p}'$ it holds that:*

$$f(\boldsymbol{p}) + \boldsymbol{v} \cdot (\boldsymbol{p}' - \boldsymbol{p}) \leq f(\boldsymbol{p}')$$

*Where $f(\boldsymbol{p}) + \boldsymbol{v} \cdot (\boldsymbol{p}' - \boldsymbol{p})$ is the tangent plane at point $\boldsymbol{p}$.*

**Definition D.8** (Subdifferential). *Let $f : \Delta^K \to \mathbb{R}$ be a convex function over the simplex. The subdifferential of $f$ at $\boldsymbol{p}$ is set of all subgradients at point $\boldsymbol{p}$, denoted by $\partial f(\boldsymbol{p})$.*

**Proposition D.4.** *Let $h_{\boldsymbol{p}} \in \arg\min_{h \in \mathcal{H}} \mathcal{L}_{\boldsymbol{p}}(h)$ be an optimal predictor. The projection of the loss vector of $h$ at point $\boldsymbol{p}$ is a subgradient of $-l^*(\boldsymbol{p})$:*

$$\boldsymbol{\Phi}\vec{\mathcal{L}}(h_{\boldsymbol{p}}) \in \partial\left(-l^*(\boldsymbol{p})\right)$$

*Proof.* Let $h_{\boldsymbol{p}} \in \arg\min_{h \in \mathcal{H}} \mathcal{L}_{\boldsymbol{p}}(h)$, and let $\boldsymbol{p}, \boldsymbol{p}' \in \Delta^K$. We need to show that $-l^*(\boldsymbol{p}) + \vec{\mathcal{L}}(h_{\boldsymbol{p}}) \cdot (\boldsymbol{p}' - \boldsymbol{p}) \leq -l^*(\boldsymbol{p}')$.

By proposition D.1, it holds that $\mathcal{L}_{\boldsymbol{p}'}(h_{\boldsymbol{p}}) = \sum_k p'_k \mathcal{L}_k(h_{\boldsymbol{p}})$. By algebraic manipulation we obtain:

$$
\begin{aligned}
\mathcal{L}_{\boldsymbol{p}'}(h_{\boldsymbol{p}}) &= \vec{\mathcal{L}}(h_{\boldsymbol{p}}) \cdot \boldsymbol{p}' \\
&= \underbrace{\overbrace{\vec{\mathcal{L}}(h_{\boldsymbol{p}}) \cdot \boldsymbol{p}}^{=0} - \vec{\mathcal{L}}(h_{\boldsymbol{p}}) \cdot \boldsymbol{p}}_{=l^*(\boldsymbol{p})} + \vec{\mathcal{L}}(h_{\boldsymbol{p}}) \cdot \boldsymbol{p}' \\
&= l^*(\boldsymbol{p}) + \vec{\mathcal{L}}(h_{\boldsymbol{p}}) \cdot (\boldsymbol{p}' - \boldsymbol{p})
\end{aligned}
\tag{5}
$$

As $\boldsymbol{p}' - \boldsymbol{p} \in T\Delta^K$, it is not affected by the orthogonal projection $\boldsymbol{\Phi} : \mathbb{R}^K \to T\Delta^K$, it holds that:

$$\boldsymbol{\Phi}\left(\boldsymbol{p}' - \boldsymbol{p}\right) = \boldsymbol{p}' - \boldsymbol{p} \tag{6}$$

Since $\boldsymbol{\Phi}$ is an orthogonal projection, it holds that $\boldsymbol{\Phi}^T = \boldsymbol{\Phi}$. In addition, for any vector $\boldsymbol{v}$ it holds that:

$$\boldsymbol{v}^T \boldsymbol{\Phi}^T = \left(\boldsymbol{\Phi}\boldsymbol{v}\right)^T \tag{7}$$

Combining equations (5, 6, 7), we obtain:

$$\boldsymbol{\Phi}\mathcal{L}_{\boldsymbol{p}'}(h_{\boldsymbol{p}}) = l^*(\boldsymbol{p}) + \boldsymbol{\Phi}\vec{\mathcal{L}}(h_{\boldsymbol{p}}) \cdot (\boldsymbol{p}' - \boldsymbol{p})$$

By proposition D.3, it holds that $\mathcal{L}_{\boldsymbol{p}'}(h_{\boldsymbol{p}}) \geq l^*(\boldsymbol{p}')$, and therefore:

$$l^*(\boldsymbol{p}) + \boldsymbol{\Phi}\vec{\mathcal{L}}(h_{\boldsymbol{p}}) \cdot (\boldsymbol{p}' - \boldsymbol{p}) \geq l^*(\boldsymbol{p}')$$

$l^*(\boldsymbol{p})$ is concave, and therefore its negation is convex. Multiplying both sides by $-1$, we obtain that $-\boldsymbol{\Phi}\vec{\mathcal{L}}(\boldsymbol{p})$ is a subgradient of $-l^*$ at point $\boldsymbol{p}$, as required. $\square$

**Proposition D.5.** *Assume $l^*(\boldsymbol{p})$ is differentiable, and let $h_{\boldsymbol{p}} \in \arg\min_{h \in \mathcal{H}} \mathcal{L}_{\boldsymbol{p}}(h)$. The gradient of $-l^*(\boldsymbol{p})$ satisfies:*

$$\nabla\left(-l^*(\boldsymbol{p})\right) = -\boldsymbol{\Phi}\vec{\mathcal{L}}(h_{\boldsymbol{p}})$$

*Proof.* Any convex differentiable function $f$ satisfies:

$$\partial f(\boldsymbol{p}) = \{\nabla f(\boldsymbol{p})\}$$

By proposition D.4, it holds that $-\boldsymbol{\Phi}\vec{\mathcal{L}}(h_{\boldsymbol{p}}) \in \partial\left(-l^*(p)\right)$, and therefore $\nabla\left(-l^*(\boldsymbol{p})\right) = -\boldsymbol{\Phi}\vec{\mathcal{L}}(h_{\boldsymbol{p}})$. $\square$

**Lemma D.1.** *Let $\mathcal{H} \subseteq \mathcal{Y}^{\mathcal{X}}$ and let $\mathcal{L} : \mathcal{Y}^2 \to \mathbb{R}$. For $\boldsymbol{p} \in \Delta^K$, denote $h_{\boldsymbol{p}} = \arg\min_{h \in \mathcal{H}} \mathcal{L}_{\boldsymbol{p}}(h)$. If $\mathcal{L}_{\boldsymbol{p}}(h_{\boldsymbol{p}})$ is a continuously differentiable function of $\boldsymbol{p}$, then game $F_k(\boldsymbol{p}) = -\mathcal{L}_k(h_{\boldsymbol{p}})$ is a potential game.*

*Proof.* Take the function $f(\boldsymbol{p}) = -l^*(\boldsymbol{p})$. By proposition D.5, the gradient of $f(\boldsymbol{p})$ satisfies:

$$\nabla f(\boldsymbol{p}) = -\boldsymbol{\Phi}\vec{\mathcal{L}}(h_{\boldsymbol{p}}) = F(\boldsymbol{p})$$

And therefore $F$ is a potential game. $\square$

We say that a set of groups is *oracle-equivalent* if they share a common oracle classifier:

**Definition D.9** (Oracle-equivalent groups). *A set of groups $K^* \subseteq K$ is oracle-equivalent if there exists $h \in \bigcap_{k \in K^*} \arg\min_{h \in \mathcal{H}} \mathcal{L}_{D_k}(h)$ which satisfies $\mathcal{L}_{k_1}(h) = \mathcal{L}_{k_2}(h)$ for all $k_1, k_2 \in K'$.*

**Proposition D.6.** *Two groups $k, k' \in [K]$ are oracle-equivalent if any only if the expected optimal loss of their convex combination $l^*((1-\alpha)\mathbf{e}_k + \alpha\mathbf{e}_{k'})$ is constant for all $\alpha \in [0, 1]$.*

*Proof.* If the groups are oracle-equivalent, then by Definition D.9 there exists

$$h^* \in \arg\min_{h \in \mathcal{H}} \mathcal{L}_{D_k}(h) \cap \arg\min_{h \in \mathcal{H}} \mathcal{L}_{D_{k'}}(h)$$

which satisfies $\mathcal{L}_{D_k}(h^*) = \mathcal{L}_{D_{k'}}(h^*) = \text{const}$ for some constant. Since $h^*$ is optimal for each group separately, it is also optimal for any convex combination of the groups $(1-\alpha)D_k + \alpha D_{k'}$, and the optimal loss therefore satisfies:

$$l^*((1-\alpha)\mathbf{e}_k + \alpha\mathbf{e}_{k'}) = (1-\alpha)\mathcal{L}_{D_k}(h^*) + \alpha\mathcal{L}_{D_{k'}}(h^*) = \text{const}$$

Conversely, assume that $k, k'$ are not oracle-equivalent. We consider two cases:

1. If $\arg\min_{h \in \mathcal{H}} \mathcal{L}_{D_k}(h) \cap \arg\min_{h \in \mathcal{H}} \mathcal{L}_{D_{k'}}(h) = \emptyset$, then let $\boldsymbol{p} = 0.5\mathbf{e}_k + 0.5\mathbf{e}_{k'}$. By linearity of expectation, for any $h \in \mathcal{H}$ it holds that:

$$\mathcal{L}_{D_{\boldsymbol{p}}}(h) = 0.5\mathcal{L}_{D_k}(h) + 0.5\mathcal{L}_{D_{k'}}(h)$$

By the assumption, no $h \in \mathcal{H}$ simultaneously minimizes both terms. Therefore, the optimal loss at $\boldsymbol{p}$ must satisfy either $l^*(\boldsymbol{p}) > l^*(\mathbf{e}_k)$ or $l^*(\boldsymbol{p}) > l^*(\mathbf{e}_{k'})$, and therefore $l^*((1-\alpha)\mathbf{e}_k + \alpha\mathbf{e}_{k'})$ is not constant.

2. Otherwise, if there exists a common optimizer $h^* \in \arg\min_{h \in \mathcal{H}} \mathcal{L}_{D_k}(h) \cap \arg\min_{h \in \mathcal{H}} \mathcal{L}_{D_{k'}}(h)$ but $\mathcal{L}_{D_k}(h^*) \neq \mathcal{L}_{D_{k'}}(h^*)$, then $l^*((1-\alpha)\mathbf{e}_k + \alpha\mathbf{e}_{k'})$ is not constant when comparing the values corresponding to $\alpha = 0$ and $\alpha = 1$.

$\square$

**Proposition D.7.** *The local minimizers of the optimal expected loss function $l^*$ are convex combinations of oracle-equivalent groups.*

*Proof.* By Proposition D.3, the optimal loss $l^*$ is concave over the simplex $\Delta^K$, and therefore its minimizers are convex combinations of simplex vertices for which the function is constant. By Proposition D.6, such combinations correspond to sets of oracle-equivalent groups. $\square$

**Proposition D.8.** *When each group has a distinct optimal classifier, the optimal loss function $l^*(\boldsymbol{p})$ only has local minimizers at vertices of the simplex.*

*Proof.* When each group has a distinct optimal classifier, the only sets of oracle-equivalent groups are the singleton sets $\{1\}, \ldots, \{K\}$. The claim then follows from Proposition D.7. $\square$

*Proof of Theorem 1.* By applying Lemma D.1 on the loss function $\mathcal{L}_k(h) = -\text{acc}_k(h)$, we obtain that $F(\boldsymbol{p})$ is a potential game with potential function $f(\boldsymbol{p}) = \text{acc}_{\boldsymbol{p}}(h_{\boldsymbol{p}})$. By Proposition D.3, we obtain that $f(\boldsymbol{p})$ is convex over the simplex. When no two groups share an optimal classifier Proposition D.8 holds, and local maximizers of $f$ exist and are located at the vertices of the simplex. By [52, Theorem 8.2.1], maximizers of $f(\boldsymbol{p})$ correspond to evolutionarily stable Nash equilibria of $F(\boldsymbol{p})$, and therefore any stable equilibrium $\boldsymbol{p}^*$ satisfies $|\text{support}(\boldsymbol{p}^*)| = 1$. Additionally, we note that coexistence equilibria may exist as $f(\boldsymbol{p})$ may have local minimizers. This proves claims 2,3,4 (Stability, Exclusion, Coexistence) in the statement of the theorem. Finally, for claim 1 (Accuracy), by [52, Lemma 7.1.1] the potential function $f(\boldsymbol{p})$ is a Lyapunov function for any evolutionary dynamic which satisfies the positive correlation property, and therefore $\frac{d}{dt}f(\boldsymbol{p}) = \frac{d}{dt}\text{acc}_{\boldsymbol{p}}(h_{\boldsymbol{p}}) \geq 0$. $\square$

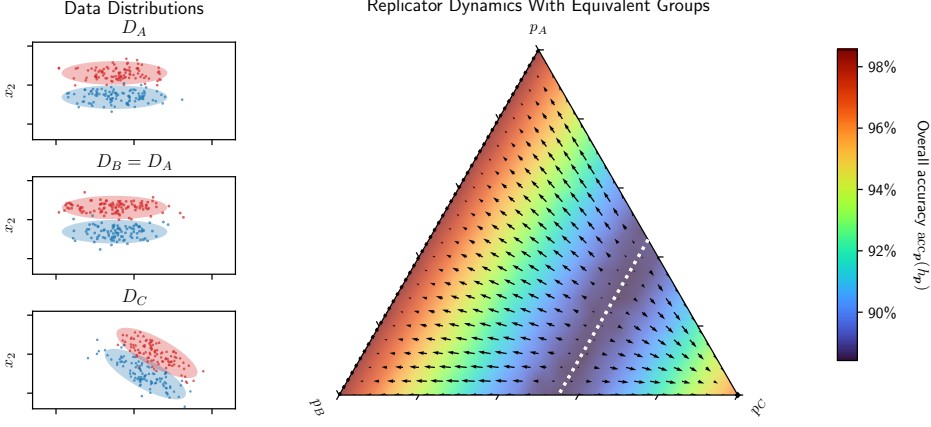

Figure 9: Evolutionary dynamics of an oracle classifier in a three-group setting with equivalent groups (Appendix D.5.1). **(Left)** Data distributions of groups $\{A, B, C\}$, where $D_B = D_A$. Groups $A, B$ are oracle-equivalent, and group $C$ is not oracle-equivalent to $A$ or $B$ (Definition D.9). **(Right)** Phase diagram of the corresponding replicator dynamics over the probability simplex (cf. Figure 7). The dynamics has three *sets* of fixed points, corresponding to Theorem D.2: The convex hull of the singleton distributions corresponding to $A, B$ (stable), the singleton distribution of group $C$ (stable), and the convex hull of $\boldsymbol{p} \approx (0.45, 0, 0.55)$ and $\boldsymbol{p}' \approx (0, 0.45, 0.55)$ (unstable).

### D.5.1 Equivalent groups

Share the same optimal classifier and achieve the same optimal accuracy (formally satisfying Definition D.9), then an extended version of Theorem 1 holds:

**Theorem D.2** (Oracle classifier with equivalent groups; Extension of Theorem 1). *Let $\mathcal{H}$ be a hypothesis class, and denote by $\mathcal{A}^{\mathrm{opt}}$ the oracle learning algorithm with respect to $\mathcal{H}$. Assume that at each population state $\boldsymbol{p}$ the oracle classifier $h_{\boldsymbol{p}} \sim \mathcal{A}^{\mathrm{opt}}(D_{\boldsymbol{p}})$ is deployed. Then it holds that:*

1. ***Accuracy:*** *Overall accuracy increases over time, $\frac{\mathrm{d}}{\mathrm{d}t}\mathrm{acc}_{\boldsymbol{p}}(h_{\boldsymbol{p}}) \geq 0$.*

2. ***Stability:*** *A stable equilibrium always exists, and there can be multiple such equilibria.*

3. ***Competitive exclusion:*** *For all stable equilibria, $\mathrm{support}(\boldsymbol{p}^*)$ is oracle-equivalent (Def. D.9).*

4. ***Coexistence:*** *Equilibria with non-equivalent groups may exist, but are unstable.*

*Proof.* In analogy to the proof of Theorem 1. The potential function remains $f(\boldsymbol{p}) = \mathrm{acc}_p(h_{\boldsymbol{p}})$, and so does the relation between extrema of the potential and equilibria of the dynamics. However, as groups are possibly equivalent, by Proposition D.7 we obtain that $f(\boldsymbol{p})$ may have additional weak maximizers in convex combinations of oracle-equivalent groups. $\square$

### D.5.2 Heterogeneous fitness

Here we extend Theorem 1 to settings where retention functions $\nu_k$ vary across groups – See Theorem D.3 below. For $K = 2$, we show that a modified version of the theorem holds for any pair of continuous, strictly increasing retention functions. For $K > 2$, a similar conclusion follows under positive affine retention functions $\nu_k(x) = ax - b_k$ with $a, b_k \geq 0$. In each case, we prove the extended theorem by constructing a transformed potential function and showing that it retains the properties required for convergence and equilibrium structure guarantees.

**Continuous increasing retention functions** ($K = 2$)**.** Consider a two-group setting, and denote $[K] = \{A, B\}$. Let $\mathcal{A}^{\mathrm{opt}}$ be an oracle classification algorithm. Denote $\boldsymbol{p} = (p, 1 - p)$, and denote $h_p \sim \mathcal{A}^{\mathrm{opt}}((p, 1 - p))$.

**Proposition D.9.** *For any oracle algorithm, it holds that $\mathcal{L}_A(h_p)$ is decreasing in $p$, and $\mathcal{L}_B(h_p)$ is increasing in $p$.*

*Proof.* Assume by contradiction that $\mathcal{L}_A(h_p)$ is not decreasing, and therefore there exist $p' > p$ such that $\mathcal{L}_A(h_p) < \mathcal{L}_A(h_{p'})$. The proof relies on a swapping argument. Denote:

$$a = \mathcal{L}_A(h_p)$$
$$a' = \mathcal{L}_A(h_{p'})$$
$$b = \mathcal{L}_B(h_p)$$
$$b' = \mathcal{L}_B(h_{p'})$$

By Proposition D.1, the average for loss for any fixed classifier is a convex combination of marginal loss functions. $h_p$ is optimal for $p$, and therefore:

$$\underbrace{pa + (1-p)b}_{=\mathcal{L}_{\boldsymbol{p}}(h_p)} \leq \underbrace{pa' + (1-p)b'}_{=\mathcal{L}_{\boldsymbol{p}}(h_{p'})}$$

Similarly, $h_{p'}$ is optimal for $p'$, and therefore:

$$p'a' + (1-p')b' \leq p'a + (1-p')b$$

Adding the two inequalities:

$$(p'-p)(a'-a) \leq (p'-p)(b'-b)$$

Since $p' - p > 0$, we divide by $(p'-p)$ to obtain:

$$a' - a \leq b' - b$$

$a < a'$ by the contradiction assumption, and therefore by it holds that $b' - b > 0$. But then:

$$p'a + (1-p')b < p'a' + (1-p')b'$$

In contradiction to the optimality of $h_{p'}$. Therefore, it must hold that $a \leq a'$. Similarly, assume by contradiction that $b' > b$ and obtain a similar contradiction. Figure 8 provides additional intuition. $\qquad\square$

**Proposition D.10** (Potential function of a two-group population game; [52, Example 3.2.3]). *Any two-group population game is a potential game, and the following is a potential function:*

$$f((p, 1-p)) = \int_0^p (F_A((x, 1-x)) - F_B((x, 1-x)))\,\mathrm{d}x \tag{8}$$

**Proposition D.11.** *Let $\tilde{f}(\boldsymbol{p})$ be a potential function for a two-group evolutionary prediction game induced by an oracle classifier with increasing retention functions. Then $\tilde{f}(\boldsymbol{p})$ doesn't have local maximizers in the interior of the simplex.*

*Proof.* Denote the transformed fitness function by $\tilde{F}_A((p, 1-p)) = \nu_A\,(\mathrm{acc}_A(h_p))$, and similarly denote $\tilde{F}_B((p, 1-p)) = \nu_B\,(\mathrm{acc}_B(h_p))$. Let $g(p) = \tilde{F}_A((p, 1-p)) - \tilde{F}_B((p, 1-p))$, and denote by $\tilde{f}((p, 1-p))$ the potential function for the transformed game, defined by Equation (8). By Proposition D.9, it holds that $\tilde{F}_A((p, 1-p))$ is increasing, $\tilde{F}_B((p, 1-p))$ is decreasing, and therefore $g(p)$ is monotonically increasing as a function of $p$. By Equation (8), the potential $\tilde{f}$ is decreasing if $g(p) \leq 0$ for all $p$, increasing if $g(p) \geq 0$ for all $p$, and unimodal (decreasing and then increasing) if $g(p)$ changes signs. Note that $g(p)$ may change signs at most once due to monotonicity. In all of these cases, $g(p)$ doesn't have a local maximizer at the interior of the simplex. $\qquad\square$

**Positive affine retention functions with group-wise shift** ($K \geq 2$). Let $\nu_k(x) = ax - b_k$ for $a, b_k \geq 0$. We show that the game admits a transformed potential function:

**Proposition D.12.** *Let $\mathcal{A}^{\mathrm{opt}}$ be an oracle classification algorithm, and let $f(\boldsymbol{p})$ be the potential function for the game $F_k(\boldsymbol{p}) = \mathrm{acc}_k(h_{\boldsymbol{p}})$. Then the game $\tilde{F}_k(\boldsymbol{p}) = \nu_k\,(\mathrm{acc}_k(h_{\boldsymbol{p}})) = a\,\mathrm{acc}_k(h_{\boldsymbol{p}}) - b_k$ admits the potential function:*

$$\tilde{f}(\boldsymbol{p}) = af(\boldsymbol{p}) - \boldsymbol{b} \cdot \boldsymbol{p}$$

*where $\boldsymbol{b} = (b_1, \ldots, b_K) \in \mathbb{R}_{\geq 0}^K$.*

*Proof.* First, consider the game $F_k^{(a)} = aF_k(\boldsymbol{p})$. By linearity of the gradient operation, the population game $F^{(a)}(\boldsymbol{p})$ is a potential game with potential $f^{(a)}(\boldsymbol{p}) = af(\boldsymbol{p})$. Then, consider the game $F_k^{(b)}(\boldsymbol{p}) = -b_k$. The game $F^{(b)}(\boldsymbol{p})$ is a constant game, and therefore admits the linear potential function $f^{(b)}(\boldsymbol{p}) = -\boldsymbol{b} \cdot \boldsymbol{p}$ (see [52, Proposition 3.2.13]). Finally, by [52, Section 3.2.4], the game $\tilde{F}(\boldsymbol{p}) = F^{(a)}(\boldsymbol{p}) + F^{(b)}(\boldsymbol{p})$ is a potential game with the potential function:

$$\tilde{f}(\boldsymbol{p}) = f^{(a)}(\boldsymbol{p}) + f^{(b)}(\boldsymbol{p}) = af(\boldsymbol{p}) - \boldsymbol{b} \cdot \boldsymbol{p}$$

as required. □

**Proposition D.13.** *Let $f(\boldsymbol{p})$ be a potential function for an evolutionary prediction game induced by an oracle classifier and positive affine retention functions with group-wise shift. Then $\tilde{f}(\boldsymbol{p})$ doesn't have local maximizers in the interior of the simplex.*

*Proof.* By the proof of Theorem 1, $F(\boldsymbol{p})$ is a potential game with potential function $f(\boldsymbol{p}) = \mathrm{acc}_{\boldsymbol{p}}(h_{\boldsymbol{p}})$. By Proposition D.3 we obtain that $f(\boldsymbol{p})$ is convex over the simplex. Multiplying a convex function by a positive constant and adding a linear function maintains convexity, and therefore $\tilde{f}(\boldsymbol{p})$ is convex as well, and in particular does not have local maximizers in the interior of the simplex. □

**Theorem 1 under heterogeneous retention.** In both cases analyzed above, we identify a modified potential function which does not have local minimizers in the interior of the simplex (Propositions D.11 and D.13). We now proceed to state a modified version of the central theorem:

**Theorem D.3** (Oracle classifier with heterogeneous retention; Extension of Theorem 1)**.** *Let $\mathcal{H}$ be a hypothesis class, and denote by $\mathcal{A}^{\mathrm{opt}}$ the oracle learning algorithm with respect to $\mathcal{H}$. For $K = 2$, assume that retention functions $\nu_k$ are monotone increasing, and for $K > 2$ assume that $\nu_k$ are positive affine with group-wise shift. Assume that at each population state $\boldsymbol{p}$ the oracle classifier $h_{\boldsymbol{p}} \sim \mathcal{A}^{\mathrm{opt}}(D_{\boldsymbol{p}})$ is deployed and the fitness of group $k$ is $\tilde{F}_k(\boldsymbol{p}) = \nu_k(\mathrm{acc}_k(h_{\boldsymbol{p}}))$. Then it holds that:*

1. **Welfare:** *Average population welfare increases over time:* $\frac{\mathrm{d}}{\mathrm{d}t} \sum_k p_k \nu_k(\mathrm{acc}_k(h_{\boldsymbol{p}})) \geq 0$.

2. **Stability:** *A stable equilibrium always exists, and there can be multiple such equilibria.*

3. **Competitive exclusion:** *For all stable equilibria, $|\mathrm{support}(\boldsymbol{p}^*)| = 1$.*

4. **Coexistence:** *Equilibria with $|\mathrm{support}(\boldsymbol{p}^*)| \geq 2$ may exist, but are unstable.*

*Proof.* In analogy to the proof of Theorem 1 (see Appendix D.5). The corresponding evolutionary prediction games are potential games, with Nash equilibria corresponding to local equilibria of the potential function, and stable equilibria corresponding to local maximizers of the potential function. By Propositions D.11 and D.13, potential functions for both families of retention functions don't have local maximizers at the interior of the simplex, implying claims (2,3,4).

For claim (1), we consider two cases, according to the family of retention functions under consideration. For positive affine functions, the transformed potential function is $\tilde{f}(\boldsymbol{p}) = af(\boldsymbol{p}) - \boldsymbol{b} \cdot \boldsymbol{p}$, where $f(\boldsymbol{p})$ is the potential function of the original game (with $\nu_k(x) = x$). Rearranging the summations:

$$\tilde{f}(\boldsymbol{p}) = a \cdot \underbrace{\mathrm{acc}_{\boldsymbol{p}}(h_{\boldsymbol{p}})}_{=\sum_k p_k \mathrm{acc}_k(h_{\boldsymbol{p}})} - \boldsymbol{b} \cdot \boldsymbol{p}$$

$$= \sum_k p_k (a \cdot \mathrm{acc}_k(h_{\boldsymbol{p}}) - b_k)$$

$$= \sum_k p_k \nu_k(\mathrm{acc}_k(h_{\boldsymbol{p}}))$$

and claim (1) follows from the fact that $\tilde{f}(\boldsymbol{p})$ is a Lyapunov function. For monotone increasing functions, assume without loss of generality that $\tilde{F}_A(\boldsymbol{p}) \geq \tilde{F}_B(\boldsymbol{p})$. Then from the positive correlation assumption $\dot{p}_A = -\dot{p}_B \geq 0$. From Proposition D.9 we obtain that $\mathrm{acc}_A(h_{\boldsymbol{p}})$ is increasing and

$\text{acc}_B(h_{\boldsymbol{p}})$ is decreasing as a function of $p_A$. Combining the observations, we obtain:

$$\frac{d}{dt}\sum_k p_k \nu_k \left(\text{acc}_k(h_{\boldsymbol{p}})\right) = \frac{d}{dt}\left(p_A \nu_A\left(\text{acc}_A(h_{\boldsymbol{p}})\right) + p_B \nu_B\left(\text{acc}_B(h_{\boldsymbol{p}})\right)\right)$$

$$= \dot{p}_A \underbrace{\left(\tilde{F}_A(\boldsymbol{p}) - \tilde{F}_B(\boldsymbol{p})\right)}_{\geq\, 0 \text{ without loss of generality}}$$

$$+ p_A \underbrace{\frac{\partial}{\partial p_A}\nu_A(\text{acc}_A(h_{\boldsymbol{p}}))}_{\geq\, 0 \text{ by Proposition D.9}} \underbrace{\dot{p}_A}_{\geq 0}$$

$$+ p_B \underbrace{\frac{\partial}{\partial p_A}\nu_B(\text{acc}_B(h_{\boldsymbol{p}}))}_{\leq\, 0 \text{ by Proposition D.9}} \underbrace{(-\dot{p}_A)}_{\leq 0}$$

And thus $\frac{d}{dt}\sum_k p_k \nu_k\left(\text{acc}_k(h_{\boldsymbol{p}})\right) \geq 0$ as required. $\qquad\square$

### D.6 Soft-SVM

**Definition D.10** (Soft-SVM; [e.g., 54, Section 15.2]). *Let $\{(x_i, y_i)\}_{i=1}^n$ be a training set, and let $\lambda > 0$. The Soft-SVM learning algorithm outputs a linear classifier $h(x) = \text{sign}(w^* \cdot x + b)$ such that $(w^*, b^*)$ are minimizers of the regularized hinge loss:*

$$(w^*, b^*) = \arg\min_{w,b} \lambda \|w\|^2 + \frac{1}{n}\sum_{i=1}^n \max\{0, 1 - y_i(wx_i + b)\}$$

#### D.6.1 Construction

Consider a two-group binary classification setting ($K = 2, \mathcal{Y} = \{-1, 1\}$). Denote the two groups by $[K] = \{A, B\}$. Denote by $\text{Triangular}(a, b, c)$ the triangular distribution with parameters $(a, b, c)$ and probability density function:

$$\mathbb{P}_{x\sim\text{Triangular}(a,b,c)}[x] = \begin{cases} \frac{2(x-a)}{(b-a)(c-a)} & x \in [a, c) \\ \frac{2(b-x)}{(b-a)(b-c)} & x \in [c, b] \\ 0 & \text{otherwise} \end{cases}$$

Let $\alpha \in [0.5, 1)$ be a population balance parameter. Define the data distributions:

$$X_A \sim \alpha \cdot \text{Triangular}(-1, 0, -1) + (1 - \alpha) \cdot \text{Triangular}(0, 1, 0)$$
$$X_B = -X_A = (1 - \alpha) \cdot \text{Triangular}(-1, 0, 0) + \alpha \cdot \text{Triangular}(0, 1, 1)$$
$$Y|X = \begin{cases} 1 & X \geq 0 \\ -1 & X < 0 \end{cases}$$

$D_A$ is the distribution of tuples $(X_A, Y|X_A)$, and $D_B$ is defined correspondingly as $D_B = (X_B, Y|X_B)$. The distributions are illustrated in Figure 3 (Left).

#### D.6.2 Stable Mutualistic Coexistence

We show that the Soft-SVM algorithm can induce a stable, beneficial coexistence. We will consider learning in the population limit $n \to \infty$, where the loss minimization objective (Definition D.10) converges towards its expected value:

**Definition D.11** (Soft-SVM in the population limit). *Denote the probability density function of the data by $f(x, y)$. In the population limit $n \to \infty$, the Soft-SVM classifier $(w^*, b^*)$ is a minimizer of the expected regularized hinge loss:*

$$(w^*, b^*) = \arg\min_{w,b} L(w, b)$$

*Where:*

$$L(w, b) = \lambda \|w\|^2 + \sum_y \int_x f(x, y) \max\{0, 1 - y(wx + b)\}dx \tag{9}$$

**Proposition D.14.** *For a one-dimensional Soft-SVM classification problem, denote and feature-label distribution by $f$, and denote by $L(w, b)$ the regularized hinge loss over the population. It holds that:*

$$\frac{\partial L(w, b)}{\partial b} = \int_{wx+b \geq -1} f(x, y = -1)\mathrm{d}x - \int_{wx+b \leq 1} f(x, y = 1)\mathrm{d}x \tag{10}$$

*Proof.* The population hinge loss of a 1D classifier:

$$L(w, b) = \lambda w^2 + \sum_y \int_{x \in \mathbb{R}} f(x, y) \max\{0, 1 - y(wx + b)\}\mathrm{d}x$$

First, we calculate the derivative $\frac{\partial L}{\partial b}$:

$$\frac{\partial L(w, b)}{\partial b} = \frac{\partial}{\partial b} \sum_y \int_{x \in \mathbb{R}} f(x, y) \max\{0, 1 - y(wx + b)\}\mathrm{d}x$$

By the Leibniz integral rule, and excluding the single non-differential point from the integral:

$$= \sum_y \int_{x \in \mathbb{R} \backslash \{ywx - 1\}} f(x, y) \frac{\partial}{\partial b} \max\{0, 1 - y(wx + b)\}\mathrm{d}x$$

$$= \sum_y \int_{y(wx+b) \leq 1} f(x, y)(-y)\mathrm{d}x$$

Expanding the summation over $y$ yields:

$$= \int_{wx+b \geq -1} f(x, y = -1)\mathrm{d}x - \int_{wx+b \leq 1} f(x, y = 1)\mathrm{d}x$$

As required.

$\square$

*Proof of Theorem 2.* Assume the Soft-SVM algorithm trains on a large dataset sampled from $D_{\boldsymbol{p}}$. Denote $\boldsymbol{p} = (1 - p, p)$, and denote the probability density functions of groups $\{A, B\}$ by $f_A(x, y), f_B(x, y)$, respectively. Since the data is separable and labels are positively correlated with the feature $x$, it holds that $w^* > 0$. Assume that regularization parameter $\gamma$ is large enough such that $w^* \leq 1$. From Proposition D.14, it holds that:

$$\left. \frac{\partial L(w, b)}{\partial b} \right|_{b=0} = \int_{wx \geq -1} f(x, y = -1)\mathrm{d}x - \int_{wx \leq 1} f(x, y = 1)\mathrm{d}x$$

When $w \in (0, 1]$, observe that the distribution defined in Appendix D.6.1 is suppported on $[-1, 1]$, and therefore both integrals cover the whole distribution. We can therefore write:

$$\left. \frac{\partial L(w, b)}{\partial b} \right|_{b=0} = \mathbb{P}_{(x,y) \sim D_{\boldsymbol{p}}}[y = -1] - \mathbb{P}_{(x,y) \sim D_{\boldsymbol{p}}}[y = 1]$$

By construction, for group $A$ it holds that

$$\mathbb{P}_{(x,y) \sim D_A}[y = -1] - \mathbb{P}_{(x,y) \sim D_A}[y = 1] = \alpha - (1 - \alpha) = -(1 - 2\alpha)$$

and similarly for group $B$:

$$\mathbb{P}_{(x,y) \sim D_B}[y = -1] - \mathbb{P}_{(x,y) \sim D_B}[y = 1] = 1 - 2\alpha$$

Therefore, for $D_{\boldsymbol{p}}$, it holds that:

$$\left. \frac{\partial L(w, b)}{\partial b} \right|_{b=0} = p(1 - 2\alpha) - (1 - p)(1 - 2\alpha)$$

The state $\boldsymbol{p} = (0.5, 0.5)$ corresponds to $p = 0.5$. For this state, it holds that $\left. \frac{\partial L(w,b)}{\partial b} \right|_{b=0} = 0$, and therefore $b^* = 0$. Since the data is separable at $x = 0$, any classifier with $w^* > 0$ and $b^* = 0$ achieves perfect accuracy, and therefore the state $\boldsymbol{p} = (0.5, 0.5)$ is beneficial for both groups.

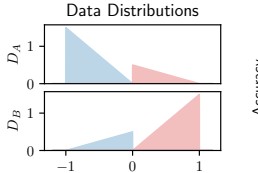 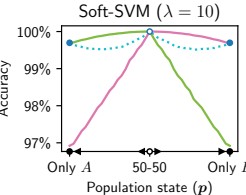 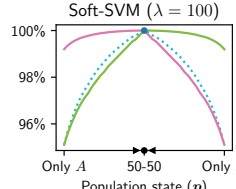 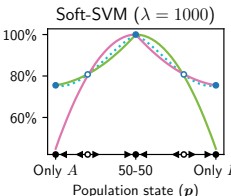

Figure 10: Varying regularization strength for Soft-SVM classifiers. **(Left)** Data distributions. **(Center Left)** Evolutionary prediction game induced by a Soft-SVM classifier with $\lambda = 10$. The game has two stable equilibria at the boundaries, and an unstable coexistence equilibrium. **(Center Right)** Game induced by a Soft-SVM classifier with $\lambda = 100$. The game has one stable coexistence equilibrium. **(Right)** Game induced by a Soft-SVM classifier with $\lambda = 1000$. The game has five equilibria: Two stable single-group equilibria, one stable beneficial coexistence equilibrium at $\boldsymbol{p} = (0.5, 0.5)$, and two unstable coexistence equilibria at $\boldsymbol{p} \approx (0.2, 0.8)$ and $\boldsymbol{p} \approx (0.8, 0.2)$.

To prove stability, we take an additional partial derivative by $p$:

$$\left.\frac{\partial^2 L(w,b)}{\partial p \partial b}\right|_{b=0, p=0.5} = 2(1 - 2\alpha)$$

And as $\alpha > 0.5$, it holds that $\left.\frac{\partial^2 L(w,b)}{\partial p \partial b}\right|_{b=0, p=0.5} < 0$. Let $\varepsilon > 0$ which is sufficiently small. For $p' = 0.5 + \varepsilon$ it holds that $\left.\frac{\partial L(w,b)}{\partial b}\right|_{b=0} < 0$, and since $L(w,b)$ is convex, the optimal $b^*$ corresponding to $p'$ satisfies $b^* > 0$, and the decision boundary is smaller than $0$. Assume that the deviation $\varepsilon$ is sufficiently small such that the new $b^*$ is also in the neighborhood of $0$. By definition, in a sufficiently small neighborhood of zero, it holds by construction for each group:

$$\text{acc}_A(h_{\boldsymbol{p}'}) = \text{acc}_A(w^* > 0, b^* > 0) = 1 - \int_{[-b^*/w^*, 0]} f_A(x, y = -1)\mathrm{d}x$$

$$\text{acc}_B(h_{\boldsymbol{p}'}) = \text{acc}_B(w^* > 0, b^* > 0) = 1 - \int_{[-b^*/w^*, 0]} f_B(x, y = -1)\mathrm{d}x$$

Hence, for a sufficiently small deviation $p' = p + \varepsilon$ it holds by the alignment of the triangle distributions that $\text{acc}_A(h_{\boldsymbol{p}'}) > \text{acc}_B(h_{\boldsymbol{p}'})$. Applying a similar argument to an opposite deviation $p'' = 0.5 - \varepsilon$ shows that an opposite relation holds, and therefore the state $\boldsymbol{p} = (0.5, 0.5)$ is also stable. $\qquad\square$

### D.6.3 Bifurcations Induced by Regularization

In Figure 10, we vary the regularization strength for the distributions specified in Appendix D.6.1 with imbalance parameter $\alpha = 0.75$. We observe that the system transition between three topologically-distinct phases: For weak regularization ($\lambda = 10$), coexistence is beneficial but not stable. For intermediate regularization ($\lambda = 100$), coexistence is beneficial and stable. Interestingly, for strong regularization ($\lambda = 1000$), the game has five equilibria – Two stable single-group equilibria, one stable coexistence equilibia, and two unstable coexistence equilibria.

### D.7 k-NN Classifiers With Label Noise

Another practical approach to loss minimization is to use a class of models that fit the training data perfectly and then interpolate. Notable examples include the nearest-neighbor algorithm, and over-parameterized neural networks [e.g., 64]. Interpolation leads to perfect accuracy on the training set by definition, but often at the price of sensitivity to noise; as a concrete example, consider how 1-NN predictions change with the addition of a single point. The problem is that even in the limit, 1-NN cannot express the true $p(y|x)$, and remains sensitive to label noise. Our next result shows this can lead to coexistence:

**Theorem D.4.** *For the 1-NN learning algorithm, there exist noisy-label data distributions that induce an evolutionary prediction game with stable coexistence.*

Formal proof in Appendix D.7.2. The construction is illustrated in Figure 3 (Center), and described formally in Appendix D.7.1. Informally, there are two groups defined symmetrically. Each group is composed of a majority and minority classes, and the majority class has label noise. For each group alone, the 1-NN algorithm makes perfect predictions on its minority class, and imperfect predictions on the majority class. Since groups are defined symmetrically, training on one group alone leads to better accuracy for the other—leading to stable coexistence. Moreover, Lemma D.2 shows that for $k$-NN with $k \geq 3$, stable coexistence can also be beneficial for both groups.

**Definition D.12** (k-Nearest-Neighbors; e.g. [54, Section 19.1]). *Let $k$ be an odd positive integer. Given a training set $S \in (\mathcal{X} \times \mathcal{Y})^n$ and a feature vector $x$, denote the $k$ nearest neighbors of $x$ in the training set by $N(x)$. The k-NN classifier returns the majority label $y$ among the members of $N(x)$.*

### D.7.1 Construction

Consider a two-group binary classification setting ($K = 2$, $\mathcal{Y} = \{-1, 1\}$). Denote the two groups by $[K] = \{A, B\}$, and let $\alpha > 0$, $\beta \geq 0.5$. The data distributions for the two groups are defined symmetrically:

- Each group is comprised of a majority subgroup and a minority subgroup. The proportion of the majority subgroup is $\beta \geq 0.5$.

- One subgroup is initially contains positively-labeled data, denoted by $D_{\text{pos}} \in \Delta(\mathcal{X} \times \{1\})$. Conversely, the other subgroup is initially comprised of data with negative labels, and denoted by $D_{\text{neg}} \in \Delta(\mathcal{X} \times \{-1\})$. The positive subgroup is the majority of group $A$, and the negative subgroup is majority of group $B$.

- For each group, the labels in the majority subgroup are flipped with probability $\alpha$. For any data distribution $D \in \Delta(\mathcal{X} \times \mathcal{Y})$, we denote its noisy version by $\tilde{D}^\alpha$.

- It is assumed that the positive and negative data distribution have bounded support, and are sufficiently far apart such that the nearest neighbor of any sample is from the same group.

Overall we have:
$$D_A = \beta \tilde{D}_{\text{pos}}^\alpha + (1 - \beta) D_{\text{neg}}$$
$$D_B = (1 - \beta) D_{\text{pos}} + \beta \tilde{D}_{\text{neg}}^\alpha$$

The construction is illustrated in Figure 3 (Center).

### D.7.2 Stable Coexistence

**Proposition D.15** (Convex combinations). *Let $\boldsymbol{p} = (1 - p, p) \in \Delta(\{A, B\})$, and denote $\gamma = p - 2p\beta + \beta$. It holds that:*
$$D_{\boldsymbol{p}} = \gamma \tilde{D}_{\text{pos}}^{\frac{\beta(1-p)}{\gamma}\alpha} + (1 - \gamma) \tilde{D}_{\text{neg}}^{\frac{\beta p}{1-\gamma}\alpha}$$

*Proof.* For any mixture coefficient $q \in [0, 1]$, the label-flipped datasets satisfy:
$$(1 - q)\tilde{D}^\alpha + qD = \tilde{D}^{(1-q)\alpha} \tag{11}$$

The convex combination of the mixture coefficients of corresponding to $D_{\text{pos}}$ and $\tilde{D}_{\text{pos}}^\alpha$, is given by:
$$(1 - p)\beta + p(1 - \beta) = p + \beta - 2p\beta = \gamma \tag{12}$$

Plugging eq. (12) into eq. (11), we obtain that the corresponding $q$ for the positive mixture is:
$$q_{\text{pos}} = \frac{(1 - p)\beta}{(1 - p)\beta + p(1 - \beta)} = \frac{\beta(1 - p)}{\gamma}$$

and for the negative mixture:
$$q_{\text{neg}} = \frac{p\beta}{p\beta + (1 - p)(1 - \beta)} = \frac{\beta p}{1 - \gamma}$$

Combining the two equations above yields the result. $\qquad \square$

**Proposition D.16** (Reflection-exchange symmetry). *Let $\boldsymbol{p} = (1 - p, p)$, and let $\boldsymbol{p}' = (p, 1 - p)$. For the prediction setting defined above and for any classifier $h^*$ trained on a dataset $S \sim D_{\boldsymbol{p}}^n$, it holds that:*

$$\mathbb{E}_{S \sim D_{\boldsymbol{p}}}[\mathrm{acc}_{D_A}(h^*)] = \mathbb{E}_{S \sim D_{\boldsymbol{p}'}}[\mathrm{acc}_{D_B}(h^*)]$$

*Proof.* From symmetry of the definition. Intuitively, $\mathbb{E}_{S \sim D_{\boldsymbol{p}}}[\mathrm{acc}_{D_B}(h^*)]$ can be obtained by reflecting $\mathbb{E}_{S \sim D_{\boldsymbol{p}}}[\mathrm{acc}_{D_A}(h^*)]$ across the $\boldsymbol{p}$ axis. $\square$

**Proposition D.17** (Expected prediction of $k$-NN). *Let $D_x \in \Delta(\mathcal{X})$ be a distribution over features, let $\alpha > 0$ be a noise parameter, and let $k$ be an odd number greater or equal to $1$. Denote the label distribution by $D_y^\alpha = 1 - 2\mathrm{Bernoulli}(\alpha)$, and denote the joint feature-label distribution by $D = D_x \otimes D_y^\alpha \in \Delta(\mathcal{X} \times \mathcal{Y})$. Let $h : \mathcal{X} \to \{-1, 1\}$ be a $k$-Nearest-Neighbors ($k$-NN) classifier trained on a dataset $S \sim D^n$ with $n \geq k$, and let $x \in \mathcal{X}$. It holds that:*

$$\mathbb{E}_{S \sim D^n}[h(x)] = -1 + 2\phi_k(\alpha)$$

*Where $\phi_k(\alpha)$ the cumulative distribution function (CDF) of a $\mathrm{Binomial}(k, \alpha)$ random variable, taken at $\lfloor \frac{k}{2} \rfloor$:*

$$\phi_k(\alpha) = \Pr\left(\mathrm{Binomial}(k, \alpha) \leq \left\lfloor \frac{k}{2} \right\rfloor\right)$$

*Proof.* Denote the $k$-NN training set by $S \sim D^n$. For any $x \in \mathcal{X}$, denote its $k$ nearest neighbors by $N(x) \in (\mathcal{X} \times \mathcal{Y})^n$. Denote by $N_{\mathrm{neg}}(x)$ the number of neighbors with negative labels:

$$N_{\mathrm{neg}}(x) = |\{(x', y') \in N(x) \mid y' = -1\}|$$

Since features and labels in $D$ are assumed to be independent, the number of neighbors with negative labels is a binomial random variable:

$$N_{\mathrm{neg}}(x) \sim \mathrm{Binomial}(k, \alpha)$$

As a $k$-NN classifier predicts the label according to the majority label in $N(x)$, the expectation value of the label of $x$, taken over the randomness of the training set, depends on the cumulative distribution function of $N_{\mathrm{neg}}(x)$:

$$\mathbb{E}_{S \sim D^n}[h(x)] = -1 + 2\Pr(h(x) = 1)$$
$$= -1 + 2\Pr\left(N_{\mathrm{neg}}(x) \leq \left\lfloor \frac{k}{2} \right\rfloor\right)$$

And using the definition of $\phi_k(\alpha)$, we obtain:

$$\mathbb{E}_{S \sim D^n}[h(x)] = -1 + 2\phi_k(\alpha)$$

$\square$

**Lemma D.2** (Expected accuracy). *For the distributions $D_A$, $D_B$ defined above, let $\boldsymbol{p} = (1 - p, p) \in \Delta(\{A, B\})$, and let $k > 0$ be an odd integer. Denote by $h_{\boldsymbol{p}}$ the $k$-Nearest-Neighbor ($k$-NN) classifier trained on data sampled from $D_{\boldsymbol{p}}$. Assume that the training set contains feature vectors from both subgroups, and that the supports of the feature vector distributions in the datasets $D_{\mathrm{pos}}$, $D_{\mathrm{neg}}$ are sufficiently far apart. The expected accuracy of $h_{\boldsymbol{p}}$ with respect to $D_A$ is:*

$$\mathbb{E}_{S \sim D_{\boldsymbol{p}}^n}[\mathrm{acc}_{D_A}(h^*)] = \beta \left( \frac{1 + (1 - 2\alpha)\left(-1 + 2\phi_k\left(\frac{\beta(1-p)}{\gamma}\alpha\right)\right)}{2} \right)$$
$$+ (1 - \beta)\left( \frac{1 + (-1)\left(1 - 2\phi_k\left(\frac{\beta p}{1 - \gamma}\alpha\right)\right)}{2} \right) \tag{13}$$

*Where $\phi_k(\alpha)$ the cumulative distribution function of a $\mathrm{Binomial}(k, \alpha)$ variable, taken at $\lfloor \frac{k}{2} \rfloor$.*

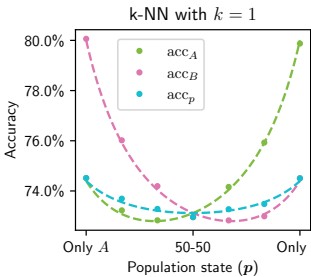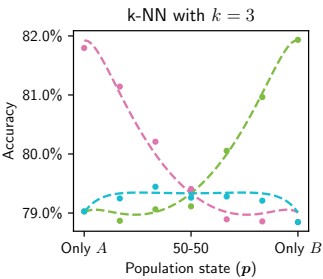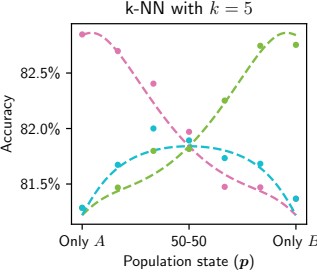

Figure 11: k-NN accuracy for the construction specified in Appendix D.7.1. The plots show alignment between the theoretical curves (Lemma D.2) and empirical simulations (mean accuracies represented by dots). For $k = 3, 5$ the game has a stable beneficial coexistence.

*Proof.* Denote the training set by $S = \{(x_i, y_i)\}_{i=1}^n \sim D_{\boldsymbol{p}}^n$, and denote the learned 1-NN classifier by $h_{\boldsymbol{p}}$. For any $x \in \mathcal{X}$, denote its nearest neighbor in the training set by $i^*(x) = \arg\min_{i \in [n]} d(x, x_i)$. The expected accuracy of $h_{\boldsymbol{p}}$ with respect to $D_A$ is given by:

$$\mathbb{E}_{S \sim D_{\boldsymbol{p}}^n}[\mathrm{acc}_{D_A}(h)] = \mathbb{E}_{S \sim D_{\boldsymbol{p}}^n, (x,y) \sim D_A}\left[\frac{1 + y y_{i^*(x)}}{2}\right]$$

Since the supports of the subgroups are sufficiently far apart, the nearest neighbor $i^*(x)$ always originates from the same subgroup as $x$. From this we obtain that $y$ and $y_{i^*(x)}$ are independent given the subgroup from which $(x, y)$ was sampled ($\in \{\mathrm{pos}, \mathrm{neg}\}$). Thus, for noisy data from the positive subgroup ($D_{\mathrm{pos}}^\alpha$), we have:

$$\begin{aligned}
\mathbb{E}_{S \sim D_{\boldsymbol{p}}^n}\left[\mathrm{acc}_{\tilde{D}_{\mathrm{pos}}^\alpha}(h)\right] &= \mathbb{E}_{S \sim D_{\boldsymbol{p}}^n, (x,y) \sim \tilde{D}_{\mathrm{pos}}^\alpha}\left[\frac{1 + y y_{i^*(x)}}{2}\right] \\
&= \frac{1 + \mathbb{E}[y]\mathbb{E}\left[y_{i^*(x)}\right]}{2} \\
&= \frac{1 + (1 - 2\alpha)\left(-1 + 2\phi_k\left(\frac{\beta(1-p)}{\gamma}\alpha\right)\right)}{2}
\end{aligned}$$

Where the identity $\mathbb{E}[y] = (1 - 2\alpha)$ is given by the definition of $\tilde{D}_{\mathrm{pos}}^\alpha$, the definition of $\gamma$ is given by Proposition D.15, and the identity $\mathbb{E}\left[y_{i^*(x)}\right] = \left(-1 + 2\phi_k\left(\frac{\beta(1-p)}{\gamma}\alpha\right)\right)$ is given by Proposition D.17.

Similarly, for clean data from the negative subgroup ($D_{\mathrm{neg}}$), we obtain:

$$\begin{aligned}
\mathbb{E}_{S \sim D_{\boldsymbol{p}}^n}\left[\mathrm{acc}_{D_{\mathrm{neg}}}(h)\right] &= \mathbb{E}_{S \sim D_{\boldsymbol{p}}^n, (x,y) \sim D_{\mathrm{neg}}}\left[\frac{1 + y y_{i^*(x)}}{2}\right] \\
&= \frac{1 + \mathbb{E}[y]\mathbb{E}\left[y_{i^*(x)}\right]}{2} \\
&= \frac{1 + (-1)\left(1 - 2\phi_k\left(\frac{\beta p}{1-\gamma}\alpha\right)\right)}{2}
\end{aligned}$$

and jointly:

$$\mathbb{E}_{S \sim D_{\boldsymbol{p}}^n}[\mathrm{acc}_A(h)] = \beta \mathbb{E}_{S \sim D_{\boldsymbol{p}}^n}\left[\mathrm{acc}_{\tilde{D}_{\mathrm{pos}}^\alpha}(h)\right] + (1 - \beta)\mathbb{E}_{S \sim D_{\boldsymbol{p}}^n}\left[\mathrm{acc}_{D_{\mathrm{neg}}}(h)\right]$$

And finally, plugging in the expected accuracy values calculated above yields the desired result. $\square$

**Proposition D.18.** *For $k = 1$, a prediction game induced by a 1-NN classifier trained on dataset with label noise has the following properties:*

1. *A stable coexistence equilibrium exists if and only if $\alpha \in \left(0, 1 - \frac{1}{2\beta}\right)$. If a stable coexistence equilibrium exists, then it exists at the uniform population state $\boldsymbol{p} = (0.5, 0.5)$.*

2. *The overall welfare at the coexistence state $\boldsymbol{p} = (0.5, 0.5)$ is lower than the overall welfare at extinction states $\boldsymbol{p} \in \{(1,0), (0,1)\}$.*

*Proof.* By applying Lemma D.2. When $k = 1$, the $\phi_k$ function satisfies:
$$\phi_{k=1}(\alpha) = \Pr\left(\text{Binomial}(1, \alpha) \leq 0\right) = 1 - \alpha$$

Plugging into eq. (13) and using the symmetry argument in Proposition D.16, we obtain:
$$\text{acc}_A\left(h^*_{\boldsymbol{p}=(1,0)}\right) = \text{acc}_B\left(h^*_{\boldsymbol{p}=(0,1)}\right) = 2\alpha\beta(\alpha - 1) + 1$$
$$\text{acc}_B\left(h^*_{\boldsymbol{p}=(1,0)}\right) = \text{acc}_A\left(h^*_{\boldsymbol{p}=(0,1)}\right) = 1 - \alpha$$
$$\text{acc}_A\left(h^*_{\boldsymbol{p}=(0.5,0.5)}\right) = \text{acc}_B\left(h^*_{\boldsymbol{p}=(0.5,0.5)}\right) = 2\alpha\beta(\alpha\beta - 1) + 1$$

For (1), a stable equilibrium exists if:
$$\text{acc}_A\left(h^*_{\boldsymbol{p}=(1,0)}\right) < \text{acc}_B\left(h^*_{\boldsymbol{p}=(1,0)}\right)$$

Plugging in the values calculated above:
$$2\alpha\beta(\alpha - 1) + 1 < 1 - \alpha$$

Which is satisfied when $\alpha \in \left(0, 1 - \frac{1}{2\beta}\right)$.

For (2), we note that $\beta \leq 1$, and therefore:
$$\text{acc}_A\left(h^*_{\boldsymbol{p}=(1,0)}\right) - \text{acc}_A\left(h^*_{\boldsymbol{p}=(0.5,0.5)}\right) = \alpha\beta(\alpha - \alpha\beta) = \alpha^2\beta(1 - \beta) \geq 0$$

$\square$

*Proof of Theorem D.4.* By Proposition D.18, using e.g. $(\alpha, \beta) = (0.2, 0.8)$. We observe that the conditions of the proposition are satisfied for this choice of constants, as $\alpha = 0.2 < 0.375 = 1 - \frac{1}{2 \cdot 0.8} = 1 - \frac{1}{2\beta}$. $\square$

### D.8 Hard-SVM With Finite Data

All practical learning algorithms must rely on a finite sample of data points for training a classifier. Learnability (in the PAC sense) implies that as data size grows, the sample becomes increasingly representative of the distribution. But the *rate* at which variance diminishes need not be equally across all regions of a distribution. In a sense, this means that one region can be learned 'faster' (i.e., with less data) than another. Our final example exploits this to enable coexistence.

**Theorem D.5.** *For the Hard-SVM algorithm, there exist linearly-separable data distributions induce an evolutionary prediction game with beneficial coexistence.*

Proof in Appendix D.8.2. The construction is illustrated in Figure 3 (Right), and described formally in Appendix D.8.1. Informally, the key property we leverage here is asymmetric class variance. In each group, one class has high variance, and the other class has low variance. When the dataset is of size $n$, this creates bias against the high-variance class, and thus convergence towards the optimal classifier at rate $O(n^{-1})$. When the two groups coexist, this bias is balanced, and the rate of convergence is exponential—leading to coexistence that is beneficial, but unstable.

**Definition D.13** (Hard-SVM; e.g. [54, Section 15.1])**.** *Given a linearly-separable training set $\{(x_i, y_i)\}_{i=1}^n \in \left(\mathbb{R}^d \times \{-1, 1\}\right)^n$, the Hard-SVM algorithm outputs a linear classifier $\text{sgn}(w^* \cdot x + b^*)$ which maximizes the margin of the decision hyperplane:*
$$(w^*, b^*) = \underset{(w,b);\|w\|=1}{\arg\max} \ \min_{i \in [n]} y_i(w \cdot x + b)$$

**Remark D.1** (One-dimensional Hard-SVM)**.** *For one dimensional data, denote by $x^+_{\min}$ the minimal $x$ with a positive label in the training set by $x^+_{\min} = \min_i \{x_i \mid y_i = 1\}$. Similarly, denote by $x^-_{\max}$ the maximal $x$ with a negative label in the training set by $x^-_{\max} = \max_i \{x_i \mid y_i = -1\}$. When $x^+_{\min} \geq x^-_{\max}$, the Hard-SVM decision margin is given by:*
$$x_{\text{margin}} = \frac{x^-_{\max} + x^+_{\min}}{2} \tag{14}$$

### D.8.1 Construction

Let $\varepsilon > 0$, and denote by $\delta(x = a)$ the Dirac delta distribution centered around $x = a$.

$$X_A \sim 0.5 \left( \text{Uniform}\left([0, 1]\right) + \varepsilon \right) + 0.5\delta(x = -\varepsilon)$$
$$X_B = -X_A$$
$$Y|X = \begin{cases} 1 & X \geq 0 \\ -1 & X < 0 \end{cases}$$

$D_A$ is the distribution of tuples $(X_A, Y|X_A)$, and $D_B$ is defined correspondingly as $D_B = (X_B, Y|X_B)$. The distributions are illustrated in Figure 3 (Right).

### D.8.2 Mutualistic Coexistence

**Proposition D.19.** *For a given positive integer $N$, Let $n \sim \text{Binomial}\left(N, \frac{1}{2}\right)$, let $x_1, \ldots, x_n \sim$ Uniform $([0, 1])$, and let $x_{\min} = \min_{i \in [n]} x_i$. Then for $N \geq 15$, it holds that:*

$$\mathbb{E}[x_{\min}] \geq \frac{1}{N}$$

*Proof.* Let $n \sim \text{Binomial}\left(N, \frac{1}{2}\right)$. By Hoeffding's inequality, it holds that:

$$\mathbb{P}_n\left[n - \frac{N}{2} \geq t\right] \leq e^{-\frac{2t^2}{N^2}}$$

And for $t = \frac{N}{4}$ we obtain:

$$\mathbb{P}_n\left[n \geq \frac{3N}{4}\right] \leq e^{-\frac{N}{8}} \tag{15}$$

For any fixed $n$, let $x_1, \ldots, x_n \sim$ Uniform $([0, 1])$, and denote the first order statistic among $\{x_i\}$ by $x_{\min,n} = \min_{i \in [n]} x_i$. As each $x_i$ is a uniform random variable over the unit interval, the first order statistic admits a beta distribution:

$$x_{\min,n} \sim \text{Beta}(1, n)$$

and therefore:

$$\mathbb{E}[x_{\min,n}] = \frac{1}{n+1} \tag{16}$$

Now consider the compound variable $x_{\min} = x_{\min,n}$ where $n \sim \text{Binomial}\left(N, \frac{1}{2}\right)$. Applying the law of total expectation:

$$\mathbb{E}[x_{\min}] = \mathbb{E}_n[\mathbb{E}[x_{\min} \mid n]]$$
$$= \mathbb{E}_n[\mathbb{E}[x_{\min,n}]]$$

By eq. (16) we obtain the expectation of $x_{\min,n}$:

$$\mathbb{E}[x_{\min}] = \mathbb{E}_n\left[\frac{1}{n+1}\right]$$

eq. (15) gives a lower bound on the expectation:

$$\mathbb{E}[x_{\min}] \geq \overbrace{\frac{1}{\frac{3N}{4}+1} \cdot \left(1 - e^{-\frac{N}{8}}\right)}^{n < \frac{3N}{4}} + \overbrace{0 \cdot e^{-\frac{N}{8}}}^{n \geq \frac{3N}{4}}$$

Then for $N \geq 15$ it holds that:

$$\mathbb{E}[x_{\min}] \geq \frac{1}{N}$$

As required.

$\square$

**Proposition D.20.** *For the distribution defined above, assume the Hard-SVM classifier is trained on dataset of size $n \geq 21$, and let $\varepsilon \in \left(0, \frac{1}{2n}\right)$. It holds that:*

1. *When the classifier is trained on $D_{\boldsymbol{p}}$ for $\boldsymbol{p} = (1, 0)$:*

$$\mathbb{E}[\text{acc}_{\boldsymbol{p}}(h)] = \mathbb{E}[\text{acc}_A(h)] \leq 1 - \frac{1}{8n}$$

2. *When the classifier is trained on $D_{\boldsymbol{p}}$ for $\boldsymbol{p} = (0, 1)$:*

$$\mathbb{E}[\text{acc}_{\boldsymbol{p}}(h)] = \mathbb{E}[\text{acc}_B(h)] \leq 1 - \frac{1}{8n}$$

3. *When the classifier is trained on $D_{\boldsymbol{p}}$ for $\boldsymbol{p} = (0.5, 0.5)$:*

$$\text{acc}_{\boldsymbol{p}}(h_n) = \mathbb{E}[\text{acc}_A(h)] = \mathbb{E}[\text{acc}_B(h)] \geq 1 - 2\left(1 - \frac{1}{4}\right)^n$$

4. *Coexistence is beneficial.*

*Proof.* For case (1), the Hard-SVM algorithm trains on data sampled from $D_A$. Denote the size of the dataset by $n$ and the corresponding Hard-SVM classfier by $h_n$. We note that $h_n$ is a random variable depending on the dataset. For a dataset $\{(x_i, y_i)\}_{i=1}^n \sim D_A^n$, Since the problem is one-dimensional, the Hard-SVM decision margin is given by eq. (14):

$$x_{\text{margin}} = \frac{x_{\max}^- + x_{\min}^+}{2} = \frac{x_{\min}^+ - \varepsilon}{2} > 0$$

And the expected accuracy of each group is given by:

$$\mathbb{E}[\text{acc}_A(h_n) \mid \omega^c] = \frac{1}{2} + \frac{1}{2}(1 - x_{\text{margin}} + \varepsilon) = 1 - \frac{x_{\min}^+ - \varepsilon}{4} + \frac{\varepsilon}{4}$$

By Proposition D.19 and for $n \geq 15$, it holds that $\mathbb{E}[x_{\text{margin}}] \geq \frac{1}{n} + \varepsilon$, and therefore:

$$\mathbb{E}[\text{acc}_A(h_n) \mid \omega^c] \leq 1 - \frac{1}{4n} + \frac{1}{8n} = 1 - \frac{1}{8n}$$

Case (2) follows from symmetry.

For case (3), denote by $\omega$ the bad event in which the training set $\{(x_i, y_i)\}_{i=1}^n \sim D_{\boldsymbol{p}}^n$ does not contain $x_i \in \{-\varepsilon, \varepsilon\}$. By the union bound, the probability of this event is bounded by $2\left(1 - \frac{1}{4}\right)^n$, and therefore the accuracy for $\boldsymbol{p} = (0.5, 0.5)$ is bounded from below by:

$$\text{acc}_{\boldsymbol{p}}(h_n) \geq 1 - 2\left(1 - \frac{1}{4}\right)^n$$

Finally, for (4), we note that $1 - 2\left(1 - \frac{1}{4}\right)^n \geq 1 - \frac{1}{8n}$ for all $n \geq 21$. $\square$

*Proof of Theorem D.5.* By Proposition D.20. $\square$

### D.9  Stabilizing Coexistence

*Proof of Proposition 2.* Let $F(\boldsymbol{p})$ be an evolutionary prediction game induced by an optimal learning algorithm $\mathcal{A}(\boldsymbol{p})$, and let $\boldsymbol{p}^*$ be an unstable coexistence equilibrium with full support ($|\text{support}(\boldsymbol{p}^*)| = K$). The game induced by $\mathcal{A}'(\boldsymbol{p})$ in the neighborhood of $\boldsymbol{p}^*$ is $F'(\boldsymbol{p}) = F(2\boldsymbol{p}^* - \boldsymbol{p})$. By Lemma D.1, the game $F(\boldsymbol{p})$ is a potential game, and therefore admits a potential function $f(\boldsymbol{p}) = \text{acc}_{\boldsymbol{p}}(h_{\boldsymbol{p}})$. Consider the potential function $f'(\boldsymbol{p}) = -f(2\boldsymbol{p}^* - \boldsymbol{p})$. By the chain rule, it holds that:

$$\nabla f'(\boldsymbol{p}) = -(\nabla f)(2\boldsymbol{p}^* - \boldsymbol{p}) \cdot \nabla(2\boldsymbol{p}^* - \boldsymbol{p}) = F(2\boldsymbol{p}^* - \boldsymbol{p}) = F'(\boldsymbol{p})$$

And therefore $f'(\boldsymbol{p})$ is a potential function for the game $F'(\boldsymbol{p})$ in the neighborhood of $\boldsymbol{p}^*$. The equilibrium $\boldsymbol{p}^*$ is unstable, and therefore by [52, Theorem 8.2.1] it is not a maximizer of $f(\boldsymbol{p})$. Moreover, by Proposition D.3 the potential function $f(\boldsymbol{p}) = \text{acc}_{\boldsymbol{p}}(h_{\boldsymbol{p}})$ is convex, and as $\boldsymbol{p}^*$ is assumed to have full support, it is also a global minimizer of $f(\boldsymbol{p})$. From this we conclude that $\boldsymbol{p}^*$ is a global maximizer of $f'(\boldsymbol{p})$, and therefore the equilibrium $\boldsymbol{p}^*$ is stable under the evolutionary game induced by $\mathcal{A}'$. $\square$

# E Implementation Details

**Code.** We implement our simulations and analysis in Python. Our synthetic-data experiments rely on scikit-learn [47] for learning algorithm implementations, our CIFAR-10 and MNIST experiments rely on PyTorch [46] and `ffcv` [38], and our ACSIncome experiment relies on scikit-learn and XGBoost [10]. We use `matplotlib` [29] for plotting, and `mpltern` [30] for ternary plots. Code is available at: https://github.com/edensaig/evolutionary-prediction-games.

**Hardware.** Synthetic data simulations were run on a single Macbook Pro laptop, with 16GB of RAM, M2 processor, and no GPU. Experiments involving neural networks (Section 6) were run on a dedicated server with an AMD EPYC 7502 CPU, 503GB of RAM, and an Nvidia RTX A4000 GPU.

**Runtime.** A single run of the complete synthetic data pipeline takes roughly 30 minutes on a laptop. A single repetition of each real-data experiment (i.e. sampling $h \sim \mathcal{A}(\boldsymbol{p})$ and computing marginal accuracies) takes roughly 10 minutes.

**Architectures.** For CIFAR-10, we use the Resnet-9 architecture and training code provided in the CIFAR-10 example code in the `ffcv` Github repository (`libffcv/ffcv`, commit `7885f40`), with modifications to control the probability of horizontal flips in training and testing. For MNIST, we use the convolutional neural network provided in the MNIST example code in the PyTorch examples Github repository (`pytorch/examples`, commit `37a1866`). The MNIST network has two convolutional layers, and two fully-connected layers, with dropout and max pooling. Training is performed for 200 epochs using SGD with learning rate $0.01$ and momentum $0.5$. For the ACSIncome experiments, we use scikit-learn and XGBoost classifiers with default regularization parameters (`LinearSVC`, `LogisticRegression`, `XGBClassifier`).

**Replicator dynamics simulation.** In the experiments, we use the discrete replicator equation for simulating evolutionary dynamics. The continuous-time replicator equation is:

$$\dot{p}_k = p_k \left( F_k(\boldsymbol{p}) - \bar{F}(\boldsymbol{p}) \right)$$

where $\bar{F}(\boldsymbol{p}) = \sum_k p_k F_k(\boldsymbol{p})$ is the average fitness across the population. In our simulations (e.g., Figures 2, 4), we use the discrete-time replicator equation given by Taylor and Jonker [57, eq. (3)]:

$$p_k^{t+1} = p_k^t \frac{F_k(\boldsymbol{p}) + 1}{\bar{F}(\boldsymbol{p}) + 1} \tag{17}$$

For the CIFAR-10 replicator simulation (Section 6.1), we discretize the two-group simplex into the grid $\{(1,0), (0.9, 0.1), \ldots, (0,1)\}$, precompute fitness values $F_k(\boldsymbol{p})$ at each grid point, and use linear interpolation to determine fitness at intermediate states.

**Confidence intervals.** In both experiments, variability is due to the random split within the train and test sets, and due to random noise when it is added. For numerical results, we report mean confidence intervals at $99\%$ confidence level.

# F Additional Empirical Evaluation

## F.1 Effect of Sampling Noise

The prediction games defined in Section 3 associate evolutionary fitness with accuracy in expectation over the training set (eq. (2), a reasonable assumption where sampling noise is small. Here we test the robustness of this assumption by quantifying the effect of sampling noise on long-term outcomes.

**Method.** We assess the effect of sampling noise on long term outcome by fitting a Gaussian process to the outcomes of different sampled training sets, and analyzing the resulting stochastic dynamics for varying initial conditions. We use the raw data from the CIFAR-10 experiment introduced in Section 6.1 (group accuracies for each training set sample). We fit a Gaussian process (GP) with an RBF kernel with length scale $0.1$, using the skicit-learn implementation with alpha regularization parameter $4$. To generate stochastic replicator dynamics, we replace the deterministic fitness terms in

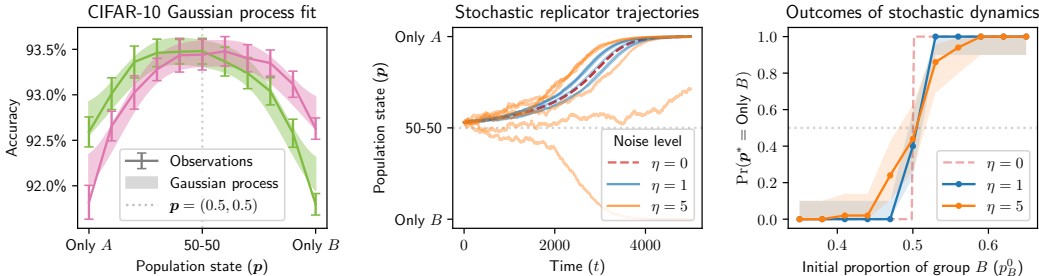

Figure 12: Effect of sampling noise in the CIFAR-10 setting (Appendix F.1). **(Left)** Gaussian process fit to the raw observations. Error bars indicate standard deviation of data, and shaded areas indicate standard deviation of the Gaussian process. **(Center)** Trajectory samples of the induced stochastic replicator dynamics, for different simulated noise levels $\alpha$. **(Right)** Distribution of long-term outcomes as a function of initial condition.

eq. (17) with samples from the Gaussian process. We simulate different noise levels by multiplying the Gaussian process covariance by a constant $\eta \geq 0$, such that $\eta = 0$ coincides with the deterministic dynamics, $\eta = 1$ coincides with the Gaussian process fit to the original data, and $\eta = 5$ represents a simulated five-fold increase in compared to the observed sampling noise. For each $\eta$, we simulate stochastic dynamics for a range of initial states $p_B^0 \in [0.35, 0.65]$, and measure the distribution of long term outcomes (dominance of group $A$ vs. group $B$) over repeated realizations.

**Results.** Gaussian process fit is illustrated in Figure 12 (Left), stochastic trajectories are presented in Figure 12 (Center), and distributions of long-term outcomes are presented in Figure 13 (Right). When there is no sampling noise (i.e., $\eta = 0$), group $B$ dominates iff $p_B^0 > 0.5$ as expected. With the natural variance of the task (i.e., using the inferred Gaussian process covariance and $\eta = 1$), results are relatively robust, and group $B$ eventually dominates with high probability for all $p_B^0 > 0.5$. When variance is excessively increased (i.e., $\eta = 5$), outcomes also become noisy, and dynamics can converge to group $A$ with some probability even for $p_B^0 \approx 0.55$.

## F.2 Time to Convergence

For unstable equilibria, a key question is how fast the system transitions to a stable equilibrium state, and what affects this rate. Here we measure the time to reach (approximate) domination by one group as a function of the initial state.

**Method.** We extend the analysis of the CIFAR-10 experiment, described in Section 6.1. To obtain a smooth population game, we use the mean curves of the Gaussian process described in Appendix F.1. For each initial condition, we simulate the replicator dynamics, and record the number of steps until one group is approximately dominant, formally $\min p^t \leq 0.01$.

**Results.** Figure 13 (Center) presents the measured convergence times. Results show that the time to convergence is roughly linear in $p_B^0$ for $p_B^0 \in [0.1, 0.4] \cup [0.6, 0.9]$, but tends to infinity as the initial state approaches the uniform distribution ($p_B^0 \to 0.5$). The implication is that for initially balanced populations, natural selection forces are relatively weak, and and higher-order effects (such as sampling noise, or other exogenous forces) can dominate the dynamics.

## F.3 Sensitivity Analysis for Stabilization

Our proposed stabilization mechanism (Section 5.2) assumes access to the true equilibrium $p^*$, which may not be known in practice. Although our experiments in Section 6.1 use an estimated $p^*$ and results comply with our theoretical findings, they do reveal how robust outcomes are to misspecification of $p^*$. For this, we present a sensitivity analysis quantifying the deviation of the reached state from the estimated equilibrium $p^*$.

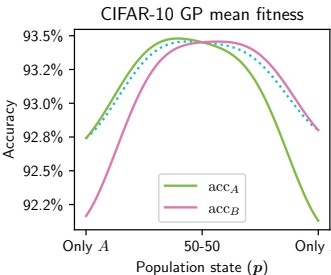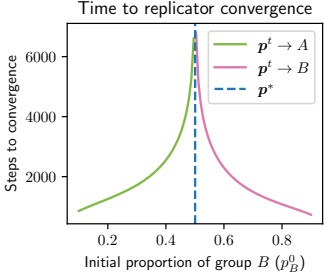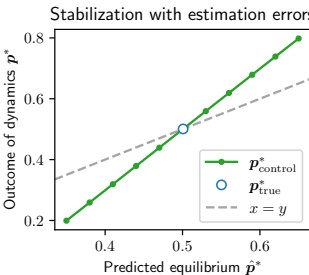

Figure 13: Empirical analysis of deterministic sensitivity aspects (Appendices F.2, F.3). **(Left)** Evolutionary prediction game induced by CIFAR-10 with horizontal flips and Resnet-9. Curves are smoothed by fitting a Gaussian process, as described in Appendix F.1. **(Center)** Time to convergence for different initial conditions. Curve is relatively linear far from the unstable equilibrium, but approach infinity as the initial condition approaches the unstable equilibrium. **(Right)** Sensitivity to equilibrium estimation errors in steering, showing a linear relation.

**Method.** We simulate the stabilization method presented in Section 5.2 for varying population states around the uniform population state equilibrium, and measure the resulting long-term outcomes of the dynamics.

**Results.** Results are presented in Figure 13 (Right), and show a linear relation between the estimated equilibrium and the eventual outcome of the stabilized dynamics. An additional observation is that misspecification affects only the population composition, and not welfare. This is because the algorithm ensures that the returned classifier acts "as if" the system is under equilibrium, in which accuracy for both groups is the same.

### F.4 Three-Group Dynamics for Different Learning Algorithms

To further establish the results of Section 6.3, we compare three-group dynamics induced by different algorithms in the same setting.

**Method.** We follow the same procedure presented in Section 6.3, and compute the dynamics for a linear SVM (from scikit-learn), logistic regression (scikit-learn), and an XGBoost classifier, all with default regularization parameters. Replicator dynamics are simulated using Equation (17), and basins of attraction are computed by simulating the dynamics until convergence. The initial condition for all algorithms is $\boldsymbol{p}^0 = \big(p_{(\text{TX})}, p_{(\text{NY})}, p_{(\text{CA})}\big) = (0.375, 0.5, 0.125)$.

**Results.** Results are presented in Figure 14. For identical initial conditions, each learning algorithm leads to a qualitatively different outcome: dynamics induced by the linear SVM converge to dominance of the Texas (TX) user group, dynamics induced by logistic regression converge towards a coexistence saddle point, and XGBoost dynamics converge towards dominance of the California (CA) user group. In particular, for the logistic regression system, note that the population state evolves along the attraction basin boundary (separatrix). By the definition of the replicator dynamics, states along the attraction basin boundary satisfy $\text{acc}_{(\text{CA})}(h_{\boldsymbol{p}}) = \text{acc}_{(\text{TX})}(h_{\boldsymbol{p}})$, and therefore exhibit a two-phase phenomenon similar to the one presented in Section 6.3 (an early balance followed by competition between the remaining groups). In contrast, all interior points in the XGBoost dynamics evolve toward CA domination, precluding the existence of the two-phase phenomenon in this setting.

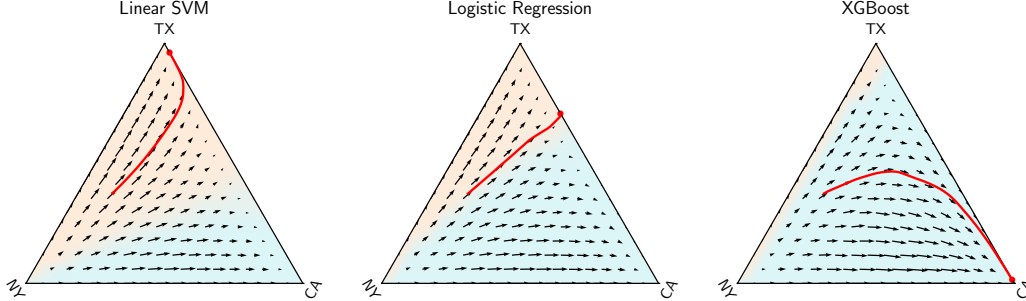

Figure 14: Comparing three-group replicator dynamics across different learning algorithms for the ACSIncome prediction task (Appendix F.4). Background colors represent basins of attraction, and red lines represent trajectories starting from the same initial condition.

