# OpenReview forum: "Evolutionary Prediction Games"
_NeurIPS.cc/2025/Conference — NeurIPS 2025 poster_

### Official Review · Reviewer_YgUX · 2025-06-28

**Clarity:** 3
**Significance:** 3
**Originality:** 3
**Rating:** 5
**Confidence:** 4

**Summary:**

This paper introduces evolutionary predictive games, which are a class of evolutionary games in which a classifier and the underlying data distribution co-evolve in a feedback loop with dynamics driven by payoffs defined by average prediction performance. Analysis is carried out to show that, under perfect learning (optimization) and perfect computation of population-level expectations, the evolutionary predictive games monotonically increase in the classifier's accuracy and converge to Nash equilibria with singleton support (meaning only one group of the data distribution remains at play). It is shown that various alternative problem setups can give rise to co-existence of multiple groups at (stable) Nash equilibrium, e.g., by using surrogate loss functions or different learning algorithms. Experiments on MNIST and CIFAR-10 are conducted to empirically corroborate the theory.

**Questions:**

1. The introduction is very sparse in terms of references. It would be good to support your motivating assertions with citations. For example: "However, while generally beneficial, promoting accuracy blindly can have unexpected, and in some cases undesired, consequences." Can you give references that demonstrate such undesired consequences? For instance, there is a body of influential work that studies accuracy-robustness tradeoffs in ML models that you may want to mention and cite. There are currently no references in the first four paragraphs.
2. Another important example of where there really should be some specific references cited is the list given in the sentence "Such dynamics arise across domains:..."
3. ‘compete’ Here, you are using single quotes, but before, you used double quotes. It would be good to maintain consistency in your notations and writing style.
4. Figure 1 is not introduced or discussed at all in the main text of the paper. It would be good to do this, to indicate to the reader when is a natural time to go and look at Figure 1.
5. Proposition 1: The "overall accuracy equality fairness criterion in expectation" property is referred to in Proposition 1, but its definition is deferred to the Appendix. It is fine to defer the proof of Proposition 1 to the Appendix, but it is preferable to include all definitions used in the main text, in the main text, for self-containedness reasons.
6. "Overall Accuracy Equality Fairness": The way you have defined this criterion doesn't quite seem to fully capture "fairness" to me. Specifically, how can you say that you achieve fairness if the equality in prediction accuracy only holds for groups that remain at play in Nash equilibrium, unless the Nash equilibrium has full support? For example, if the dynamics converge to a Nash equilibrium with a single index of support, meaning all but one group have effectively been driven out of the game, then how is that fair for the groups that have been driven out?
7. "fixed points of the dynamical system coincide with the Nash equilibria" You may want to mention that this is a property known as "Nash stationarity."
8. It seems a little misleading in the paragraph of line 169 to claim that all of your three assumptions are satisfied by the dynamics described in Section 1 and Appendix C, and to throw in the "exception" that imitative dynamics don't satisfy Nash stationarity into the footnote 1, and to not mention in the main text that the replicator dynamics are imitative dynamics. That is, the replicator dynamics that you seem to focus on do not satisfy your assumption of Nash stationarity, but the reader would need to piece this together themselves by carefully reading footnote 1 and identifying that replicator dynamics are imitative (which you don't explicitly define or mention). If your main theoretical results hold for dynamics that have unstable non-Nash stationary points, then why bother making the Nash stationarity assumption at all?
9. Line 197: There are two periods in the sentence with footnote 2.
10. Line 298 typo: "dynamcis"

**Ethical Concerns:**

["NO or VERY MINOR ethics concerns only"]

**Final Justification:**

I have read the author responses to all of the (also positive) reviews. The authors have very clearly and carefully addressed all of my concerns, and, in particular, have better clarified the key technical assumptions underlying their stability theory, and the nuances behind conventional fairness definitions, how they inadequately capture the notion of groups being driven out of games entirely, and how the contributions of this paper elucidate these inadequacies. With these clarifications and associated revisions in mind, I find this to be a strong theoretical paper (and, although I cannot see the new Folktables dataset experiment, the authors have also convinced me that their empirical evaluations have also strengthened), with deep insights into the connections between evolutionary dynamics and fairness in learning. Therefore, I raise my score from 4 to 5.

**Limitations:**

yes

**Paper Formatting Concerns:**

No major formatting issues.

**Quality:**

3

**Strengths And Weaknesses:**

Quality: After carefully reviewing the theoretical developments of the paper, the submission appears to be rigorous and technically sound. The key claim (namely, that an analysis is provided for the proposed evolutionary prediction games showing that they exhibit competitive exclusion in the idealized setting of unlimited knowledge of the data distribution, and that stable co-existence can arise in the setting of finite data) is supported through the theoretical developments, and corroborated by the experiments. The theoretical and experimental methods used are appropriate for the problem at hand, and the manuscript culminates into a nice, "complete" work. The authors do not explicitly discuss weaknesses or limitations of their work (see my "Questions" #6 and #8 for specific instances where this could be improved), but they do briefly mention avenues for future work and related discussion in the conclusion of the paper.

Clarity: For the most part, the submission is well-written and well-organized. I have enumerated a few small suggestions for improvement to the writing and presentation in the "Questions" below. Enough details of the theoretical proofs and experimental setup are provided (in the supplemental materials) to reproduce the results. The primary weakness in the paper's clarity is that the introduction is quite sparse in terms of the number of references, making it potentially difficult for a reader to jump to relevant work that is prerequisite in fully understanding and appreciating the work at hand. See my "Question" #1 below for more details.

Significance: I find the theoretical findings of competitive exclusion in the ideal setting and stable co-existence in other, non-ideal settings, to be significant to those interested in the intersection of evolutionary game theory and ML. The results are likely not very significant to ML as a whole, which is okay. I suspect that other researchers are likely to build upon the theoretical framework and ideas proposed here. The submission does not necessarily address tasks in better ways than previous works, but rather it provides a novel theoretical framework that, to the best of my knowledge, is unique to the community. The theory developed indeed advances our understanding of the proposed evolutionary prediction games, as the conclusions are not immediately or intuitively obvious.

Originality: Not only does the work introduce a novel problem setup at the interesting intersection of evolutionary game theory and machine learning, but it provides novel theoretical insights about the problem. The authors could better clarify how their work differs from previous contributions by providing more citations throughout the motivations, problem setup, and introduction of the paper (see my "Question" #1 below).

---

> ### Author Rebuttal · Authors · 2025-07-31
>
> Thank you for the positive feedback, and for the very constructive and thorough review! We address your questions below:
>
> > (1) The introduction is very sparse in terms of references. It would be good to support your motivating assertions with citations. For example: "However, while generally beneficial, promoting accuracy blindly can have unexpected, and in some cases undesired, consequences." Can you give references that demonstrate such undesired consequences? For instance, there is a body of influential work that studies accuracy-robustness tradeoffs in ML models that you may want to mention and cite. There are currently no references in the first four paragraphs.
>
> Thank you for pointing this out, we will gladly add supporting references throughout. One type of unintended consequences that we had in mind are those of concern in recommender systems, media platforms, and online markets, such as: content homogenization, opinion polarization, filter bubbles and echo chambers, hyper-popularization, and diminishing tail consumption. We will add references to papers which connect these phenomena to prediction. Another type are emergent social biases, such as those studied in the fairness literature, which have a well-established connection to accuracy maximization. We did not consider the connection to robustness that you mention, but this certainly makes sense, and we will gladly discuss this point as well.
>
> > (2) Another important example of where there really should be some specific references cited is the list given in the sentence "Such dynamics arise across domains:..."
>
> Thank you for raising this point. We will expand the list of references.
>
> > (3) ‘compete’ Here, you are using single quotes, but before, you used double quotes. It would be good to maintain consistency in your notations and writing style.
>
> Fixed!
>
> > (4) Figure 1 is not introduced or discussed at all in the main text of the paper. It would be good to do this, to indicate to the reader when is a natural time to go and look at Figure 1.
>
> Thank you for noticing this. We’ve added a reference at the end of Section 2.
>
> > (5) Proposition 1: The "overall accuracy equality fairness criterion in expectation" property is referred to in Proposition 1, but its definition is deferred to the Appendix. It is fine to defer the proof of Proposition 1 to the Appendix, but it is preferable to include all definitions used in the main text, in the main text, for self-containedness reasons.
>
> Thanks for this suggestion, we will move the definition to the main body of the paper.
>
> > (6) "Overall Accuracy Equality Fairness": The way you have defined this criterion doesn't quite seem to fully capture "fairness" to me. Specifically, how can you say that you achieve fairness if the equality in prediction accuracy only holds for groups that remain at play in Nash equilibrium, unless the Nash equilibrium has full support? For example, if the dynamics converge to a Nash equilibrium with a single index of support, meaning all but one group have effectively been driven out of the game, then how is that fair for the groups that have been driven out?
>
> This is a great question. The incompatibility between traditional fairness definitions and population dynamics *is precisely the point* we are trying to make. We apologize if this was unclear, and will make sure to clarify it.
>
> One straightforward implication is that if you train a classifier to comply with fairness constraints, dynamics can cause this guarantee to break over time; this idea has already been explored to some extent in the fairness literature, mostly following [1]. But another more nuanced implication – which you describe precisely – is that a system can *appear* to comply with a fairness definition, but only because some groups were already driven out of the system, and fairness is measured only over the groups that remain.
>
> Our message here is that fairness should consider not only what has happened, but also what *could* have happened (in our case, in terms of which groups to compare). In fact, we think that settings in which a strict subset of groups exists for the majority of the time is very likely, and have included an additional experiment that demonstrates this: Using the Folktables dataset [2], we show that in a three-group setting ($K=3$), dynamics first drive out one group (while keeping the other two balanced), and only then introduce competition between the two groups that remain. Auditing the prediction algorithm for fairness after the first group is driven out provides an appearance of fairness for the two groups that remain, with little ability to identify the preceding existence of the group that was driven out of the system.
> We believe this form of “counterfactual” fairness has not received sufficient attention in the literature, and hope our work can promote discussion and motivate future work on this topic.
>
>
> > (7) "fixed points of the dynamical system coincide with the Nash equilibria" You may want to mention that this is a property known as "Nash stationarity."
>
> We agree. We do use the term “Nash stationarity” in Appendix D.2, but will make sure to promote its usage to the main body of the paper for clarity.
>
> > (8) It seems a little misleading in the paragraph of line 169 to claim that all of your three assumptions are satisfied by the dynamics described in Section 1 and Appendix C, and to throw in the "exception" that imitative dynamics don't satisfy Nash stationarity into the footnote 1, and to not mention in the main text that the replicator dynamics are imitative dynamics. That is, the replicator dynamics that you seem to focus on do not satisfy your assumption of Nash stationarity, but the reader would need to piece this together themselves by carefully reading footnote 1 and identifying that replicator dynamics are imitative (which you don't explicitly define or mention). If your main theoretical results hold for dynamics that have unstable non-Nash stationary points, then why bother making the Nash stationarity assumption at all?
>
>
>
> Thank you for raising this point. Our intent was to simplify the presentation, but this perhaps came at the expense of clarity and precision. Overall we were very careful to explicitly and accurately state any and all assumptions that we make, and so we apologize if this part in particular caused confusion. We clarify our assumptions below, and will update the paper accordingly.
>
> In more detail, our theoretical analysis inherits the assumptions about dynamics from the convergence guarantees provided in [3], which are proven under the assumptions of either Nash-stationary *or* imitative dynamics (see Appendix D.2 for a summary of the relevant results we rely on). While imitative dynamics indeed do not satisfy the Nash stationary assumption, they do satisfy a similar condition with respect to a slightly-coarser equilibrium concept of the population game – namely, stationary points of imitative dynamics correspond to “restricted equilibria” of the game, a set that in particular includes all Nash equilibria (see, e.g., [3, Section 8.1]). For our purposes, this particular form of deviation from Nash stationarity does not affect our results or analysis.
>
> For this reason, we chose to focus in the exposition primarily on Nash-stationary, which captures more cleanly the structure we exploit, and in our mind provides better intuition. Nonetheless, we will make sure to revise this section to more precisely state the conditions under which our results hold, in line with the above description.
>
> We would however like to note that the fact that our results hold for imitative dynamics does not imply that they hold for any dynamics permitting arbitrary unstable non-Nash points. The question of whether results hold also for other dynamics with milder assumptions is very interesting, and we will highlight this in the paper.
>
> > (9,10) Line 197: There are two periods in the sentence with footnote 2.; Line 298 typo: "dynamcis"
>
> Fixed!
>
> –-
>
> And finally, we would like to thank you once again for the positive and constructive review! Please let us know if our response helps address your concerns. If any questions remain, or if additional ideas come up, we would be more than happy to continue the conversation.
>
> References:
> * [1] “Delayed Impact of Fair Machine Learning”, Liu et al., IJCAI 2019.
> * [2] “Retiring Adult: New Datasets for Fair Machine Learning”, Ding et al., NeurIPS 2021.
> * [3] “Population Games and Evolutionary Dynamics”, Sandholm, 2010.

---

> > ### Comment · Reviewer_YgUX · 2025-08-02
> >
> > I thank the authors for their very thorough responses and revisions. All of my concerns have been addressed. In particular, the authors have clarified the key technical assumptions underlying their stability theory (i.e., reliance on restricted equilibria rather than Nash stationarity), and the nuances behind conventional fairness definitions, and how they inadequately capture the notion of groups being driven out of games entirely, and how the contributions of this paper elucidate these inadequacies. With these clarifications and associated revisions in mind, I find this to be a strong theoretical paper, with deep insights into the connections between evolutionary dynamics and fairness in learning. I also appreciate the new experiment showing the 3-group example in which one group is driven out at the start, further highlighting the nuance of "counterfactual" fairness. Therefore, I raise my score from 4 to 5.

---

### Official Review · Reviewer_iKna · 2025-06-30

**Clarity:** 2
**Significance:** 2
**Originality:** 2
**Rating:** 4
**Confidence:** 3

**Summary:**

Under the framework of evolutionary game theory, this paper provides evolutionary prediction games to study the interplay between the accuracy of prediction algorithms and user retention rates. Moreover, a theoretical analysis is provided to investigate the equilibria of the games. This is interesting. However, some details of the evolutionary prediction games should be clarified, like how does a user change the group. In addition, from the perspective of improving the performance of prediction algorithms, this paper lacks a relevant study and explanation.

**Questions:**

Please see the weaknesses above.

**Ethical Concerns:**

["NO or VERY MINOR ethics concerns only"]

**Final Justification:**

The authors have explained the rationale behind dividing the population into multiple groups, clarified the process by which individual users make decisions on group selection, and stated that they have added experiments comparing different prediction algorithms. Based on the authors’ response, I now appreciate the significance of studying the co-evolution of prediction algorithms and individual strategies within an evolutionary game-theoretic framework. I have therefore decided to raise my evaluation of the paper.

**Limitations:**

Based on the study of the co-evolutionary dynamics between the prediction algorithms and user behavior, can the authors give some insights into the design of prediction algorithms?

**Quality:**

2

**Strengths And Weaknesses:**

**Strengths**
This paper originally presents evolutionary prediction games to study the feedback loop between the accuracy of prediction algorithms and user behavior. This provides a new perspective for prediction algorithms.

**Weaknesses**
1. The introduction is not easy to follow. The accuracy of sentences should be further improved. For example, in line 19, what kinds of decision-making are included in personalized education? In line 23, what does social outcome mean here? I think the experiments do not focus on any data from real platforms. In line 25, "But in social settings, ...", I think this is not about social settings. For any platforms, which collect data from users to train their prediction algorithms, they may exhibit the feedback loop between users and the accuracy of prediction algorithms. In line 35, "Common to ...", What does group structure mean here? What does the relation between prediction algorithm and group?
2. Why do you divide the population into $K$ groups? Is it necessary for the evolutionary prediction games?
3. In line 104, the group-specific distribution $D_k$ is assumed to be fixed. Meanwhile, users are able to decide to leave a group and reshape the population composition $p$. Thus, once a user leaves the group $k$, the distribution $D_k$ changes. This seems to contradict the assumption.
4. In lines 111-113, can you give more details about how users change their groups? For example, under what conditions will a user leave their current group? How does a user choose which new group to join?
4. In line 211, how can you ensure the existence of $A^{opt}$? What are the conditions for the existence of $A^{opt}$?
6. Theorem 2 states the existence of such an evolutionary prediction game. What are the conditions for the existence?
7. This paper studies the co-evolutionary dynamics between prediction algorithms and user behavior. Based on the stable states of the dynamics, can you give some insights into the design of prediction algorithms?
8. In experiments, can you give a comparison between different prediction algorithms?

---

> ### Author Rebuttal · Authors · 2025-07-31
>
> Thank you for your feedback and detailed review. In your review, the “weaknesses” section consists mostly of questions, with one additional question under “limitations”. Our response below fully addresses all of these questions. We hope you would be willing to reconsider your evaluation given our response herein.
>
> > (1.a) In line 19, what kinds of decision-making are included in personalized education?
>
> Personalized education uses predictions to improve learning experiences. For example, a common task is to construct a personalized learning curriculum for students based on predicted difficulty levels of possible exercises. See [1] for a recent survey.
>
> > (1.b) In line 23, what does social outcome mean here? … In line 25, "But in social settings, ...", I think this is not about social settings.
>
> We use the term “social” as it is used in economics, and within the machine learning literature, as it is used for example in performative prediction (see, e.g., [2,3]). The term “social outcome” refers to the positive or negative effects on society that occur as a result of the interaction with the prediction system. Accordingly, the term “social settings” refers to settings in which the prediction algorithm’s effect on society is significant and should be considered. This includes platforms that collect data from their users to train their prediction algorithms, assuming this process also has an impact on those users.
>
> > (1.c) In line 35, "Common to ...", What does group structure mean here? What does the relation between prediction algorithm and group?
>
> “Group structure” means that the population responds to a deployed classifier in a way which can be modeled as a group-wise operation for some partition of the population into groups (e.g. according to demographics, behavioral patterns, social group etc.). We give formal examples for such relations in Appendix C (“Microfoundations”).
>
> The training data for the prediction algorithm is sampled from the mixture of groups, and the mixture coefficients change in response to prediction quality. Illustration is provided in Figure 1, and formal definitions are provided in Section 3.
>
> > (2) Why do you divide the population into $K$ groups? Is it necessary for the evolutionary prediction games?
>
> We focus on a finite number of groups $K$ for two main reasons. First, prior work shows that learning dynamics under infinitely-many or a continuum of groups can be considerably more involved (see for example [4], and also Appendix B.1), and therefore we believe that working with a finite number of groups is a reasonable first step. Second, the notion of a finite set of groups is common in the literature, for example in fairness. Note however that our notion of “group” is much more general as it need not be based on some intrinsic, immutable property, and can be quite flexible, as illustrated in Appendix C.
>
> > (3) In line 104, the group-specific distribution $D_k$ is assumed to be fixed. Meanwhile, users are able to decide to leave a group and reshape the population composition $p$. Thus, once a user leaves the group $k$, the distribution $D_k$ changes. This seems to contradict the assumption.
>
> We make a distinction between the group distribution $D_k$, which is indeed assumed to be fixed, and the actual users from group $k$ that are a part of a sampled set of users, which is not assumed to be fixed (and is very much expected to vary). The meaning of group relative size “increasing” or “decreasing” have different interpretations and manifest differently in different types of dynamics. For example, in replication processes users invite peers from the same group, whereas in imitation users are assumed to copy the behavior of peers that are more successful (see Appendix C). Similarly, some dynamics allow individual users to remain in the system over time, while others assume different users are sampled at each time step. All of these comply with the assumption that there *exists* some fixed group distribution $D_k$ from which data is sampled, and we therefore view this as a reasonable assumption.
>
> > (4) In lines 111-113, can you give more details about how users change their groups? For example, under what conditions will a user leave their current group? How does a user choose which new group to join?
>
> As described above, these depend on the specific type of dynamics under consideration. Our results are general in the sense that they apply to a large and expressive family of dynamics that adhere to certain global properties, and which includes many varied types of micro-dynamics. For examples and further details please see Appendix C.
>
> > (5) In line 211, how can you ensure the existence of $A^{opt}$? What are the conditions for the existence of $A^{opt}$?
>
> The oracle algorithm $A^{\\mathrm{opt}}$ is a theoretical object that we use as a construct in our analysis. It is guaranteed to exist whenever the corresponding optimization problem has a solution (Equation (4)). This will depend on the choice of model class $\\mathcal{H}$. One example is when $\\mathcal{H}$ is compact (e.g. a space of parameterized classifiers with bounded norm), which holds for many common classes. Another example is when $\\mathcal{H}$ includes all functions, in which case $A^{opt}$ yields the Bayes-optimal classifier.
>
> > (6) Theorem 2 states the existence of such an evolutionary prediction game. What are the conditions for the existence?
>
> A formal constructive proof is provided in Appendix D.6. An illustration of the approach is provided in Figure 3 (Left).
>
> > (7) This paper studies the co-evolutionary dynamics between prediction algorithms and user behavior. Based on the stable states of the dynamics, can you give some insights into the design of prediction algorithms?
>
> Certainly. One central insight is that the natural selection forces induced by learning algorithms tend to push the population towards single-group dominance as the algorithms become better at generalization. Algorithm designers should be aware of this property, and plan intervention methods if single-group dominance is not aligned with the system goals. For example, this can be achieved by employing our stabilizing algorithm from Section 5.2. Another example is to facilitate an exogenous influx of users of diverse groups, e.g. via marketing efforts. We will expand the discussion on this in the final version of the paper.
>
> > (8) In experiments, can you give a comparison between different prediction algorithms?
>
> We have added new experiments on empirical fairness datasets, including a comparison between different learning algorithms. See our response to Reviewer oLKx.
>
> --
>
> References:
> * [1] “A Comprehensive Survey on Deep Learning Techniques in Educational Data Mining”, Lin et al., Data Science and Engineering 2025.
> * [2] “The predictability of social events”, Grunberg & Modigliani, Journal of Political Economy 1954.
> * [3] “Performative Prediction: Past and Future”, Hardt & Mendler-Dünner, to appear in Statistical Science 2025.
> * [4] “Strategic distribution shift of interacting agents via coupled gradient flows”, Conger et al., NeurIPS 2023.

---

> ### Comment · Reviewer_iKna · 2025-08-01
>
> Your response addressed my concerns, and I have increased my rating score.

---

### Official Review · Reviewer_Azxr · 2025-07-02

**Clarity:** 3
**Significance:** 2
**Originality:** 2
**Rating:** 5
**Confidence:** 4

**Summary:**

This paper instantiates a model of a population of user participation decisions in a prediction system. The population is broken into types. If predictions about a type are good, more individuals of that type will join the system. If predictions about a type are less good, fewer individuals will participate. The system at a given point in time is described by the type-makeup of the population and different population distributions may yield different predictors (optimized for accuracy). This describes situations like content recommendation systems where user participation decisions and prediction quality may generate positive feedback loops. The model is instantiated as an evolutionary game. The main results in the paper are theoretical. They include:
- Thm 1: If the accuracy of the predictor is uniform over a hypothesis class, accuracy improves, there exists a stable (attracting neighboring states) equilibrium, one type makes up the whole population in any stable equilibrium, and there may be unstable equilibria with more than two types.
- Thm 2: If the predictor is optimized by a proxy loss, there exists a equilibrium with multiple types that is stable. The intuition is that the proxy induces
- Prop 2: If the data used to train the predictive system can be reweighted, unstable equilibria can be made stable.

**Questions:**

What does it mean to be fitness maximizing? I don’t see where it was defined. Is it the maximum (over p) fitness summed over types?

**Ethical Concerns:**

["NO or VERY MINOR ethics concerns only"]

**Final Justification:**

This is a strong paper which addressing the important question of performativity/dynamics in recommendations. The authors formulate this question naturally in terms of evolutionary games and include novel elements that tailor the setting to content/recommendations. Their analysis and discussion support their conclusions and I am pleased to see the additions they propose to make to the paper.

**Limitations:**

Yes.

**Paper Formatting Concerns:**

None.

**Quality:**

3

**Strengths And Weaknesses:**

Strengths:

- The analysis seems correct.
- The area of prediction under dynamics or performativity is interesting and under-explored.

Weaknesses:

- Real platforms seem to have a diversity of users, it seems like the results in the paper are discordant with real, e.g., recommendation systems. What explains this? Is it that the evolution dynamics are not as powerful in real life? Is it explained by the proxy objectives that platforms must use? Are the platforms training on a diversity of data? Are platforms not in equilibrium? I would have liked more grappling with the implications of these results for platform practices or how we understand them.
- Generally, I did not think the theoretical take-aways were very surprising. Theorem 2 in particular seems weak since it is an existence proof that relaxing optimality of the objective can yield stable multi-group equilibria.
- I did not feel the related work on evolutionary prediction games was sufficiently detailed. Results in this paper seem like they might be special cases of general analyses of population games. (Intuitively, it seems like the conclusion of Thm 1 should hold quite generally and any setting where growth begets growth should yield these kinds of conclusions.) On the other hand, I am not familiar enough with population games to verify whether this kind of claim is proved elsewhere. The related work says “competitive exclusion” is well-studied, which suggests to me there should be related results.
- I did not find the experiments particularly informative. It would be nice to highlight qualitative take-aways or more nuances rather than just reporting the type breakdown in the equilibria.

---

> ### Author Rebuttal · Authors · 2025-07-31
>
> Thank you for your detailed review and feedback. Based on your suggestions, we’ve extended our experimental analysis and added a new empirical investigation, which we believe will strengthen the paper - thank you for this suggestion! Please see our response to your questions and concerns below.
>
> > Real platforms seem to have a diversity of users, it seems like the results in the paper are discordant with real, e.g., recommendation systems. What explains this?
>
> This is an excellent question, but the answer is somewhat nuanced. As we state in the paper, our results should be taken to describe the *tendency* of population dynamics – when selective pressure is the primary force, when the learning algorithm is optimal retraining, absent any interventions, and at the limiting equilibrium. Hence, possible explanations for why a particular system at a given point in time can sustain several groups are:
> * **Time**: dynamics have not yet reached equilibrium, or converge very slowly.
> * **Learning**: the algorithm is not optimal (e.g., uses a proxy or finite data), or is designed to balance groups (e.g., via steering, regularization, or robust training).
> * **Intervention**: system administrators may intentionally intervene to facilitate diversity; for example, consider how firms invest significant resources in marketing (e.g., Meta reports investing ~$2B per year [1]) to ensure a constant influx of users from diverse social groups.
> * **Dynamics**: Other forces are at play, such as lack of alternatives, cross-group influence, rigidity of demand, non-linear heterogeneous utility, etc.
>
> Nonetheless, an important but subtle point is that when we observe a system consisting of several groups, **we do not observe other groups that could have been supported**, as they may have already been driven out. Thus, what we perceive as a mixed state may actually be a snapshot of a process in which “weaker” groups have already vanished, with others possibly to follow. This is in fact a likely outcome – **we have added to the paper a new experiment using the Folktables dataset** [2] that shows for $K=3$ how dynamics first diminish one group (while preserving balance between the others), and only then exhibit competition between the two remaining groups. We believe this perspective has important implications on fairness and its regulation.
>
> In biology, the question of coexistence is also prevalent, despite consensus that natural selection is the primary force at play. The study of coexistence therefore focuses on how it is sustained *despite* natural selection. This has revealed many biological and ecological mechanisms that can facilitate coexistence, such as resource partitioning, environment variation, and stabilization via intermediates. In this work we focus on learning itself as a possible stabilizing mechanism (Sections 5,6), which we view as a first step towards the study of coexistence in this context (as also noted by Reviwer oLKx). We also believe that prediction games in particular, and evolutionary game theory more generally, can aid future research in exposing additional mechanisms that can facilitate diversity in environments with learning.
>
> As per your request, we will add to the paper a detailed discussion on the practical implications of our results.
>
> > I did not feel the related work on evolutionary prediction games was sufficiently detailed.
>
> We will gladly extend our discussion of evolutionary game theory in the related work section, and in particular highlight the unique challenges evolutionary prediction games present in this context.
>
> > Intuitively, it seems like the conclusion of Thm 1 should hold quite generally and any setting where growth begets growth should yield these kinds of conclusions.
>
> While we agree with the conclusions of “growth begets growth”, proving that oracle learning algorithms imply this property is not trivial. This is due to two main reasons. First, to our knowledge, the structure of evolutionary prediction games is novel, as group fitness is defined implicitly as a solution of a loss optimization problem (see our answer below). Second, our results in Section 5 and Appendices D.7, D.8 show that the conclusion of Theorem 1 is surprisingly “brittle” with respect to the assumptions about the learning algorithm. We show that when the algorithm is limited in either computation, data, or consistency, the “growth begets growth” property doesn’t always hold. More broadly, evolutionary dynamics are very expressive, and able to express complex phenomena such as limit cycles, chaos, and at the extreme even simulate Turing machines [3]. Oracle learning algorithms are themselves complex in many ways, and it is interesting to see that they induce well-behaved competition dynamics.
>
>
> > The related work says “competitive exclusion” is well-studied, which suggests to me there should be related results.
>
> While competitive exclusion is a widely-observed biological phenomenon, evolutionary prediction games are unique in that the interaction between groups is induced by the solution (argmin) of a loss minimization problem. The implicit nature of optimization problems introduces complexity into the analysis of our game dynamics, which makes it difficult to draw on existing results. To our knowledge, there are no previously studied games in the evolutionary game theory literature that possesses this structure, and we believe that some of our techniques may be of independent interest.
>
> > (In the experiments) It would be nice to highlight qualitative take-aways or more nuances rather than just reporting the type breakdown in the equilibria.
>
> This is an excellent idea! As per your request, **we have extended our empirical analysis to include an in-depth analysis of learning dynamics and outcomes** for the existing experiments, as well as an additional experiment on new data (mentioned above). Results include:
> 1. **The effect of sampling noise:** As our theoretical analysis considers accuracy in expectation over the training set, it is interesting to understand the effects of sampling noise on long-term outcomes. Here we tested the effect of sample variance (controlled by fitting a Gaussian process to the outcomes of different sampled training sets) on long term outcome (dominance of group A vs. group B) for varying initial states $p^0_B \in [0.45,0.55]$. When there is no variance, group B dominates iff $p^0_B>0.5$ as expected. With the natural variance of the task (i.e., using the inferred Gaussian process covariance), results are relatively robust, and group B eventually dominates w.p. ~1 for all $p^0_B>0.51$. When variance is excessively increased (e.g., 5-fold), then outcomes also become noisy, and dynamics can converge to group A with some probability even for $p^0_B=0.55$.
> 2. **Time to convergence:** For unstable equilibria, a key question is how fast an equilibrium state is reached, and what affects this rate. Here we measured the time to reach (approximate) domination by one group as a function of the initial state $p^0$. For the CIFAR-10 dataset, results show that when $p^0_B\in[0.6,0.8]$ (or $[0.2,0.4]$), the time to convergence is roughly linear in $1-p^0_B$ (or $p^0_B$). However, as $p^0$ approaches the uniform distribution, time to convergence tends to infinity. The implication is that for initially balanced populations, natural selection forces are weak.
> 3. **Sensitivity analysis for stabilization:** Our stabilization mechanism (Section 5.2) requires access to the true equilibrium $p^\*$, which may not be available in practice.
> Our current experiments use an estimated $\tilde{p}^\*$ – and results comply with our theoretical findings – but this does not tell us how robust outcomes are to misspecification of $p^\*$. For this, we add a sensitivity analysis showing that the deviation of the reached state from the true $p^\*$ is linear in the estimation error of $\tilde{p}^\*$.
> An interesting observation is that misspecification affects only the population composition, and *not* welfare. This is because the algorithm ensures that the returned classifier acts “as if” under equilibrium, in which accuracy for both groups is the same.
> 4. **Additional dataset - Folktables**: We have added experiments on an additional social dataset, namely Folktables [2], which is based on a large collection of US census data. Here features describe individuals and labels describe their yearly income level. We used ‘state’ as a group (with the motivation that geography plays an important role in determining consumer sectors), and focused on three US states: New York, California, and Texas. Here we used both classifiers trained with XGBoost and logistic regression. We computed the induced vector field of the game and plotted both the population composition $p$ and group accuracies over time. As discussed above, our goal here was to showcase the intricate ways in which dynamics can progress in a multi-groups setting.
>
> We will add these results and a thorough discussion to the final version of the paper using the extra space.
>
> > What does it mean to be fitness maximizing?
>
> An equilibrium is fitness-maximizing if it pareto-dominates other equilibria with respect to fitness (i.e., the accuracy of each group is higher than its accuracy in any other equilibrium, and thus all groups are better-off if the system reaches this equilibrium). We will clarify this in the paper.
>
> –
>
> Finally, we would like to thank you again for the insightful and positive review! Please let us know if our response helps address your key concerns. If anything remains unanswered, or if additional questions arise, please let us know.
>
> References:
> * [1] Meta Platforms, Form 10-K - Annual report, 2024, SEC Accession No. 0001326801-25-000017. “Advertising Expense”.
> * [2] “Retiring Adult: New Datasets for Fair Machine Learning”, Ding et al., NeurIPS 2021.
> * [3] “No-Regret Learning in Games is Turing Complete”, Andrade et al., EC 2023.

---

> > ### Comment · Reviewer_Azxr · 2025-08-02
> > **Rebuttal response**
> >
> > Thanks for these comments. They have addressed my concerns. I've raised my scores.

---

### Official Review · Reviewer_oLKx · 2025-07-03

**Clarity:** 4
**Significance:** 4
**Originality:** 4
**Rating:** 5
**Confidence:** 4

**Summary:**

This paper introduces an interesting framework that the authors name evolutionary prediction games. The motivation is that distributions of trianing data are composed of users who choose to use the platform, which can often depend on the quality of predictions. Thus there is a feedback loop where the prediction model can shape the population of users and the population of users will impact the distribution of data the model continues to be trained on. The authors analyze this setting use a population game, coming from evolutionary game theory. Their Theorem 1 shows that repeatedly training with new data will lead to dominance of the groups that had highest fitness to begin with, essentially allowing them to dominate the platform. Their second theorem provides conditions, however, under which there can be coexistence. Furthermore, the authors provide experimental results on MNIST and CIFAR-10 demonstrating the theoretical results.

**Questions:**

Although there is discussion in the supplement around related work, I wonder if it would be possible to highlight more in terms of what the author's believe evolutionary potential games will allow us to achieve in terms of gains in production ML systems compared to previous literature?

Alluding to the weakness mentioned above, I wonder if there is any setting perhaps from a fairness benchmark dataset that would be closer to the motivations of the paper?

Given the results in Theorem 1 and Theorem 2, is there a situation in which the system could first evolve into domination by one group but by retraining grow to the mixed setting with multiple groups besides just changing to assumptions of Theorem 2?

**Ethical Concerns:**

["NO or VERY MINOR ethics concerns only"]

**Final Justification:**

It is great that the reviewers were able to provide a new experiment from the fairness benchmark dataset. I appreciate the authors directly addressing my question to include results from a fairness setting as well as the more in-depth analyses they provided for their existing experiments. I give high weight to this as my initial perceived weakness of the paper was the representativeness of the experiments. Given that the motivation, positioning, and theory of the paper were already solid and motivated my original positive assessment of the paper, I do believe it certainly deserves my already original positive review of a 5 and I will increase my confidence in this assessment. I recommend Accept.

**Limitations:**

Yes.

**Paper Formatting Concerns:**

None noticed.

**Quality:**

4

**Strengths And Weaknesses:**

The framing of evolutionary prediction games is really interesting and addresses a real issue faced by platforms that heavily rely on their prediction algorithms for user-retention (eg. TikTok). I really like it. The theorem are the natural type of result a reader would wish to see upon introduction of a new setting such as this, and the technical details are very well explicated in the main text and proven in the supplementary material. The technical results are relevant to the introduced setting but non trivial. Clear proof sketches and helpful interpretations are given for the main two theorems. Additionally, the figures are very clear and helpful. The experiments are simple but illustrative.

The main weakness that stands out to me is that the experimental results while illustrative of the theoretical results, don't immediately appear to be very representative of the main motivations for this framework in practice. Although it would of course be difficult to obtain a platform's worth of such data, an experiment more closely mimicking such settings would be very compelling.

---

> ### Author Rebuttal · Authors · 2025-07-31
>
> Thank you for the positive and thoughtful feedback! We address your questions below, including new experiments based on your suggestions:
>
> > I wonder if it would be possible to highlight more in terms of what the authors believe evolutionary potential games will allow us to achieve in terms of gains in production ML systems compared to previous literature?
>
>
> Evolutionary prediction games can help designers and operators of ML systems anticipate performative population shifts, and integrate this foresight into their decision-making. One key message is that applying conventional retraining risks favoring some groups over others in the long term, even if unintentionally. Evolutionary prediction games can help understand which groups are at risk, why they are at risk, and highlight possible avenues for interventions, if such are needed. One example is to decide on using a stabilizing learning algorithm (such as the one we propose in Prop. 2) instead of retraining to help steer towards desired states. Another is to target the retention of at-risk groups, either by providing benefits or reducing costs (e.g., via subsidies), in order to improve their fitness. A third is to understand if and when to increase the influx of certain groups, e.g. via marketing efforts, to balance the natural tendencies of the system. We will extend the discussion to emphasize these insights.
>
>
>
> > Alluding to the weakness mentioned above, I wonder if there is any setting perhaps from a fairness benchmark dataset that would be closer to the motivations of the paper?
>
> Thank you for this question! Based on your suggestion, **we have conducted a new experiment with the Folktables dataset** [1] used commonly in the fairness literature. We detail them below, and will gladly add these new experiments and a discussion of their results to the paper.
>
> In more detail, we use the ACSIncome prediction task, and associate groups with data originating from different states in the US, with the idea that firms operate in different states but train a model on data aggregated from all sources. We simulate replicator dynamics with $K=3$ using states NY, TX, and CA, and initialize with uniform $p^0$. We explore both XGBoost and logistic regression as learning algorithms.
>
> We use this new dataset to explore the implications of population dynamics on fairness. Results show that population dynamics exhibit two distinct dynamical phases: an initial phase where one group diminishes (while the other two remain balanced with similar accuracies), and a second phase in which the two remaining groups compete (until one of them dominates). Our results here highlight a shortcoming of static fairness definitions: auditing the system towards the end of the first phase will indicate that fairness holds – but only in appearance, since members of the two large groups are easily observed, but the third group has almost vanished. For further discussion please see our answer to Reviewer YgUX.
>
> > Given the results in Theorem 1 and Theorem 2, is there a situation in which the system could first evolve into domination by one group but by retraining grow to the mixed setting with multiple groups besides just changing to assumptions of Theorem 2?
>
>
> Our results state that for an *oracle* learning algorithm satisfying the conditions of Theorem 1, dynamics always converge towards single-group domination (with the only technical exception being the “equivalent groups” case discussed in Appendix D.5.1). This means that dynamics which first evolve towards single-group dominance and then towards coexistence cannot be induced for oracle algorithms, which are characterized by Theorem 1.
>
> However, this need not necessarily hold under other learning algorithms. One example is when the algorithm itself is allowed to vary over time. For example, in Section 5.2 we show that a mixed population setting can be induced by increasing the weight of underserved groups in the retraining of the optimal algorithm. It is possible to design a reweighing scheme which only “activates” once the population reaches a certain level of single-group dominance, and the dynamics induced by such an approach seem to resemble the scenario you describe.
>
> --
>
> Finally, we would like to thank you again for the positive and encouraging review, and for the great suggestions. If any questions remain or arise, please let us know!
>
> References:
> * [1] “Retiring Adult: New Datasets for Fair Machine Learning”, Ding et al., NeurIPS 2021.

---

> > ### Comment · Reviewer_oLKx · 2025-08-02
> >
> > Thank you for the informative and thorough responses. I will stick with my original,  positive assessment of a 5.

---

### Author Response · Authors · 2025-08-05

Thank you for the encouraging feedback, and for the very helpful suggestions! The new insights and results will appear in the revised paper.

---

### Decision · Program_Chairs · 2025-09-17

**Decision:**

Accept (poster)

**Comment:**

This paper introduces a novel theoretical framework called evolutionary prediction games for studying feedback loops between ML prediction accuracy and user population dynamics.  Their motivation stems from the idea that prediction models can shape the population of users engaging with an ML platform, and the population of users will, in turn, impact the distribution of data the model continues to be trained. They present two theoretical results on the equilibria of the games, which show that (1) repeatedly training with new data will lead to dominance of the groups that had the highest fitness to begin with, essentially allowing them to dominate the platform, and (2) present certain conditions under which there can be coexistence between different groups. Experiments on toy datasets MNIST (w/ label noise) and CIFAR-10 (w/ augmentations) demonstrate the theoretical findings on equilibria conditions for two user groups in each setting and further demonstrate the effectiveness of the proposed stabilization algorithm for beneficial coexistence between groups.

All reviewers agree on the significance [oLKx, Azxr, iKna], usefulness of the approach [oLKx, iKna], technical clarity and soundness [oLKx, Azxr, YgUX], and originality [Azxr, YgUX] of the approach. The main limitations of the work are listed as experiments not being close to real-world ML production systems [oLKx, Azxr] and the need for clarification in certain assumptions [iKna, YgUX]. During the rebuttal, the authors claim to have conducted additional experiments on a fairness dataset (Folktables) and explored the implications of population dynamics on fairness with three different user groups derived from real-world US census data. The insights from this experiment confirm the theoretical findings that a false appearance of fairness can emerge between two groups while one is completely driven out by the dynamics of the game. The authors indicate that such implications from their theory can be beneficial for different real-world applications to monitor which groups survive in the long run. All reviewers have expressed that most of their concerns have been successfully addressed by the authors during the discussion period.

The authors must include the newly presented results/insights on the more realistic fairness dataset and update the final version of their paper with the references and clarifications they have addressed in discussion with all the reviewers (additional references, clarification of Nash stationarity assumption, discussion on insights of prediction algorithms, implications on real-world applications and ML production systems).